# Millennial-to-orbital-scale subsurface ocean warming and Polynya formation off Dronning Maud Land during the last glacial

**Tainã M. L. Pinho** [1,2] ✉, **Dirk Nürnberg** [3], **A. Nele Meckler** [4], **Gesine Mollenhauer** [1,2,5], **Juliane Müller** [1,5], **Gerrit Lohmann** [1,5], **Lester Lembke-Jene** [1], **Salma Hidayat** [2], **Vincent Rigalleau** [1], **Frank Lamy** [1] & **Ralf Tiedemann** [1]

We present a millennial-scale multi-proxy reconstruction of changes in properties of the upper water column near the East Antarctic ice shelf based on planktonic foraminifera from a unique sedimentary archive spanning the glacial period from 75,000 to 20,000 years. Our results imply that variations in the thermohaline structure between Antarctic Surface Water and Warm Deep Water (WDW) may have resulted in either strengthening the stratification of the upper water column or promoting polynya formation (convective overturning). Oceanic subsurface warming during glacial Antarctic stadials and periods of low obliquity, combined with increased salinity and nutrient content, suggests the breakdown in stratification and polynya presence. This glacial polynya formed off Dronning Maud Land (DML) reflects a hybrid coastal-open-ocean polynya mode. We attribute the development of the Glacial DML Polynya to sea-ice induced subsurface warming of WDW and a decrease in density stratification in combination with circulation changes in the atmosphere and ocean. The polynya-driven oceanic heat release during the glacial stadials may have increased the moisture supply to Antarctica and thus promoted the accumulation of ice and the thickening of an advancing ice sheet at the continental shelf margin.

A special feature of the Southern Ocean south of 60°S is the relatively low stability of the thermohaline structure of the upper layer[1], which reacts sensitively to even small changes in the climate system with far-reaching consequences for the atmospheric-oceanic gas and heat exchange, the global overturning circulation, the ventilation of the deep ocean, as well as the carbon cycle[2,3]. Today, the stratification of the upper water column is characterized by the difference in density between the cold and fresh Antarctic Surface Water (ASW) and the underlying warm and salty Circumpolar Deep Water (CDW) (Fig. 1a, c, d). The CDW is hereafter referred to as the Warm Deep Water (WDW), a

derivative of the CDW that is advected poleward within the eastern limb of the Weddell Gyre[4] (Fig. 1a). The ASW/WDW boundary marks the Antarctic Slope Front (ASF)[5]. In the Weddell Sea off Dronning Maud Land (DML) (Fig. 1c, d), the ASF is located at ~200 m water depth and, in response to increased easterly wind stress, dips to ~700 m at the continental slope, below the shelf edge (~500 m) (Fig. 1c, d). This position protects the adjacent ice shelves (Ekström, Jelbart, Fimbul)[6–8] from onshore WDW intrusion and accelerated melting. The ASW is ~3.5 °C colder and ~0.5 salinity units fresher than the underlying WDW. Despite the generally equal contributions of temperature and salinity

[1]Alfred Wegener Institute for Polar and Marine Research, Bremerhaven, Germany. [2]University of Bremen, Faculty of Geosciences, Bremen, Germany. [3]GEOMAR Helmholtz Centre for Ocean Research Kiel, Kiel, Germany. [4]Bjerknes Centre for Climate Research and Department of Earth Science, University of Bergen, Bergen, Norway. [5]University of Bremen, MARUM, Bremen, Germany. ✉e-mail: taina.pinho@awi.de

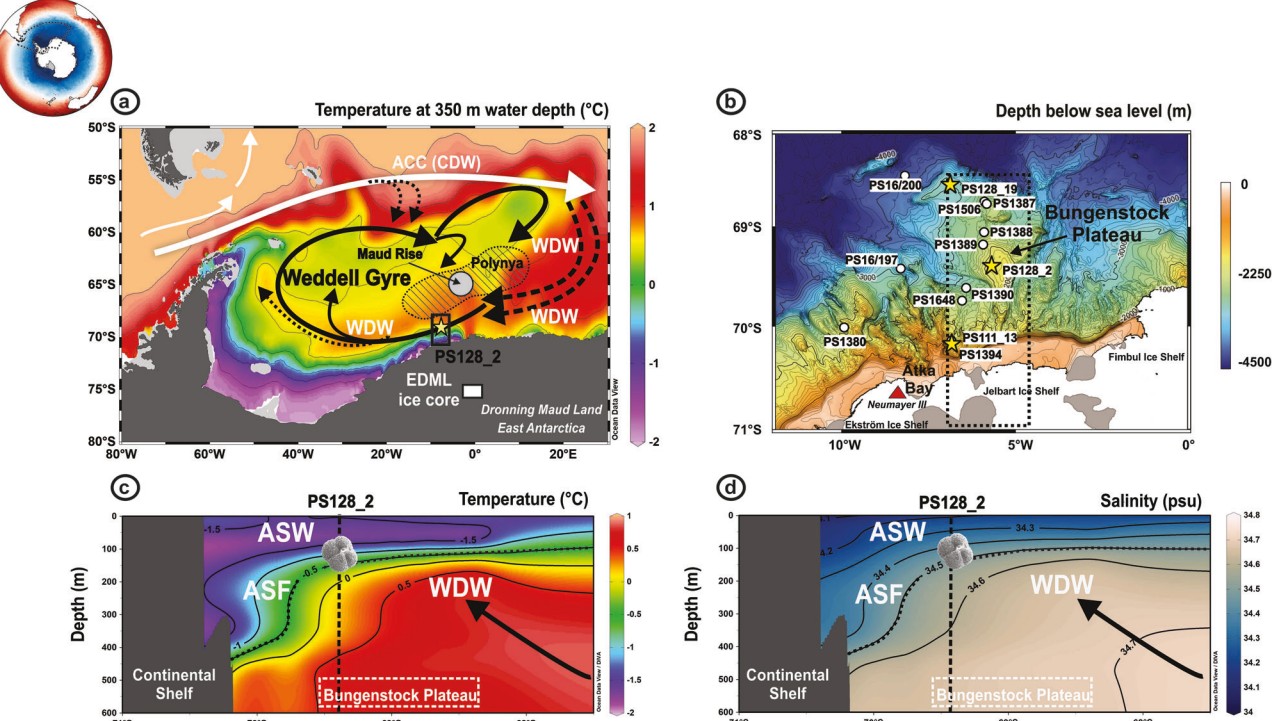

**Fig. 1 | Schematic of modern subsurface water mass circulation in the Weddell Gyre. a** Modern annual subsurface temperature distribution at 350 m water depth in the Weddell Gyre[43]. The area of the extremely large Weddell Polynya of the 1970s is shown by the hatched area[44]. The yellow star indicates the location of core PS128_2 (69.41°S, 5.59°W; 1868 m water depth) and the white rectangle the EPICA Dronning Maud Land (EDML) ice core location. **b** Bathymetric map of our study area. For reference, nearby sediment cores from previous RV *Polarstern* expeditions related to the presence of the Weddell Polynya during glacial periods are shown on the bathymetric map[27,29,45]. **c, d** Temperature[43] and salinity[46] profiles for the upper 600 m water depth from a N-S transect covering regions of Atka Bay

and Bungenstock Plateau, as shown in the bathymetric map, panel b. Temperature[43] and salinity[46] data were retrieved from Word Ocean Atlas 2023, which provides observationally based climatologies. ACC Antarctic Circumpolar Current, ASF Antarctic Slope Front ASF, CDW Circumpolar Deep Water, WD Warm Deep Water. The thick black dashed line shows the pycnocline (identical with the ASF). The vertical dashed line represents the location of core PS128_2. Today, the habitat depth of *N. pachyderma* sin. reflects the water mass signature of the ASW as shown in (**c**) and (**d**). Maud Rise is represented by a white circle. The dashed rectangle indicates the latitudinal extent of the Bungenstock Plateau feature, which is deeper than -600 m water depth.

to density and stratification, salinity is the main contributor in polar regions[1]. Even small perturbations in the forcing and feedback mechanisms that affect mainly the salinity budget can strengthen or completely destroy the stratification. In the course of global warming, numerous studies have shown that the current forcings, such as a southward shift and intensification of the subpolar westerly winds as well as increased meltwater release, have already led to stronger water column stratification driven by surface freshening and subsurface warming[9–11]. Observations and model simulations indicate that the more southerly position of strong westerlies weakens the easterly winds around Antarctica, which would reduce the ASF strength, and thus allow WDW to penetrate across the shelf and to melt the floating ice shelves from below[8,12]. The associated release of meltwater would further strengthen the stratification and finally interrupt the brine rejection in polynya regions and thus the formation of Antarctic Bottom Water (AABW)[13–15]. This would have a major impact not only on the dynamics of the lower limb of the global overturning circulation but also on the ventilation in the abysses of the world's oceans. On the other hand, the current warming trend was briefly interrupted by an opposite, destratification scenario. During the winters of the mid-1970s, a combination of processes, including density-driven instabilities via interactions with topographic features such as Maud Rise[16,17], atmospheric forcing[18,19] and atmosphere-ice-ocean dynamics[20] led to the breakdown of the stratification and the formation of the Great Weddell Polynya (maximum extent of 350,000 km², Fig. 1a)[21]. This polynya ventilated the deep ocean by convective overturning to depths of -2700 m and extended from the Maud Rise southwestward

along the Bungenstock Plateau (e.g., refs. 22,23). It has been argued that the overturning has maintained the open-ocean polynya through the massive release of heat from the deep ocean, bringing WDW into the surface layer, which may have limited the formation of winter sea-ice compared to periods without polynya formation[22,23]. Model simulations suggest that deep convection in the Weddell Sea via open-ocean polynyas was more active in the colder past, such as during glacials, and has been weakened by anthropogenic forcing[23,24].

The recent interactions described above reflect a complex balance of feedbacks that can stabilize and destabilize the Antarctic ice sheet and the Southern Ocean overturning circulation, but can also be subject to change. Investigating past changes in the stratification of the polar upper-ocean can therefore provide insights into understanding the interplay between meltwater discharge and ocean heat transport, which is critical for predicting future climate change. So far, however, little is known about the properties of the seawater adjacent to the East Antarctic continental slope, the forcing factors, and the atmosphere-ice-ocean interactions that have controlled the dynamics of the ASW and WDW in the past.

Here, we reconstruct past orbital- and millennial-scale variations in upper-ocean stratification from sediment core PS128_2, recovered from the Bungenstock Plateau off DML in the eastern Weddell Sea (Fig. 1). Our proxy record is from -70 km north of the shelf break and covers the last glacial period from ~75–20 kilo annum (ka) before present (BP). We track past vertical shifts in WDW extent by reconstructing changes in subsurface temperature and salinity, and test the hypothesis whether the Great Weddell Polynya of the mid-1970s has

resembled a typical feature of the last glacial period[24]. The presence of an open-ocean glacial polynya and the associated ice-ocean-atmosphere feedback mechanisms are challenging to reconstruct and have never been demonstrated with palaeoceanographic data.

For the last glacial period, the sediment archive on Bungenstock Plateau provided an exceptionally continuous occurrence of abundant and well-preserved calcitic foraminifers (e.g., refs. 25–29), which enabled the use of multiple carbonate proxies. This is rather rare in sediments around Antarctica. We measured stable oxygen and carbon isotopes ($\delta^{18}O$, $\delta^{13}C$), Mg/Ca ratios, and clumped isotopes ($\Delta_{47}$) on the calcitic shells of the subsurface-dwelling planktonic foraminiferal species *Neogloboquadrina pachyderma* sinistral (*N. pachyderma* sin.). The use of these proxies was limited to the last glacial period and, unfortunately, not applicable to the deglacial and Holocene periods. The low number of well-preserved planktonic foraminifera in the deglacial and Holocene sediment samples (e.g., refs. 25–29) only permitted a low-resolution $\delta^{18}O$ record to be generated but prevented the measurement of further Mg/Ca and clumped isotopes ($\Delta_{47}$) proxies. For the glacial period, however, the abundant occurrence of planktonic foraminifer allowed us to measure $\Delta_{47}$, Mg/Ca, $\delta^{18}O$, and $\delta^{13}C$ and reconstruct past variations in ocean temperature, salinity, and nutrients recorded at the preferred habitat-depth of *N. pachyderma* sin. at ~50–150 m[30,31]. For the temperature reconstruction and its verification, we measured two independent parameters, $\Delta_{47}$ and Mg/Ca. With this approach, we aimed to test whether the often too warm Mg/Ca-based temperature reconstructions for cold polar marine environments (refs. 32–38) could be corrected with $\Delta_{47}$ data in order to solve a long-standing problem of polar paleoceanography. Ten $\Delta_{47}$-based temperatures are paired with independent Mg/Ca data to support and verify the absolute Mg/Ca-derived subsurface temperature range, which is made possible by good agreement between the two independent temperature proxies (see Methods). Our approach resulted in modified $\Delta_{47}$-based subSurface Temperature (subST) Mg/Ca variations, hereafter referred to as $subST_{Mg/Ca\ vs.\ \Delta47}$, with lower temperatures and amplitudes compared to other temperature calibrations (see Supplementary information). From 75 to 20 ka, $subST_{Mg/Ca\ vs.\ \Delta47}$ vary between 1.9 and −2.75 °C. Considering the freezing point of sea water, the minimum value appears about 0.7 °C too low, which, however, lies within the error range of our reconstruction. The $\Delta_{47}$-based temperature data are in good agreement with the $subST_{Mg/Ca\ vs.\ \Delta47}$ reconstruction (see Methods: r = 0.75). For the salinity reconstruction, the correction of the foraminiferal $\delta^{18}O$ record for changes in temperature ($subST_{Mg/Ca\ vs.\ \Delta47}$) and global ice volume[39] is used as a qualitative proxy for seawater salinity ($\delta^{18}O_{sw-ivc}$)[40]. The planktonic $\delta^{13}C$ ratio serves as a proxy for the dissolved inorganic carbon isotope composition ($\delta^{13}C_{DIC}$) of the surrounding water, which provides information on its nutrient inventory and source water mass. For example, the Pleistocene $\delta^{13}C_{DIC}$ values of Antarctic WDW range below 0.6 ‰, while those of the ASW are significantly higher[41,42].

The chronology of core PS128_2, from 75 to 20 ka, is based on a combination of Accelerator Mass Spectrometry (AMS) radiocarbon ($^{14}C$) datings on *N. pachyderma* sin. and planktonic $\delta^{18}O$ stratigraphy (Supplementary Information and Table 1). In order to better document the reliability and quality of our age model towards the end of the last glacial period, we also show the planktonic $^{14}C$ datings and $\delta^{18}O$ data (core PS128_2) for the last 20 ka, which document the deglacial climate transition into the Holocene. In addition, we used a core-to-core correlation to supplement the Holocene stratigraphy of core PS128_2 (upper 0.5 m) with $^{14}C$ datings from core PS111_13, which lies 90 km further north on the continental slope (Supplementary Information, Supplementary Fig. 1, Supplementary Table3 1). All foraminifers from the interval 20-0 ka were used to create an age model for core PS128_2, which unfortunately excluded the measurement of further calcitic proxies like $\Delta_{47}$ and Mg/Ca.

## Results and Discussion

### Glacial upper-ocean stratification

Our *N. pachyderma* sin. $\delta^{18}O$ record ($\delta^{18}O_{NPS}$) from core PS128_2 shows significant variability on orbital timescales, primarily attributable to changes in global ice volume (Fig. 2a, LS16, Lisiecki and Stern[47]). However, the magnitude to which variations can be related to changes in local seawater temperature or salinity remains often uncertain. The $\delta^{18}O_{NPS}$ record is not cleary related to changes in Antarctic air temperature, as reflected by the EPICA Dronning Maud Land (EDML) $\delta^{18}O$ ice core record[48] (Figs. 2a and 3a), nor to changes in $subST_{Mg/Ca\ vs.\ \Delta47}$ (Fig. 3c). Important to note, however, is that our $subST_{Mg/Ca\ vs.\ \Delta47}$ record shows orbital-scale and millennial-scale subsurface warmings of 1–2 °C, which are anticorrelated with Antarctic air temperatures[48] (Fig. 3a, c). This anticorrelation is robust despite chronological uncertainties in the age model of core PS128_2. The anticorrelation is pronounced during the time interval 75–38 ka BP, given the comparatively higher sedimentation rate, which is twice as high as during the interval from 38–20 ka BP (Fig. 2b, Supplementary Fig. 2d), allowing to examine millennial-scale changes with improved temporal resolution. In the time interval ~38–20 ka BP, the temporal resolution and density of measurements is partly too low to detect a clear anticorrelation between $subST_{Mg/Ca\ vs.\ \Delta47}$ warming and air temperature cooling on millennial timescales (Fig. 3a, c). Such temperature inversions, characterized by colder air and sea surface temperatures but warmer subsurface temperatures, are currently known from regions south of 60°S around Antarctica[49].

Superimposed on the millennial-scale $subST_{Mg/Ca\ vs.\ \Delta47}$ variability are patterns of orbital-scale variability that appear to be associated with cyclic variations in Earth's obliquity (Fig. 3c). This is not entirely surprising, since changes in obliquity cause variations in seasonality and strongly affect the total summer energy budget received at high latitudes with a period of ~41 thousand years. The interval from ~38–20 ka BP, which includes the Last Glacial Maximum (LGM; ~26–19 ka BP, ref. 50) and the penultimate glacial Marine Isotope Stage (MIS) 4 (~71–59 ka BP, ref. 51) are both characterized by long-term $subST_{Mg/Ca\ vs.\ \Delta47}$ maxima coincident with minima in obliquity (Fig. 3c). The long-term $subST_{Mg/Ca\ vs.\ \Delta47}$ minimum in between is associated with an obliquity maximum.

The variability of $subST_{Mg/Ca\ vs.\ \Delta47}$ and $\delta^{18}O$-derived ice-volume corrected salinity proxy ($\delta^{18}O_{sw-ivc}$) is mainly in-phase, indicating warm and saline subsurface conditions, while subsurface cooling events are associated with fresher conditions (Fig. 3c, d). Warm and saline subsurface conditions occur during drier and colder Antarctic climate intervals[20]. This suggests that, in addition to changes in ice volume, salinity in particular controls the $\delta^{18}O_{NPS}$ signal in the vicinity of the East Antarctic ice sheet (EAIS), despite the opposing effect of temperature on the calcite $\delta^{18}O$ signal (Fig. 2a, Fig. 3c–e).

The $\delta^{13}C$ record of *N. pachyderma* sin. provides additional information on the signature of subsurface waters. Decreases in $\delta^{13}C$ likely reflect increases in nutrient content, provided that the effects of variations in biological productivity and air-sea carbon exchange play a minor role[61]. Low $\delta^{13}C$ values (high nutrient contents) predominantly correlate with high $subST_{Mg/Ca\ vs.\ \Delta47}$ and salinity conditions during the glacial period (Fig. 3c, d, f). This combination suggests the presence of WDW at the preferred habitat depth of *N. pachyderma* sin. at 50–150 m[19,20], while the opposite would characterize the fresh, cold, and nutrient-depleted ASW at this level. In general, an uplift of the ASF, e.g., from its modern position of ~200 m into the habitat of *N. pachyderma* sin. (50–150 m) is registered by an increase in subsurface temperature and salinity, which, in combination with low $\delta^{13}C_{DIC}$ ratios, is interpreted as an increased influence of WDW.

Our reconstructed vertical changes in the presence of ASW versus WDW are considered robust, even when taking into consideration some variability in the habitat depth of *N. pachyderma* sin[19,20]. The comprehensive study by Greco et al.[31] provides no

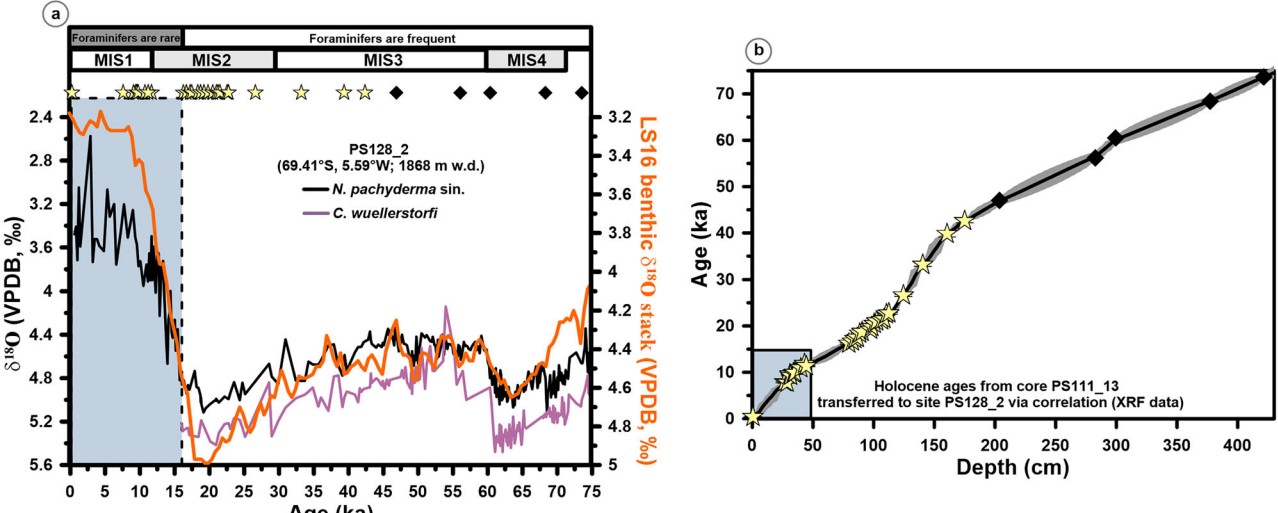

**Fig. 2 | Stratigraphy of sediment record PS128_2. a** Comparison of planktonic (*Neogloboquadrina pachyderma* sin., black) and benthic (*Cibicides wuellerstorfi*, purple) δ[18]O records of cores PS128_2 with the stratigraphic reference record LS16 Deep South Atlantic δ[18]O stack (orange)[47]. For *C. wuellerstorfi*, we apply a 0.64‰ interspecific offset to compensate for seawater disequilibrium[52]. **b** Age (ka)-depth (cm) relationship based on our age model (Supplementary Information, Supplementary Fig. 2). Stars on top of panel "a" depict calibrated radiocarbon ages ([14]C datings) and black diamonds depict stratigraphic δ[18]O tie-points. The Holocene [14]C

datings have been transferred from core PS111_13 to core PS128_2 by correlating their XRF records (see Supplementary Information, Supplementary Fig. 1, Supplementary Tab. 1). The blue area spanning the last 15 ka indicates the typical sedimentary sequence marked by a sharp limitation of biogenic carbonate due to low productivity of foraminiferal shells and their preservation at our site location[25–27] (further details on the Supplementary Information). MIS Marine Isotope Stages, which refer to past periods defined by δ[18]O values, representing warm and cold climate intervals, are used as stratigraphic markers[47] (see panel "a").

evidence for an effect of salinity, temperature, and stratification on the preferred habitat depth of *N. pachyderma* sin. at 50–150 m water depth. However, they found a relationship with changes in sea-ice cover and surface chlorophyll concentrations that can slightly alter the habitat depth. Accordingly, with low to moderate sea-ice cover and moderate chlorophyll concentrations such as today, *N. pachyderma* sin. would record the temperature in a possibly slightly deeper habitat of 75–150 m[31], still consistent with our assessment of the current habitat depth of ~120 m (see Supplementary Information, Supplementary Fig. 4). In contrast, *N. pachyderma* sin. would prefer a shallower habitat depth of 50–100 m when sea-ice cover is present throughout the year, and chlorophyll concentrations are very low. The same habitat is also proposed for areas with high chlorophyll concentrations and very low sea-ice cover[31], like polynyas. Therefore, it is likely that the pronounced subST$_{Mg/Ca\ vs.\ \Delta47}$ maxima during the MIS 4 and LGM indicate both the strongest warming of the WDW and its strongest expansion into shallower water depths, which would favor the development of a polynya system off DML. Although the Antarctic *N. pachyderma* genotype may be adapted to sea ice conditions[62], strong agreement between the independent temperature proxies $\Delta_{47}$ and Mg/Ca (Supplementary Fig. 8), corroborated by *N. pachyderma* sin. δ[13]C (Fig. 3c, f), and supported by our apparent calcification depth estimate (Supplementary Fig. 4), provides evidence that our foraminiferal geochemical proxies are not affected by sea ice (refs. 33,63).

As we have no proxy information on past changes in surface water conditions, we cannot yet exclude the possibility that during full glacial conditions with year-round sea-ice cover, a density-driven barrier still existed between the very surface and the sea-ice related shallower habitat of *N. pachyderma* sin. (~50–100 m, Greco et al.[31]), separating a potential thin layer of ASW from WDW and preventing wind mixing. However, a complete, multi-year sea-ice cover would have shut down biological productivity and thus also the presence of planktonic foraminifera[26,64], which is in sharp contrast to the high abundance of foraminifera in our glacial sediments.

## Occurrence of the glacial DML Polynya

During the LGM and partly also during MIS 4, the geographic extent of ice sheets and glacier cover was the largest in the last 75,000 years. These time intervals correspond to low-obliquity phases with exceptionally cold Antarctic air temperatures during the last glacial period (Fig. 3a, c). An advancing and slightly elevated EAIS likely extended to the DML continental shelf break[65,66] and most of the Weddell Sea was covered by sea-ice year-round[67] (Fig. 3g). Such conditions would have intensified katabatic winds[68] in the nearby sea-ice-covered marine environment and favored the formation of glacial polynyas in the open ocean. During several intervals of the last glacial period, specifically ~38 – 20 ka BP, (including the LGM), 43–40 ka BP, 52–48 ka BP, ~58–55 ka BP, 72–63 ka BP and 75–73 ka BP (Fig. 3), the cold and fresh ASW likely did not dominate the upper 200 m of the water column above the Bungenstock Plateau, as it does today. Instead, pronounced increases in subsurface temperature and salinity combined with decreases in δ[13]C$_{DIC}$ (Fig. 3c, d, f) indicate times when the influence of WDW was closest to the ocean surface. We interpret this vertical redistribution of heat as an expression of upwelled WDW that thermodynamically induced the formation of an open-ocean polynya off DML by potentially melting existing sea-ice and preventing further sea-ice to be formed[20]. As the WDW is characterized by a relatively high salinity, its injection into the surface layer would further destabilize the water column, as the upwelled saline water is cooled by the atmosphere, becomes denser, and returns to depth. We refer to this as the Glacial DML Polynya. To evaluate whether deep convection occurred, we compared our δ[18]O values of benthic and planktonic foraminifera from core PS128_2 (Fig. 2a). The average benthic-planktonic δ[18]O difference is 0.3 ± 0.1‰. This small offset suggests that convective mixing and cooling of the deep ocean possibly did not penetrate to 2000 m depth, at least at our site. However, this interpretation is uncertain because the observed offset is close to the combined analytical and correction uncertainties. The measurement precisions for benthic and planktonic δ[18]O together amount to ±0.12‰, and the uncertainty in the species-specific correction for benthic δ[18]O adds ~0.1‰. Given that these errors

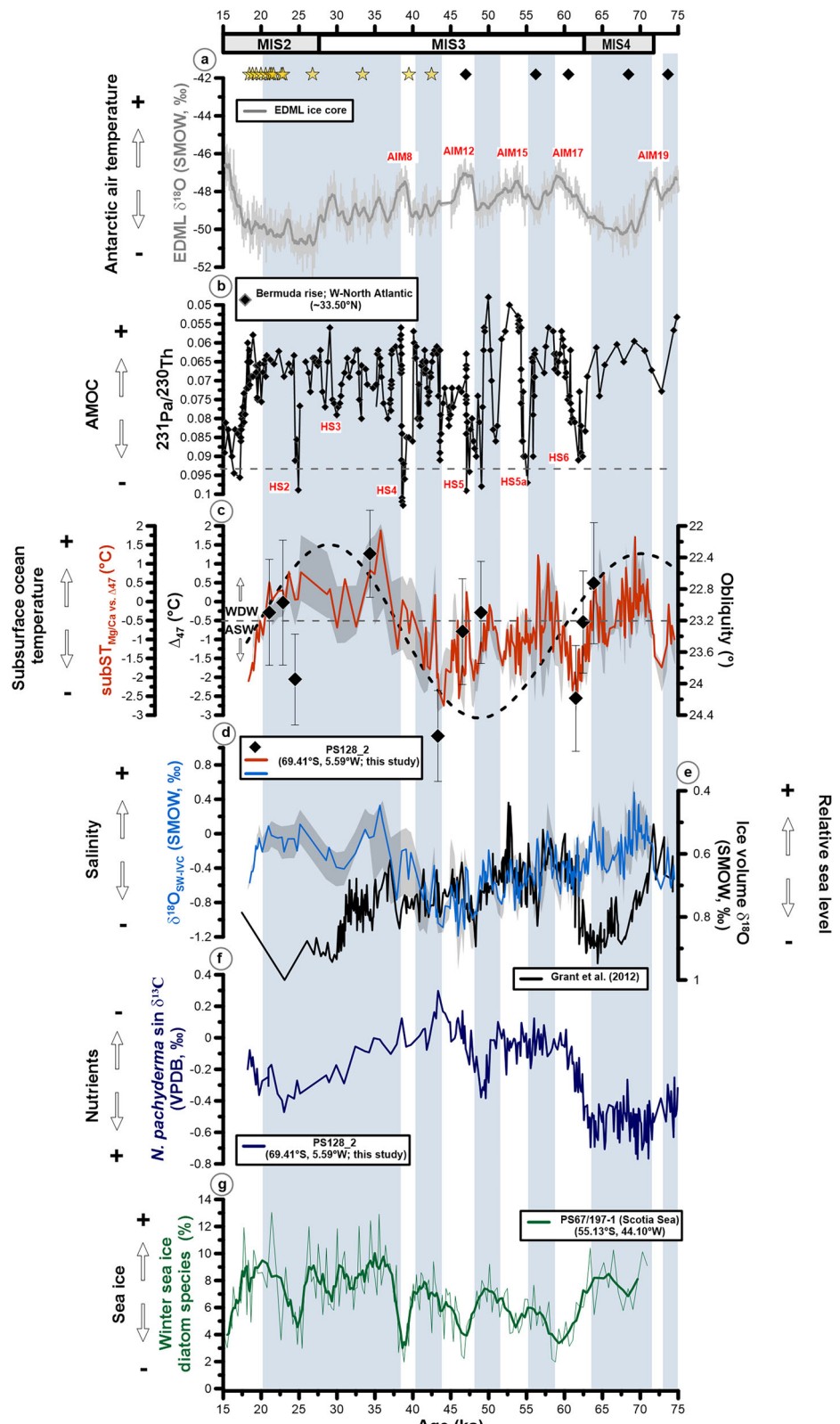

are similar in magnitude to the observed offset, the benthic–planktonic δ18O gradient cannot be interpreted with confidence. Accordingly, we refrain from the use of benthic and planktonic δ18O for stratigraphic purposes only (Fig. 2a).

In addition to our reconstructed shoaling of WDW during glacial/cold periods, the properties of this water mass could have changed over time. One proposed mechanism is again related to sea-ice

processes[69,70]. Haumann et al.[70] used model experiments to reconstruct the thermal development of the surface and subsurface of the Southern Ocean from the 1980's to the early 2010's in relation to sea-ice derived freshwater fluxes. Their results suggest that enhanced wind-driven northward sea-ice export, which intensified the extraction of freshwater near Antarctica and its release into the open ocean, strengthened the salinity-driven upper-ocean stratification between

**Fig. 3 | Proxy comparison of our results with marine and ice core records since the last glacial period. a** EPICA Dronning Maud Land (EDML) $\delta^{18}O$ ice core[48]. **b** $^{231}Pa/^{230}Th$ record from Bermuda Rise in the North Atlantic[53–56] showing AMOC strength variations with AMOC shutdowns when $^{231}Pa/^{230}Th$ is >0.093[57] as shown by the dashed horizontal line. **c**, Subsurface temperature derived from clumped isotopes $\Delta_{47}$ shown for reference (black diamonds). Mg/Ca temperature record based on the calibration between $\Delta_{47}$ and Mg/Ca ratios (subST$_{Mg/Ca\ vs.\ \Delta_{47}}$) (Supplementary Information). The black shaded area and error bars show the Monte Carlo-based confidence intervals (95% Confidence Interval (CI) for subST$_{Mg/Ca\ vs.\ \Delta_{47}}$, and 68 % CI for $\Delta_{47}$ temperatures). The dashed line shows obliquity (°)[58]. The horizontal dashed line shows the modern temperature boundary between WDW and ASW. **d**, Ice-volume-corrected $\delta^{18}O_{sw}$ record ($\delta^{18}O_{sw-ivc}$) indicative of changes in salinity. The black shaded area shows the calculated $\delta^{18}O_{sw-ivc}$ by considering the 95% Monte Carlo CI from subST$_{Mg/Ca\ vs.\ \Delta_{47}}$. **e** Global ice-volume $\delta^{18}O$[39] used to correct $\delta^{18}O_{sw}$. **f** Planktonic $\delta^{13}C$ record from core PS128_2 reflects changes in subsurface nutrient conditions. **g** High percentages of winter sea-ice diatom species (*Fragilariopsis curta* and *Fragilariopsis cylindrus*, combined as the *F. curta* group) are indicative of stronger local winter sea-ice influence in the Scotia Sea at core PS67/197-1 (more frequent/longer seasonal sea-ice cover, proximity to the winter sea-ice edge, and associated water-mass conditions)[59,60]. Stars on top of "**a**" depict calibrated radiocarbon ages ($^{14}C$ datings). Black diamonds depict stratigraphic $\delta^{18}O$ tie-points between core PS128_2 and the LS16 $\delta^{18}O$ reference record[47]. Blue shaded bars show orbital and millennial-scale reductions in the upper-ocean stratification and shallowing of WDW. AIM = Antarctic Isotope Maximum. HS = Heinrich Stadials. MIS = Marine Isotope Stages.

the Subantarctic Front and the sea-ice edge. As a result, reduced vertical mixing led to the retention of ocean heat and a warming of ~0.5 °C in the WDW between ~100 and 500 m water depth south of 55°S. This effect may have been even more pronounced during the LGM, as the extensive and partly perennial sea-ice cover in the Southern Ocean[71] likely promoted the buildup of heat at the subsurface[69]. This is supported by the relatively good agreement between the long-term subsurface temperature maxima and maxima of diatom-derived winter sea-ice concentrations based on the relative abundance of diatoms species in the Atlantic sector of the Southern Ocean at ~55°S[59,60] (Fig. 3c, g), with the exception that the subsurface temperature decline at 20 ka BP preceded the winter sea-ice decline by ~3 kyr. Nonetheless, we assume that the described sea-ice-driven process[70] contributed to the extent of WDW warming in our study area, although it does not necessarily explain the reconstructed expansion of the WDW into shallower water depths during MIS 4 and 2. However, strong subsurface warming combined with a shallow pycnocline would reduce the density of the WDW and weaken the stratification that allows the ASF and the sea-ice to exist[72]. This mechanism has some similarities with the hypothesis of Dokken et al.[69] in the Nordic Seas, according to which warm Atlantic deep water is isolated under a fresh surface layer during cold stadials, until pycnocline breakdown and sea-ice retreat allow convective overturning during the interstadials and regional warming.

Alternatively, warmer WDW temperatures may reflect a stronger southward transfer of CDW heat across the Antarctic Circumpolar Current (ACC) in the Atlantic–Indian sector of the Southern Ocean during MIS 4 and 2. Wu et al.[73] suggested that under low obliquity conditions, a more northerly ACC position around ~45°S placed the current directly under an equatorward-displaced Southern Hemisphere westerly belt and coincided with enhanced meridional density contrasts. These changes likely increased isopycnal tilt, strengthening the glacial ACC in the Indian sector[73]. In addition, a coeval reduction in the leakage of the Agulhas Current into the South Atlantic would have promoted the buildup of heat and salinity in the South Indian Ocean, further amplifying the meridional density gradient[73]. Taken together, stronger density gradients and increased surface heat-flux anomalies associated with sea-ice expansion and intrusions of warm water would have intensified buoyancy forcing and, in turn, reinforced ACC transport in the South Indian Ocean during MIS 4 and 2[73]. Since CDW is supplied to the Weddell Gyre from the Indian-sector Southern Ocean, a strengthened ACC would likely enhance poleward heat transport, increasing the delivery of warm CDW into the gyre and thereby promoting warmer WDW.

While wind-induced and sea-ice-related processes likely prime the system for the development of the Glacial DML Polynya and deep convection in our study region, the topographic effect of seamounts has been proposed as one of the main potential triggers[20,74]. Bathymetric features such as Maud Rise are known to have the potential to destroy upper-ocean stratification through enhanced topographic-induced upwelling of salt and heat via WDW, possibly in combination with a spin-up in the Weddell Gyre's barotropic circulation[16]. In

addition to that, an eastward, wind-induced Ekman transport of saline waters onto the northern flank of Maud Rise may have contributed to an increased salt supply, which more than compensated for the freshening sea ice melt[75]. These processes are related to the Maud Rise Polynya, which is considered a precursor for the development of the historic Great Weddell Polynya (e.g., refs. 16,17). Our study site, the Bungenstock Plateau, forms another obstacle downstream of Maud Rise, where the water column today is characterized by a distinct updoming of the isopycnals by several hundred meters (Supplementary Fig. 9). Accordingly, this plateau might have similarly caused upwelling and contributed to a density anomaly that characterized the Great Weddell Polynya during the 1970s and perhaps that of the last ice age.

Micropalaeontological data from various Weddell Sea sediment cores and paleobiological data from DML provide additional support for the existence of the Glacial DML Polynya within the perennial sea-ice zone just offshore of the ice front[25–29]. The glacial deposits of many sediment cores in our study region (Fig. 1b) contain continuous and abundant calcitic shells of foraminifers[25,76,77]. Such glacial archives are the exception on the Antarctic continental margin and have been interpreted to reflect enhanced biological productivity in a polynya setting[26,27], whereby the glacial increase in deep water alkalinity in the Weddell Sea favored the preservation of calcitic shells[78]. The increased productivity during glacial periods extended at least to the northern edge on Bungenstock Plateau[28], as indicated by sediment records from sites PS1506 (68.7°S; 2,426 m) and PS1388 (69.0°S; 2,526 m; Fig. 1b), both north of PS128_2 (Fig. 1b). This implies that the Glacial DML Polynya was present during the last glacial period as well as in preceding glacials extending northward to at least 68.7°S. Additional evidence for a glacial open-ocean polynya is provided by fossil stomach oil ("mumiyo") deposits from breeding colonies of snow petrels in DML[79,80]. As the distance between the breeding sites in DML and the sea-ice edge during the LGM was >1500 km[81] and the maximum foraging range of snow petrels is only ~400 km, year-round accessible polynyas were most likely a prerequisite for their survival[64]. The Glacial DML Polynya would be only ~300 km away and could explain the occurrence of snow petrel colonies in DML during the LGM.

Taken together, these evidences point to the existence of the Glacial DML Polynya, which extended from the ice edge in the south northwards to at least 68.7° S at ~5–6° W. This northern position overlaps with the southern boundary of the historic Great Weddell Polynya. Observations from the mid-1970s place the southern edge of the Great Weddell Polynya near ~69°S at ~5°W (Fig. 1a), and waters around ~68°S remained polynya-prone even in 1977 (a year without an active polynya), indicating a sensitive preconditioning zone where ocean heat can thin or remove sea ice[44].

Relative to this modern baseline, a "glacial Great Weddell Polynya" likely stretched to at least 69.4°S (PS128_2), representing only a modest expansion of ~45 km southward beyond the modern southern edge near 69°S. This would place the Glacial DML Polynya within a plausible "glacial Great Weddell Polynya" environment. Such excursions fall well

within observed behavior, as the modern Great Weddell Polynya can shift by ~100 km over just three days[44].

## Glacial forcing and feedback mechanisms

The formation of the Glacial DML Polynya and the associated changes in upper-ocean stratification can be attributed to a combination of forcing and feedback mechanisms acting jointly, although on different time-scales (Fig. 4). Some of the involved atmosphere-ice-ocean interactions that led to the formation of the Great Weddell Polynya in the 1970s may also have played a central role in the prolonged openings of a polynya during the last glacial period, despite different environmental conditions. The development of the historic Great Weddell Polynya has been attributed to decadal perturbations in near-surface stratification caused by changes in precipitation-evaporation fluxes and related to changes in the Southern Ocean overturning and Southern Annular Mode (SAM)[20,82]. It has been argued that during a prolonged negative SAM ("-SAM"), the equatorward shift and weakening of the Southern Hemisphere westerly winds allowed the drier and colder winds blowing northward from Antarctica to strengthen and expand. Consequently, the colder and drier atmospheric conditions favored increased sea-ice cover in the Weddell Sea and a saltier ASW, thereby weakening the stability of the pycnocline. This pre-conditioning of the surface layer in combination with topographically induced upwelling at bathymetric obstacles (e.g. Maud Rise) is considered to be the main trigger for the Great Weddell Polynya, deep convection, and WDW injection into the surface layer[20]. In a positive SAM phase ("+SAM"), the changes are reversed: the westerlies strengthen and shift southward; the surface becomes fresher and less dense, preventing the formation of a polynya. In contrast, the most recent Maud Rise Polynya formation in 2016 and 2017 occurred in the context of "+SAM"[83]. The polynya was smaller in comparison to the Great Weddell Polynya in the 1970s and restricted to the Maud Rise region, but was also characterized by weakly-stratified upper ocean conditions[75,83]. It has been discussed that the polynya formation in 2016 and 2017 was driven by an additional salt input related to intensified eastward surface wind stress in combination with a spin-up in Weddell Gyre's barotropic circulation[16].

In the past, SAM-like changes may also have affected the centennial and millennial-scale climate variability in the Southern Ocean (e.g., refs. 84,85) (Fig. 4). Although very little is known about its behavior during glacial times, "-SAM" was thought to have been more persistent during the LGM[24] (Fig. 4). Compared to the present, including the Holocene, the glacial boundary conditions are likely to have significantly improved the probability of forming the Glacial DML Polynya in the open ocean, as summarised in the schematic illustration Fig. 4: (1) An equatorward shift of the westerlies[86] would ensure drier and colder atmospheric conditions and help to destabilise the upper-ocean stratification. This is in line with LGM simulations from the Paleoclimate Modelling Intercomparison Project (PMIP)[87], which consistently indicate glacially reduced precipitation in the Southern Ocean. However, the models show little consistency with regard to changes in westerly wind strength[86,87]. (2) A larger Antarctic ice sheet with grounded ice advancing towards the shelf break would allow for very intense, cold, dry, and persistent katabatic winds[88] supporting the formation of polynyas, sea-ice, and deep convection. During glacial periods, when grounded ice sheets advanced across the Antarctic continental shelves, coastal shelf polynyas similar to those observed today[17,18,81] were likely absent. Instead, the ice advance would shift the occurrence of glacial polynyas northward to regions above the deep ocean (Fig. 4a). Katabatic winds, an important forcing factor in the formation of modern coastal polynyas, may therefore also have played a crucial role in the development of glacial polynyas in the open ocean. Accordingly, the glacial polynya formation would characterize a hybrid coastal–open-ocean polynya mode by combining processes of latent (coastal) and sensible heat (open ocean) polynyas. Strong katabatic

winds push sea ice away from the ice-covered shelf, exposing open water. When new ice forms in this open water, it releases latent heat into the atmosphere, keeping the water near freezing. Enhanced oceanic heat flux to the surface would increase the sensible heat exchange between surface water and cold air, keeping the area open. (3) During the glacial, the strong katabatic winds would cross the ice-covered shelf and reach out to the open ocean, which, in combination with a strong "-SAM" would make the central Weddell Sea gyre more prone to open ocean polynyas[24]. (4) The increased presence of (seasonal) sea-ice expanding far north in the Southern Ocean retains subsurface heat in the WDW, which is then recirculated within the Weddell Gyre[70] (Fig. 3c, g), explaining the degree of subsurface warming. Additionally, an intensified ACC would have boosted poleward heat delivery to the Weddell Gyre, further warming WDW[73]. (5) The flow velocity at the eastern boundary of the Weddell Gyre accelerates with "-SAM", increasing the southwesterly flow of WDW[89,90] and thus the topographically induced upwelling at Maud Rise and Bungenstock Plateau. (6) Model simulations suggest that the glacial freshwater fluxes from basal melting and calving were reduced during the LGM compared to today[68]. This should have increased the salinity of the ASW. Although this argument is debated[91], the increased formation of sea-ice in polynyas should more than compensate for the freshwater effect from basal melt.

(7) Local feedback mechanisms may have supported the conservation of the Glacial DML Polynya. A large extent of the polynya would create a strong low-pressure system. Together with the high pressure over the EAIS, the steeper thermal gradient between the cold Antarctic air and the polynya region would intensify the katabatic winds, thereby maintaining deep oceanic convection[20] (Fig. 4).

Since background climate conditions control the Great Weddell Polynya, we argue that glacial boundary conditions amplified regional preconditioning, favoring a strong and extensive Glacial DML Polynya, probably as large as the modern Great Weddell Polynya. Concurrently, ice advances towards the shelf break and katabatic winds would have shifted glacial polynyas northward into a hybrid coastal–open-ocean mode. Under glacial conditions, a southward extension of the glacial Great Weddell Polynya into our study region would be plausible, since this setting would be physically well-founded.

Overall, we found strong evidence that glacial subST$_{Mg/Ca\ vs.\ \Delta 47}$ variability on millennial and orbital timescales, and the inferred Glacial DML Polynya formation, are linked to internal climate dynamics and external orbital forcing interacting on different timescales. On orbital timescales, the long-term variability of the subST$_{Mg/Ca\ vs.\ \Delta 47}$ coincides with changes in the obliquity cycle (Fig. 3c). The interval of sustained polynya formation from ~38–20 ka BP (subST$_{Mg/Ca\ vs.\ \Delta 47}$ maxima) corresponds to an obliquity minimum (Fig. 3c), i.e., a period of enhanced radiative gain in the tropics and radiative loss at high latitudes. Accordingly, low obliquity strengthens the meridional temperature gradient between the mid and high latitudes, which in turn causes stronger westerlies and an intensified poleward transport of heat and moisture[92]. The preceding period of high obliquity from ~62–39 ka BP is dominated by density-driven upper-ocean stratification, as indicated by the low-frequency reductions in temperature and salinity (Fig. 3c, d).

On millennial timescales, the variability of subST$_{Mg/Ca\ vs.\ \Delta 47}$ appears to be related to changes in the Atlantic Meridional Overturning Circulation (AMOC)[53–56] (Fig. 3b). In our record, millennial-scale warming in subST$_{Mg/Ca\ vs.\ \Delta 47}$ off DML parallels decreases in $^{231}$Pa/$^{230}$Th from Bermuda Rise in the North Atlantic, indicating AMOC strengthening (Fig. 3b, c). Cooling intervals coincide with increases in $^{231}$Pa/$^{230}$Th during Heinrich Stadials (HS), which are cold glacial stadial events in Greenland related to phases of AMOC weakening[53–56]. This relationship is particularly clear from ~75–38 ka BP, when sampling resolution is higher (~125 years between adjacent samples) (Fig. 2b). From ~38–20 ka BP, the correlation weakens, likely due to the two-fold

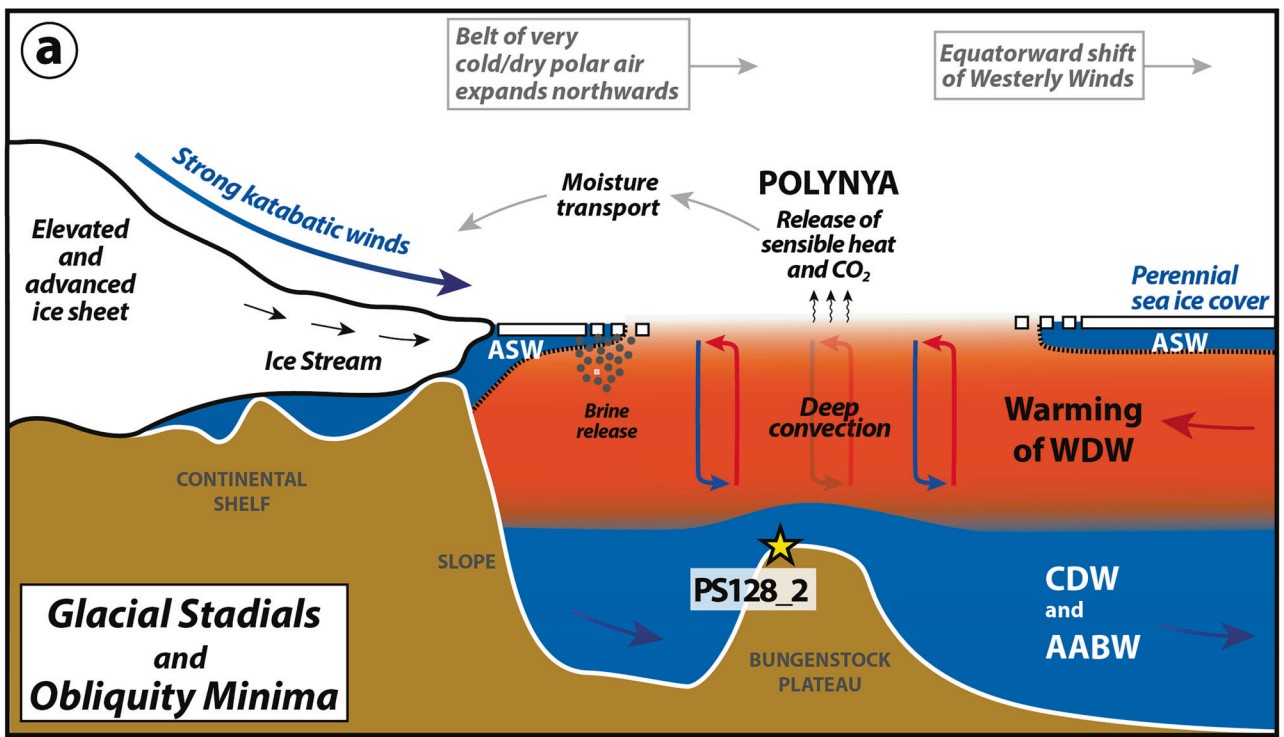

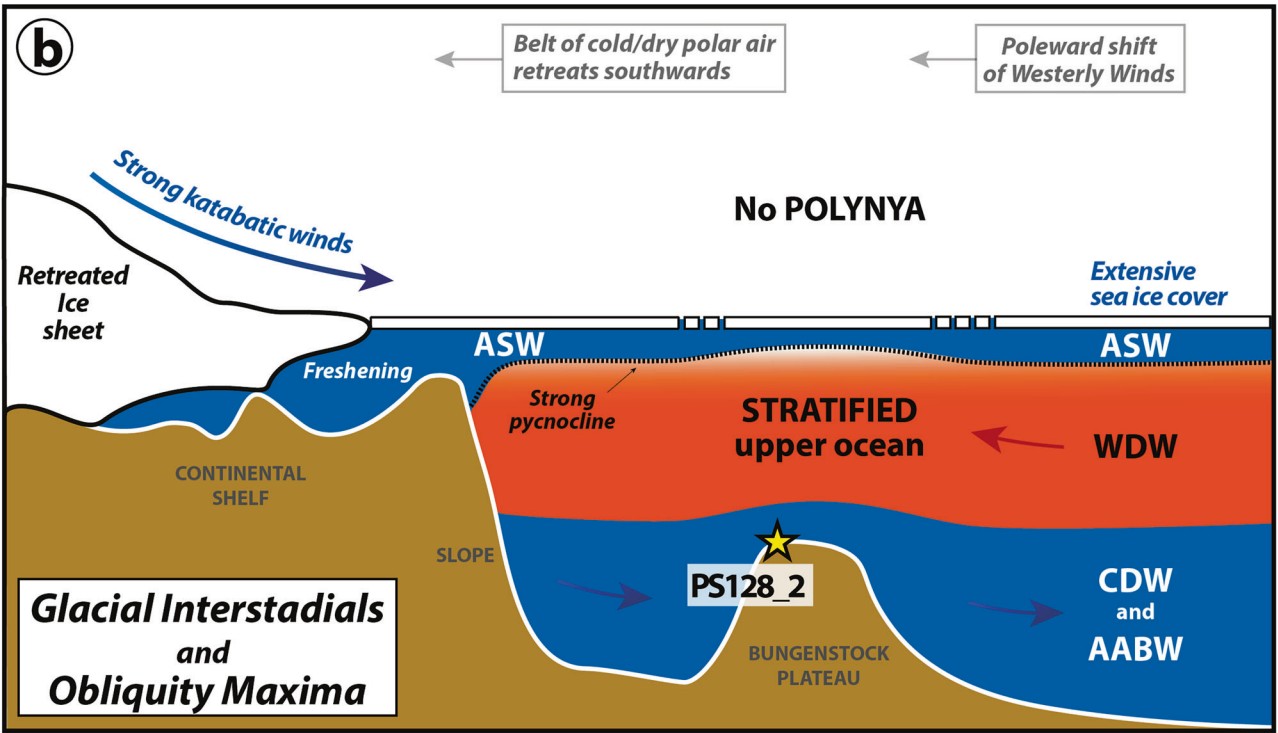

**Fig. 4 | Schematic Illustration of water masses and processes that affected the changes in upper-ocean stratification during glacial maxima** (a) **and minima** (b). Antarctic Surface Water (ASW), Warm Deep Water (WDW), Circumpolar Deep Water (CDW), Antarctic Bottom Water (AABW).

lower temporal resolution (~250 years) (Fig. 2b), which is reflected in the suppressed millennial-scale variability of the subST$_{Mg/Ca \text{ vs. } \Delta47}$ record. When comparing long-term variability, this period is characterized by a relative strong and more stable AMOC compared to the previous period (~75–38 ka BP) due to the absence of pronounced short-term minima, with the exception of the extreme HS2 minimum[55] (Fig. 3b). The amplitudes of the minima HS3 (at ~31 ka BP) and HS2 (~24

ka BP) are controversially discussed and have been suggested to be much lower[93] than inferred from the $^{231}Pa/^{230}Th$ proxy (Fig. 3b). If so, this would further support a predominantly strong AMOC with low-variability during ~38–20 ka BP, coinciding with a long-term maximum in subST$_{Mg/Ca \text{ vs. } \Delta47}$.

The positive correlation between millennial Southern Ocean subsurface warmings and AMOC strengthenings (Fig. 3b, c) suggests

an interhemispheric teleconnection and a forcing consistent with the contrasting temperature changes in Greenland and Antarctica and the bipolar seesaw paradigm[94]. Evidence from climate proxies and model results shows that glacial intensification in the AMOC during Northern Hemisphere interstadials is associated with a northward shift of the Intertropical Convergence Zone (ITCZ) and the Southern Hemisphere westerly winds, Antarctic cooling, and a northward extension of the sea-ice margin in the Southern Ocean[95–97]. Such conditions are similar to the "-SAM" as mentioned above and would help to precondition the polynya formation by accumulating heat in the subsurface and reducing salinity in the surface layer, so that the water column becomes buoyantly unstable (Fig. 4a). Strong katabatic winds could then tip the system into a convective state.

The Great Weddell polynya is recognized for producing substantial volumes of AABW[22], and may therefore have influenced the global overturning circulation. At our study site, however, the small and relatively stable benthic-planktonic $\delta^{18}O$ offset is masked by a relatively large uncertainty range, which unfortunately does not allow a reliable interpretation on the mixing depth and the formation of AABW (Fig. 4a). Future research should incorporate additional proxy reconstructions to better assess the role of AABW formation in the context of the Glacial DML polynya and the deep convective processes between Maud Rise and Bungenstock Plateau.

Still, it remains unclear how the obliquity-driven changes in the meridional temperature gradient have modified the millennial-driven shifts and intensities of the westerlies, particularly with respect to the preconditioning of the DML Polynya formation during the glacial Antarctic cold events (Fig. 4a). Model results indicate, on the one hand, an equatorward shift and weakening of the westerlies during the glacial Antarctic cold events (coinciding with Greenland interstadials), and, on the other hand, partly the opposite, an intensification of the westerly winds during glacial obliquity minima[92,95]. Since the position and strength of the westerly winds affect the poleward atmospheric transport of heat and moisture as well as the upwelling at the Antarctic divergence, they are an obvious candidate to influence our study area. Due to the apparent gap in our knowledge about the nature of changes in the westerly winds, we can only speculate that the opposing effects on wind strength may have balanced each other out, thereby resulting in only relatively minor changes in westerly wind intensity. The majority of models suggest that the millennial-scale latitudinal shifts in the westerly winds resulted in a northward expansion of the belt of very cold and dry polar air, accompanied by strong katabatic winds and sea-ice propagation during glacial Antarctic cold stadials (Greenland interstadials) and, conversely, during warm interstadials known as Antarctic Isotope Maxima (AIM) (Greenland stadials, HS).

As our results point to a polynya-driven heat release to the atmosphere during periods of millennial-scale and obliquity-related Antarctic cooling associated with enhanced northward heat transport via an intensified AMOC, they do not provide a mechanism that contributed to or amplified the Antarctic warming events (i.e., AIMs; Fig. 4a) during the last glacial period. During the glacial interstadials around Antarctica, the stratified upper water column may have hindered the release of heat to the surface ocean and atmosphere, which would have rather contributed to a reduction in DML air temperatures or a slowdown of the rate of surface warming along the coastal margin. Conversely, the polynya-driven release of heat during the glacial stadials in Antarctica (cold millennial-scale events) may have dampened the cooling but likely increased the moisture supply to Antarctica[98] and thus promoted the accumulation of ice and the thickening of an advancing ice sheet at the DML shelf margin.

Our Southern Ocean record of subsurface temperature variability adds another piece to the bipolar seesaw puzzle and to the discussion of millennial climate interactions within the Southern Ocean cryosphere-ocean-atmosphere system.

Our proxy data from site PS128-2 on the Bungenstock Plateau indicate that the recurring presence of the Glacial DML Polynya was a typical feature of the last ice age. Though the Glacial DML Polynya may share some similarities with the openings of the Great Weddell Polynya in the 1970s and the Maud Rise Polynya in 2016 and 2017, they are the result of different processes and impacts on different timescales. While the formation of the Glacial DML Polynya is consistent with a hybrid polynya mode that combines characteristics of a coastal polynya and an open ocean polynya, the offshore events of the last century are primarily open-ocean driven. We therefore propose that the modern occurrences are not a direct remnant of the last ice age, but rather an expression of a regionally persistent sensitivity to atmosphere-ice-ocean interactions, which was strongly expressed during the last glacial, when polynya-deep-convection operated on millennial timescales. We conclude that external obliquity-forced variations in polar insolation on orbital timescales in combination with internal millennial variability originating in the Southern Ocean, controlled the polynya formation and the associated atmosphere-ice-ocean interactions. In addition, internal variability, which appears to be linked to AMOC dynamics, might also have contributed to these changes. The driving force behind this is difficult to identify, but both the glacial AMOC and the DML Polynya are very sensitive to changes in surface salinity. Even if this process is linked to the thermal bi-polar ocean seesaw hypothesis[99], it remains unclear to what extent the interhemispheric coupling mechanism occurred via the ocean and the atmosphere.

Since the described glacial boundary conditions for polynya formation are not unique to the Atlantic sector of the Southern Ocean, we suggest that the glacial DML Weddell Polynya may have been only one of many polynyas in the polar Southern Ocean. Whether polynyas can be a source or a sink for atmospheric $CO_2$ is still disputed[100–102] and, like the forcing and feedback mechanisms, can only be disentangled with the use of climate models.

## Methods
### Core material and core processing
Three gravity cores were collected during two RV *Polarstern* expeditions: PS128_2-2 (69.4115°S, 5.5897°W) and PS128_2-3 (69.4107°S, 5.5890°W) (recovered during expedition PS128 in 2022[103]) and PS111_13-4 (70. 0936°S, 6. 8501°W) (collected during expedition PS111 in 2018[104]). Cores PS128_2-2 and PS128_2-3 were retrieved from almost identical coordinates (ca. 96 m apart from each other), while PS111_13-4 was recovered at a location 90 km shoreward relative to the PS128_2 sampling site (Fig. 1c). Magnetic susceptibility[103] and X-ray fluorescence (XRF) core scanning data were used for a detailed core-to-core correlation.

### Magnetic susceptibility
Magnetic susceptibility measurements were conducted onboard using a GEOTEK Multi-Sensor Core Logger (MSCL) on cores PS128 2-2 and PS128 2-3 to assess magnetic mineral content, an indicator of terrigenous sediment variations. The MSCL provided continuous cross-sectional measurements at 1 cm resolution. Raw data were drift and volume-corrected to obtain true susceptibility values (kappa) in $10^{-6}$ SI units.

### X-ray fluorescence (XRF) scanning
X-ray fluorescence (XRF) core scanning provided non-destructive chemical composition analysis of cores PS128_2-2, PS128_2-3, and PS111_13-4. Measurements were performed at the Alfred Wegener Institute (AWI), Bremerhaven, using an AVAATECH XRF core scanner, with 1 cm steps (area 10 × 12 mm) at three-run settings: 10 kV, 30 kV, and 50 kV.

## Stratigraphic correlation of sediment records and related measuring procedure

Marine sediment cores PS128_2-2 (69.4115°S, 5.5897°W) and PS128_2-3 (69.4107°S, 5.5890°W) (less than 100 m apart from each other)[103], were used to generate a composite record that doubles the amount of sediment material available for multiproxy analyses and merges proxy information from both cores. We correlated the magnetic susceptibility records and converted the depth scale of core PS128_2-2 to the sediment depth of core PS128_2-3 (Supplementary Fig. 1). Planktonic and benthic stable isotopes ($\delta^{18}$O, $\delta^{13}$C, $\Delta_{47}$), planktonic Mg/Ca ratios, as well as planktonic $^{14}$C were completely measured on PS128_2-3 for the time interval >18 ka. For the interval from 18 to 16 ka, all planktonic foraminifers were used up by $^{14}$C datings, $\delta^{18}$O and $\delta^{13}$C measurements. For the past 16 ka, the number of well-preserved foraminifers was too low to perform $^{14}$C, Mg/Ca and $\Delta_{47}$ measurements, despite a possible merging of identical sample sections from the two cores. The low numbers of foraminifers only allowed for $\delta^{18}$O and $\delta^{13}$C measurements.

The XRF scanning at AWI could only be performed on core PS128_2-2, as core PS128_2-3 was completely sampled on board. The core-to-core correlation was used to transfer the XRF scanning data from PS128_2-2 to the depth scale of PS128_2-3. In the following, we considered the merged data sets from PS128_2-2 and PS128_2-3 as information from site PS128_2.

In order to improve the stratigraphy at our site location, we correlated the XRF titanium record (ln Ti) of site PS128_2 to that from PS111_13-4 (Supplementary Fig.1). This allows the transfer of Holocene $^{14}$C ages from PS111_13-4 to site PS128_2. The interval 158-266 cm from core PS111_13-4 corresponds to the depth interval of 27-45 at site PS128_2. Accordingly, the sedimentation rate is six times higher at PS111_13-4, as expected in core sites closer to the Antarctic margin.

## $^{14}$C radiocarbon measurements

For $^{14}$C radiocarbon dating, we picked ~100 well-preserved planktonic individuals (*N. pachyderma* sin), weighing >1 mg. All $^{14}$C measurements were carried out on a Mini Carbon Dating System (MICADAS) at AWI, Bremerhaven (see Mollenhauer et al.[105] for more details). The $^{14}$C radiocarbon datings are presented in Supplementary Table 1.

## Age model

The PS128_2 age model integrates 37 accelerator mass spectrometry (AMS) $^{14}$C radiocarbon datings (Supplementary Tab.1) and $\delta^{18}$O stratigraphy. Most of $^{14}$C radiocarbon ages are concentrated in the last glacial period, which is exceptionally rare in Antarctica. To refine the chronology of the uppermost 44.5 cm, we leveraged the XRF-based correlation between cores PS128_2 and PS111_13-4 (Supplementary Fig.1) and incorporated 10 AMS $^{14}$C dates originally measured in core PS111_13-4 (Supplementary Table 1). The AMS $^{14}$C ages were calibrated to calendar ages using the most recent Marine20 calibration curve[106,107]. To account for the Holocene regional surface ocean $^{14}$C depletion, we applied a local reservoir age of $\Delta$R of 665 years, as provided by the marine radiocarbon reservoir database (Reimer and Reimer[108], http://calib.org/marine/). As this value does not consider the likely additional glacial $^{14}$C depletion in the polar glacial ocean, we follow the approach of Heaton et al.[107] For the glacial period (>11.5 ka), we used the $\Delta$R of 1,200 years (665 + 525 years). The additional 525 years reflects the regional mean $\Delta$R between latitudes of 68.75°S and 71.25°S under a simulated glacial climate[107]. The $\sigma$-1 uncertainty range of the calibrated $^{14}$C ages defines the chronological constraint (Supplementary Table 1), which is needed for further developing our age model. Beyond the range of $^{14}$C age dating, we established a small number of five chronological control points using $\delta^{18}$O tie-points (Supplementary Fig. 2). These were determined by aligning our planktonic $\delta^{18}$O record to the well-established benthic Deep South Atlantic $\delta^{18}$O reference stack (LS16)[47] (Supplementary Fig. 2a, b). For

the age model development, we used the age-depth modelling approach based on Bayesian statistics (BACON)[109] (Supplementary Fig. 2c). Age models were generated using BACON v. 2.2 (ref. 109) within PaleoDataView v. 0.8.3.4 (ref. 110), with modified parameters of mem.mean (0.4) and mean.strength (4). The models were based on 30,000 age-depth realizations using a t-distribution (t.a = 9, t.b = 10).

The evolution of our planktonic and benthic $\delta^{18}$O records shows a strong correlation (Supplementary Fig. 2b). The benthic $\delta^{18}$O values are consistently higher than the planktonic $\delta^{18}$O values, and both records align well with the LS16 $\delta^{18}$O stack[47] (Supplementary Fig. 2a, b). The remarkable alignment between both planktonic and benthic $\delta^{18}$O records and the LS16 $\delta^{18}$O stack[47] during Marine Isotope Stage (MIS) 3, 2, 1, and the Glacial Termination I (i.e., spanning the entire $^{14}$C dating range) provides independent compelling evidence for the robustness and accuracy of our age model (Supplementary Fig. 2a, b). We emphasize that the benthic $\delta^{18}$O record is used exclusively for stratigraphic purposes.

## *Neogloboquadrina pachyderma* sin. Mg/Ca analysis

We carried out Mg/Ca analysis following the cleaning procedures of refs. 111,112, including both oxidative and reductive steps. Approximately 100 specimens of *N. pachyderma* sin. were handpicked at nearly every 1 cm resolution under a binocular microscope from the sediment fraction 250 – 500 μm. The Mg/Ca analyses were performed on an ICP–OES (VARIAN 720-ES, GEOMAR). The ECRM752-1 standard was utilized to monitor the conditions of the instrument (ref. 113). The long-term analytical precision is ±0.1 mmol/mol for Mg/Ca. In order to exclude clay contamination and post-depositional Mn-rich carbonate coatings on Mg/Ca ratios, we measured Fe/Ca, Al/Ca, and Mn/Ca ratios in conjunction with Mg/Ca. The generally accepted threshold value for Fe/Ca, Al/Ca, and Mn/Ca is 0.1 mmol/mol[114]. Al/Ca values are higher than 0.1 mmol/mol, there is no correlation to Mg/Ca ratio (Supplementary Fig. 3). Most of the Fe/Ca and Mn/Ca values are below 0.1 mmol/mol, with a few exceptions (Supplementary Fig. 3). Hence, silicate contamination is negligible.

## Stable oxygen and carbon isotopic composition of planktonic and benthic foraminiferal species

Stable isotopes ($\delta^{18}$O and $\delta^{13}$C) of *N. pachyderma* sin. were measured in combination with the Thermo Scientific MAT 253 mass spectrometer and an automated Kiel IV Carbonate Preparation Device at AWI, Bremerhaven. In addition, we also measured benthic foraminifera species *Cibicides wuellerstorfi* $\delta^{18}$O at AWI. For the planktonic foraminifera, six to eight specimens were handpicked every 1 cm under a binocular microscope from the sediment fraction 125–250 μm, while one to two specimens of benthic foraminifera were selected from the sediment fraction 250–500 μm, whenever possible. All $\delta^{18}$O and $\delta^{13}$C results are reported in ‰ relative to VPDB (Vienna Pee Dee belemnite) by calibrating with IAEA603 or NBS-19 and by using an in-house calibration standard (Solnhofen limestone, SHK). The external reproducibility of $\delta^{18}$O measurements is based on our SHK standard measured over a year-round period, together with samples. The long-term analytical precision for $\delta^{18}$O is <0.06‰ and for $\delta^{13}$C < 0.04‰.

## Oxygen isotopic composition of seawater ($\delta^{18}$O$_{sw-ivc}$)

The regional ice-volume-corrected $\delta^{18}$O$_{sw}$ record ($\delta^{18}$O$_{sw-ivc}$) was calculated by subtracting changes in global $\delta^{18}$O$_{sw}$ produced by continental-ice variability, applying the relative sea-level curve of Grant et al.[39]. The remaining temperature effect was removed from $\delta^{18}$O$_{sw-ivc}$ values using the Shackleton's[115] temperature versus $\delta^{18}$O$_{calcite}$ equation:

$$\delta^{18}O_{calcite}(\text{‰ VPDB}) = (21.9 - 3.16*(31.061 + T(^{\circ}C))*0.5) + \delta^{18}O_{sw}(\text{‰ VPDB})$$

We converted the calculated $\delta^{18}O_{sw\text{-}ivc}$ from VPDB to Vienna Standard Mean Ocean Water (VSMOW) unit following Hut[116]:

$$\delta^{18}O(‰\ VPDB) = 0.9998 * \delta^{18}O_{sw}(VSMOW) - 0.27‰$$

$\delta^{18}O_{sw\text{-}ivc}$ values are positively correlated to salinity conditions, i.e., high $\delta^{18}O_{sw\text{-}ivc}$ values indicate more saline conditions, while low values indicate fresher conditions.

We conducted an error propagation analysis to evaluate the uncertainties in $\delta^{18}O_{sw\text{-}ivc}$ calculations. This analysis accounted for measurement uncertainties in Mg/Ca and $\delta^{18}O$, calibration errors in the Mg/Ca-temperature relationship, and uncertainties in the salinity-$\delta^{18}O_{sw}$ relationship. We obtained a propagated $2\sigma$-error in $\delta^{18}O_{sw\text{-}ivc}$ of ±0.34‰ for *N. pachyderma* sin., which presents a similar value to a previous study in the subpolar Pacific (Riethdorf et al.[117]). We did not convert the $\delta^{18}O_{sw\text{-}ivc}$ into salinity units because the modern linear relationship between $\delta^{18}O_{sw}$ and salinity may have changed in the past due to variations in the sea-ice regime, ocean circulation, and freshwater budget (ref. [118]).

## Apparent calcification depth

We estimated the apparent calcification depth of *N. pachyderma* sin. using the relationship between seawater $\delta^{18}O$ composition and temperature, as defined by the Shackleton's[115] equation (Supplementary Fig. 3). Seawater $\delta^{18}O$ ($\delta^{18}O_{sw}$) were extracted from Mackensen's[26] dataset for stations PS16/200 (68.5°S; 7.7°W) and PS16/197 (69.48°S; 8.22°W), while annual mean temperatures were taken from nearby WOA 23 stations[43] (Supplementary Fig. 3). The predicted $\delta^{18}O$ of the foraminifer at 115 m and 120 m water depth for 68.5°S and 69.48°S, respectively (marked by white stars in Fig. S3) would match the measured *N. pachyderma* sin. $\delta^{18}O$ values of the surface samples (3.8 ‰ for both locations). The surface *N. pachyderma* sin. $\delta^{18}O$ values are measured from our surface MUC cores PS128_19-1 (68.53°S; 7°W) and PS128_2-4 (69.41°S; 5.6°W). The depth variation corresponds to the modern pycnocline structure, where offshore stations exhibit shallower pycnoclines and correspondingly shallower habitat depths, while onshore stations show deeper pycnoclines with relatively deeper habitat depths.

## *Neogloboquadrina pachyderma* sin. $\delta^{13}C$ offset correction

At site PS128_2-4 (-69.41°S; 5.6°W), the surface *N. pachyderma* sin. $\delta^{13}C$ value (0.9 ‰) is offset by +0.3‰ relative to the ambient seawater dissolved inorganic carbon (0.5 ‰) (Mackensen et al.[26]), assuming an apparent calcification depth of *N. pachyderma* sin. at 115 m water depth (see above). Accordingly, we corrected the downcore *N. pachyderma* sin. $\delta^{13}C$ record by applying an offset of -0.3‰ to account for ambient seawater disequilibrium.

## Clumped isotope-based temperature ($\Delta_{47}$)

In this study, we present a pioneer clumped isotope ($\Delta_{47}$) record covering the last glacial period close to Antarctica. We followed the sample preparation and cleaning protocol as proposed in Meinicke et al.[119]. A large number of approximately 1000 planktonic foraminiferal tests (*N. pachyderma* sin.) were picked from 10 distinct depth intervals from core PS128_2, which are evenly distributed throughout the downcore record. The chambers were carefully opened and cleaned once with methanol and then a few times with DI water in an ultrasonic bath. The analysis was performed using two Thermo Scientific MAT 253 Plus mass spectrometers coupled to Kiel IV carbonate preparation devices (Thermo Fisher Scientific, Bremen, Germany) at the University of Bergen (Norway). Organic contaminants in the sample gas were removed using a custom-built Porapak trap installed in the Kiel device held at $-50\,°C$ during measurements. The $\Delta_{47}$ data were transferred into the I-CDES scale (Bernasconi et al.[120]) using the community standards ETH 1-3, measured alongside the samples with a standard-to-sample ratio of approximately 1:1. The

reference scale transfer was conducted in a moving window approach using the adjacent 30 standard measurements on either side. Measurements were conducted over a 6-month time period from April to October 2024, with temporal spacing of replicate measurements assuring independent correction of individual replicate measurements. External reproducibility of the samples (standard deviation across all replicate measurements after corrections) ranged from 0.022 ‰ to 0.033 ‰, which is similar to the observed reproducibility of the monitoring standards IAEA C-2 (AVG: 0.640 ‰; SD: 0.026 ‰; $N = 198$) and Merck (AVG: 0.516 ‰; SD: 0.029‰; $N = 187$) during the time period where the analyses were performed. This similar reproducibility between samples and monitoring standards as well as average values of the monitoring standards closely matching those determined by the community (Bernasconi et al.[120]) attest to the robustness of the $\Delta_{47}$ measurements.

The averages of the replicate $\Delta_{47}$ measurements per sample interval were converted to temperature using the planktonic foraminifera-based calibration proposed by Meinicke et al.[119], updated to the I-CDES scale by Meinicke et al[119].: ($\Delta_{47} = (0.0397 \pm 0.0011) * 106/T^2 + (0.1518 \pm 0.0128)$). Confidence intervals include the uncertainty of the calibration and were calculated with a Monte Carlo approach ($N = 5000$) based on paired uncertainties of intercept and slope of the calibration. We note that using alternative calibrations suggested recently (Daeron and Gray[121]; Daeron and Vermeesch[122]) would decrease temperatures by roughly 3 °C and 1 °C, respectively. The calibration of Daeron and Gray[121] would, for most sample,s result in unrealistic temperatures well below the freezing point of seawater.

Importantly, our Mg/Ca ratios paired to $\Delta_{47}$-based temperature show good agreement, albeit with the comparatively high analytical uncertainty of $\Delta_{47}$ data. This supports the reliability of our temperature reconstructions.

## Mg/Ca-based temperature reconstruction

Our glacial foraminiferal Mg/Ca record is the southernmost record ever published, and the applicability of this proxy in an extreme polar setting has not been tested so far. After applying regional species-specific Mg/Ca vs temperature equations available for *N. pachyderma* sin., we obtained unrealistic temperature ranges being much too high for the extreme polar conditions, prompting the need for an alternative approach, such as refs. [32–38]. (Supplementary Fig. 5 and Table 2). We here attempt to prove that the sensitivity of the Mg/Ca proxy is high enough to reliably reflect extremely low ocean temperature changes. Besides proxy-related considerations, we compare the Mg/Ca results to other ocean temperature proxies.

The dissolution of calcite tends to selectively remove Mg-rich calcite, potentially impacting the reconstruction of Mg/Ca-based temperature, therefore, producing "colder" temperature artifact[123]. We weighed a total of ~80 *N. pachyderma* sin. specimens (250–500 μm) and observed minimal calcite mass reduction during the entire studied interval, providing indication that our planktonic foraminifera Mg/Ca is not controlled by dissolution (Supplementary Fig. 6). Scanning electron microscopy (SEM) images further confirm the excellent preservation state of *N. pachyderma* sin. (Supplementary Fig. 7). Both the weights and SEM images originate from the same sample interval and were independently analyzed in paired assessments with different temperature proxy measurements (see below).

We calibrated our downcore Mg/Ca against the currently most independent temperature proxy applied for carbonate microfossil, specifically the $\Delta_{47}$ (Supplementary Fig. 8a). This approach aligns with that of Kozdon et al.[35] in the Norwegian Sea, where Mg/Ca was independently related to another temperature proxy, $\delta^{44/40}Ca$. For our calibration (subST$_{Mg/Ca\ vs.\ \Delta47}$), we focused on 10 pairs of Mg/Ca and $\Delta_{47}$-temperature data points (Supplementary Fig. 8a). The subST$_{Mg/Ca\ vs.\ \Delta47}$ calibration dataset demonstrates high internal consistency, with 80% of sample pairs exhibiting strong correlation (r = 0.85; r² = 0.72).

Two anomalous pairs were identified where both Mg/Ca and $\Delta_{47}$-derived temperature values diverged significantly. For these outlier pairs, triplicate Mg/Ca analyses were conducted, confirming anomalously high Mg/Ca values relative to the corresponding $\Delta_{47}$-derived temperatures, which yielded physically implausible values below the freezing point of seawater. To maintain methodological rigor while acknowledging the physical limitations of the marine system, we implemented a minimum temperature threshold at -2 °C applied to $\Delta_{47}$-derived temperature estimates where mean values fell below this physically constrained limit (Supplementary Fig. 8). While the measured values are below the freezing point, the upper limit of the 68% CI reaches above 2 °C, except for the sample at 179.5 cm (−3.6±1.2 °C). The subST$_{Mg/Ca\ vs.\ \Delta47}$ relationship reveals a moderately strong positive linear relationship (r = 0.75; r$^2$ = 0.57). The subST$_{Mg/Ca\ vs.\ \Delta47}$ calibration uncertainties have been integrated based on a Monte Carlo approach (10,000 iterations) (Supplementary Figs. 8c and 2d). The subST$_{Mg/Ca\ vs.\ \Delta47}$ calibration equation is given by:

$$Mg/Ca = (0.879 \pm 0.041) + (0.128 \pm 0.033) \times T(\Delta_{47})$$

Interestingly, our proposed subST$_{Mg/Ca\ vs.\ \Delta47}$ relationship reveals a slope strikingly similar to Kozdon et al[35]., albeit with a constant offset (Supplementary Fig. 8b). While both calibrations yield comparable temperature amplitudes, validating our established subST$_{Mg/Ca\ vs.\ \Delta47}$ relationship, Kozdon et al.[35]'s equation produces unrealistically warm absolute temperatures, ~4 °C warmer than our subST$_{Mg/Ca\ vs.\ \Delta47}$ (Supplementary Fig. 8c). The consistency in temperature amplitudes validates the thermal sensitivity of our Mg/Ca record despite the presence of outliers in our calibration.

Furthermore, we established an independent approach based on a cold end-member and the modern surface Mg/Ca sample (cold-warm end-member) to test our subST$_{Mg/Ca\ vs.\ \Delta47}$ relationship. For this additional calibration, we assume a limiting temperature of −2 °C, representing the seawater freezing point to the lowest Mg/Ca ratios from a cluster of five downcore samples during the last glacial period. The modern surface Mg/Ca reference sample was measured on *N. pachyderma* sin. from station PS128_19-1 (68.53°S; 7°W) in the vicinity of core PS128_2. We assumed the annual temperature at 120 m water depth (World Ocean Atlas 23[124]) as the likely habitat depth of *N. pachyderma* sin. (see apparent calcification depth estimates, Supplementary Fig. 4). This result is similar to the calculated isotopic calcification depth of 115 m at a nearby site location PS16/200 (68.53°S; 7.7°W)[26] (Supplementary Fig. 4). This independent calibration yielded a slope comparable to our subST$_{Mg/Ca\ vs.\ \Delta47}$ relationship, though the cold-warm end-member approach produced systematically warmer temperature estimates. The cold-warm end-member calibration equation is given by:

$$Mg/Ca = (0.851 \pm 0.025) + (0.142 \pm 0.014) \times T$$

The two calibrations demonstrate maximum congruence in the warm temperature range, while exhibiting a minor difference with a maximum divergence of ~0.5 °C in the coldest temperature domain. Both calibrations produced virtually the same temperature amplitude in our downcore time-series record (Supplementary Fig. 8c).

We demonstrate that our Mg/Ca calibrations perform robustly in extreme polar settings. By applying our established calibrations (i.e., subST$_{Mg/Ca\ vs.\ \Delta47}$ and cold-warm end-member) to the surface Mg/Ca dataset located south of 64°S (Vázquez Riveiros et al.[33]), we find realistic Antarctic Surface Water (ASW) temperatures of −1.76 °C ± 0.506 °C (The subST$_{Mg/Ca\ vs.\ \Delta47}$) and −1.39 °C ± 0.456 °C (cold-warm end-member). Such temperatures reflect the modern ASW signature (i.e., −2 °C to −0.6 °C), which defines the likely habitat depth range where *N. pachyderma* sin. calcifies. Therefore, we suggest that our Mg/Ca calibrations are sensitive to cold temperatures (below 4 °C)

and can resolve small temperature variations in extreme polar conditions, addressing previous uncertainties about proxy performance in cold, polar waters. Testing these calibrations against new polar datasets and different oceanographic settings would help establish their broader applicability in paleoceanographic reconstructions.

## Data availability
All relevant data in this paper are available at PANGAEA Data Publisher and EarthChem Library (https://doi.pangaea.de/10.1594/PANGAEA.992012, https://doi.pangaea.de/10.1594/PANGAEA.992006 and https://doi.pangaea.de/10.1594/PANGAEA.992005) and (https://doi.org/10.60520/IEDA/114360).

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

## Acknowledgements
Tainã M. L. Pinho acknowledges the support from the AWI Helmholtz-Zentrum für Polar- und Meeresforschung through the institutional research program "Changing Earth—Sustaining Our Future" through the PhD program INSPIRES project and Open Access publication fund of Alfred-Wegener-Institut Helmholtz-Zentrum für Polar- und Meeresforschung (Projekt DEAL). Victoria Taylor, Nanna Dyrbye, Heidi Bernhoft von Obstfelder, and Enver Alagoz are thanked for assistance with clumped isotope measurements.

## Author contributions
T.M.L.P and R.T designed the study. G.M provided radiocarbon data. T.M.L.P prepared all samples for a variety of measurements and generated the isotope data. D.N facilitated Mg/Ca measurements and additional stable isotope measurements. A. N.M provided the clumped isotope measurements. L.L.J performed XRF core scanning and data processing. T.M.L.P., D.N., D.N, G.M, J.M, L.L.J. and R.T contributed to sample collection. T.M.L.P, D.N, A.N.M, G.M, J.M, G.L, L.L.J, S.H, V.R, F.L, R.T contributed to the final manuscript.

## Funding

## Competing interests
The authors declare no competing interests.
