## [Peer Review file · Nature Communications]

Millennial-to-orbital-scale subsurface ocean warming and Polynya formation off Dronning Maud Land during the last glacial

Corresponding Author: Dr Tainã Pinho

Version 0:

Reviewer comments:

Reviewer #1

(Remarks to the Author)

Review of "Millennial-scale subsurface ocean warming in the eastern Weddell Sea during the last glacial period" by Pinho et al.

This study investigates the occurrence of open-ocean polynyas in the Weddell Sea during the glacial period from 75 000 - 20 000 years before present. By analysing proxy data from the Weddell Sea the changes in the subsurface ocean are derived. The results indicate occurrence of polynyas during this time and hypothetical formation processes are outlined.

The manuscript is interesting to read, but I would recommend some revisions. In a revision the authors' should provide more guidance of paleo terminology for the broader field of ocean and atmospheric science, clarify some messages and do structural changes, improve the visual representation of their graphics, and a careful review of the cited literature and referencing of figures. Further, some structural changes for better readability would improve the accessibility of the results. Yet, the manuscript provides novel insights in the past of polynyas.

Major Comments

Paleo terminology:

It would be useful for readers, who are not familiar with proxies, to explain more clearly which proxy represents which ocean variable and which combinations are used (lines 91-105).

A more detailed explanation on subSTMg/Ca vs.47 is needed, which seems the key result of the analysis. It is not clear how this data was derived or where these results are presented, as Figure 2d shows a line for subSTMg/Ca and diamonds for subST47, but not the always referred subSTMg/Ca vs.47.

Further it would help the reader if some information about the temporal resolution of the results derived from the used ice core was provided.

Moreover, abbreviations like MIS, AIM,CI, SMOW, VPDB should be explained in the manuscript and if processes like HS, AIM are important in relation to the polynya, they should briefly be summarized for readers unfamiliar with them.

Clarification of messages

The section "Glacial occurrence of the Great Weddell Polynya" interprets and compares the results to modern open-ocean polynyas. Yet, in the section "Glacial Forcing and feedback mechanisms" especially point (2) indicates that processes related to modern coastal polynya formation, like katabatic winds, are important. I would recommend the authors to clarify if point (2) is related to the previously shown results or if in the case of the here studied polynya a combination of processes is important and if the results here indicate a polynya comparable to a modern open-cean polynya or coastal polynya.

I would recommend to restructure the first 2 paragraphs in the Section "Glacial occurrence of the Great Weddell Polynya" to tell the reader, where the species live and how it moves in the vertical and then interpret this with shoaling/ deepening of the water masses (lines 193-228)

In lines 271-275 , the authors claim that the Bungenstock Plateau works similarly as Maud Rise, and explain that isopycnals are doming over the plateau, yet Figure 1 c,d) which show transects across the Plateau, do not show an indication of isopycnal doming. I would recommend the authors to show more proof of the statement that Bungenstock Plateau works

similarly as Maud Rise.

The link of the investigated polynyas to the AMOC needs to be more incorporated (lines 358 - 383), the current manuscript does not provide an easily accessible link of those two processes. Large open-ocean polynyas, like the Great Weddell Sea polynya, are known to form large amounts of AABW, which could play a role in the global overturning, it would be good if the authors could expand the discussion towards this.

The important role of obliquity forcing is pointed out towards the end of the manuscript (lines 248,420-423) yet if it is an important finding regarding the formation of polynyas, I would recommend to highlight this earlier in the manuscript.

Citation of references

In several places in the manuscript the citations do not support the statement, e.g., the citation 56 (Gordon, 1991) is not an online resource and the cited results are not quantifiable. In the context of line 270, where this citation is used, a potential citation could be Cheon and Gordon (2019); citation 43 (Gordon et al. 2007, line 271), the study does not investigate the role of a barotropic gyre spin-up as initial trigger for the Great Weddell Polynya; citation 73 (Cheon and Gordon (2019), line 345) in this work the role of katabatic winds is not investigated. In some places, the reference is located in the wrong place in the text, as for example, line 88.

Visual representation of graphics:

In general the Figures 1 and 2 would need improvement for better readability and access for readers.

Figure 1: The use of discrete color levels will help the reader to interpret the data shown more easily. A diverging colormap, like the red/blue one, should have the white area at a value of 0 and often different colormaps are used for different ocean variables.

Further, please add a mark for Maud Rise in a) as its position is not commonly known and the text refers several times to it. Please add the transects shown in c) and d) to a) or b) so readers gain some orientation. Also please provide details for data shown in a,c,d (e.g., are those observations, models, observational climatologies).

Figure 2: There is a lot of information in this figure which makes the figure very hard to read. It is not obvious, which of the information are retrieved from the here presented ice core PS128_2 and which are from other studies. Even though the presentation is common for Paleo studies, it is not very common in other fields, e.g. having the youngest date on the right and the oldest on the left.

I would recommend the authors, to improve the readability of the figure in several places, e.g., in 2c) inverting the y-axis and adding a horizontal line for the mentioned threshold, would make it easier to interpret the results for the reader; some of the lines are unconnected to the y-axis it would help to have horizontal lines for guidance.

Further, the coloring of some periods is confusing, especially as the blue red gradient in some parts is similar to the colorbars used for ocean and air temperature, is the coloring needed? Are the highlighted periods exactly the periods of polynya occurrence? The text refers to 3 periods of polynya occurrence (38-20ka, 59-56ka and MIS4) yet there are 5 periods highlighted.

It is not clear what is shown in figure 2h. Is it sea ice extent or a concentration or a sea ice extent of the area? The units do not fit the caption and text (line 252)

Throughout the text of Section "Glacial upper-ocean stratification" the referencing of Figure 2 is not aligning with the Figure itself.

Minor Comments:

lines 42-43: does the description refer to the sections shown in Figure 1c,d?

lines 47-50: indicate that temperature and salinity are both contributing equally in density and stratification, yet salinity is the main contributor to density in polar regions.

line 63: the Great Weddell Polynya in the 1970s was not only triggered by density instabilities, it's one of many important processes, please rephrase this sentence.

line 69: it is not known for sure, that there is no ice formation in the Great Weddell Polynya, as it has shown for preceding smaller scale polynyas that they can form large amounts of sea ice (see Zhou et al., 2022), please rephrase this sentence

lines 85: define ka

lines 90: it would be worth expanding this a bit more, as in the manuscript feedbacks are also explored.

line 130: EDML is undefined.

line 133: age model, what does it mean, this word is used for the first time here

line 147-150: purpose of this statement it feels right now unconnected.

line 179-181: The red and blue lines in Figure 2 represent similar patterns, therefore the statement and the link to the d18O proxies is not clear.

line 186: "glacial interval" - which period is meant?

line 206: 'preventing further sea-ice from forming' - rephrase

line 212: Are you referring to your results or Greco et al. ?

line 254: is kyr =ka?

line 255: "the described sea-ice-driven process" are the authors referring here to Haumann et al.?

line 266: "main trigger" -the topographic effect is a potential trigger, not the main, as the polynya formation in the 1970s is a combination of processes - please rephrase

line 282: what does "glacial increase" in this context mean?

line 282: doubling of the word " water".

lines 276-290, does this statement refer to only the LGM or all polynya periods?

lines 312-335: some of these statements are not fully aligned with results and statements in previous sections, e.g., the formation of sea ice - previous statements throughout the text said there is no ice formation in polynyas.

line 359: point to the figure showing AMOC

line 362: what is the temporal resolution

line 364: please remind the reader of the periods which are compared in the following.

line 404: what is happening during an Antarctic warming event, which role does it play in context of the study

Reviewer #2

(Remarks to the Author)

This paper presents a multiproxy analysis of seafloor sediment cores collected off Dronning Maud Land (DML) in the Weddell Sea to elucidate changes in water column structure and stratification in the Antarctic Ocean between 75,000 and 20,000 years ago. The results suggest that variations in the thermal structure between Antarctic Surface Water (ASW) and Warm Deep Water (WDW) may have driven upper-ocean stratification or the formation of polynyas. This is a significant finding that may offer valuable insights into the origin of the present-day Great Weddell Polynya and enhance our understanding of modern climate phenomena involving ice sheet–ocean–atmosphere interactions.

While the study presents novel and compelling results, several aspects require revision or further clarification. Most notably, the absence of Holocene data for some proxies is concerning. Although the paper focuses on the glacial period, the historical Great Weddell Polynya is a Holocene phenomenon. Therefore, it is essential to compare the glacial record with Holocene-scale oceanic polynya events. Additionally, the omission of available Holocene data may raise questions about the selective presentation of results and whether inconvenient data have been intentionally withheld.

L 82-84: "we reconstruct past orbital- and millennial-scale variations in upper-ocean stratification from sediment core PS128_2, recovered from the Bungenstock Plateau off DML in the eastern Weddell Sea (Fig. 1)."

Comment: It should be clarified that this study utilizes composite records from three cores, not from PS128_2 alone.

Furthermore, the Supplementary Information does not clearly indicate which core each proxy dataset was derived from. More fundamentally, the rationale for including core PS111_13-4—located approximately 90 km away—requires explanation.

L 127: "Fig. 1" >> "Fig 2a"

L 139-140: "our reconstructed glacial subSTMg/Ca vs. $\Delta 47$ variability between ~70 and 20 ka BP (Fig. 2c)"

Comment: An explanation is required for the absence of Mg/Ca ratio data during the Holocene.

L 141, L 141, L 144, L 150: "(Fig. 2c)" >> "(Fig. 2d)"

L 167: Caption of Figure 2h "diatom faunal"

Comment: "faunal" is used for animals. Diatoms are plants, so "floral" is correct.

L 178: "(Fig. 2c, e)" >> "(Fig. 2d, e, f)"

L 182-183: "The $\delta^{13}\text{C}$ record of *N. pachyderma* sin. provides additional information on the signature of subsurface waters."

Comment: An explanation is needed for the absence of planktonic $\delta^{13}\text{C}$ data during the Holocene. Since Fig. 2a presents planktonic $\delta^{18}\text{O}$ data, it is presumed that $\delta^{13}\text{C}$ measurements were obtained simultaneously and should therefore be available.

Comment: The authors state that the $\delta^{18}\text{O}$ offset between planktonic and benthic foraminifera is stable; however, a detailed examination reveals several intervals where the two are in very close agreement. If so, these periods should be explained as not representing deep convection that would imply the presence of Antarctic Bottom Water (AABW). Additionally, the absence of Holocene records for benthic foraminifera requires clarification. Moreover, carbon isotope ratios of benthic foraminifera, which are typically measured alongside oxygen isotopes, should be incorporated into the discussion of bottom water properties; if they are not included, the reasons for their omission should be provided.

L 239-275 and L 292-311: About mechanism of open-ocean polynya

Comment: Recently, Narayanan et al. (2024, *Science Advances*, 10, eadj0777) discussed that the formation of open-ocean polynyas in 2016 and 2017 was linked to salt export driven by Ekman transport under strong easterly winds. Their study also explores the influence of the Southern Annular Mode (SAM). It is recommended that the authors address the consistency or divergence between the present interpretations and those of Narayanan et al. (2024), and incorporate this comparison into the discussion.

Reviewer #3

(Remarks to the Author)

I read the manuscript with interest and appreciate the effort of the authors. Unfortunately, I cannot recommend publication in *Nature Comms*. In general, I found that interpretations were not always supported by the data, and/or data were interpreted a certain way with little or no explanation as to why or exploration of alternative interpretations. As such, I suggest a longer-format paper, where some of the nuances of the data (some of which is found in the supplement) can be discussed and addressed in the text. Finally, I recommend a bit of caution when discussing SAM and other modern teleconnections during glacial periods; a safer bet is simply referring to westerly wind variations.

As a general comment, the number of proofreading errors throughout the manuscript was surprising. I observed many typos,

inconsistent referencing (missing citations; some references were numerical, some by name where not appropriate, etc.), and referencing errors (check all publication dates and spellings of authors names) in both the MS and the supplement. I do not have time to highlight or correct them, but care should be taken to ensure they are corrected in any future version.

A few things about the data:

1. The age model seems ok, but something should be said about the age reversals, the actual error vs analytical error, and some discussion of the sedimentation rate change at 19 ka is... required.
2. I was surprised not to see any reference to *N. pachy* habitat at the Antarctic Margin, where this core is located. It is well known (see Hendry et al., 2009) that *N. pachy* lives in brine channels in the sea ice, and this can impact all manners of geochemical results. That the benthic and planktic isotopes appear similar is maybe support for them living in the water column, maybe. Or, it may reflect another problem.
3. Following up on the above, why are the benthic and planktic stable isotope values so similar (and so positive at the LGM?). Also, how do the authors explain the reversal in benthic and planktic isotope gradient at 275 cm? I also do not see the significant variability on millennial timescales, as stated by the authors on line 126. How do the authors define significant? How is $\delta^{13}\text{C}$ of planktics (and *N. pachy* in the polar regions specifically) usually interpreted? Is it usually interpreted? As an aside, I was surprised to see NBS-19 still being used as a standard, as it is no longer available and has not been for years.
4. I notice that the authors state that theirs is the most southern Mg/Ca dataset ever published, but Hillenbrand et al (2017), Rathburn (1999), Mawbey et al. (2020), and others have published data from the Antarctic margin farther south than this site. Were any corrections applied to the Mg/Ca data (e.g., Evans, Gray)? This seems particularly important given the salinity interpretations. The discussion of the choice of temperature calibration should also be clearer. What is the error of the calibration chosen? Why might the coldest temperatures occur between 40 and 45 Ka? How might the different morphotypes of *N. Pachy* (s) seen in the SEM photos impact Mg/Ca data? All things to think about and address.
5. Temperatures for clumped isotopes and Mg/Ca below the freezing point of seawater suggest an issue (see habitat issue above and correction comment). This needs to be addressed and discussed in the text.

Version 1:

Reviewer comments:

Reviewer #1

(Remarks to the Author)

Review of the revised submission "Millennial-scale subsurface ocean warming in the eastern Weddell Sea during the last glacial period" by Pinho et al.

I thank the authors for revising their manuscript, especially describing paleo terminology better, improving the visual representation of the results and efforts taken to improve the manuscript.

Yet, I am not completely convinced by some statements made in the manuscript and would still appreciate some more guidance through some parts of the manuscript. Therefore I would like some more revisions as pointed out by the comments below.

Major comments:

Modern day polynya classifications

-In modern day physical oceanography, there is a distinction between the Weddell Sea Polynya in the 1970s and Maud Rise Polynyas in 2016 and 2017, as they are resulting from different processes and also have impacts on different scales.

-line 63: Several studies have shown that the formation of the Weddell Sea Polynya / Great Weddell Polynya was triggered by several processes and not only a density instability. Please rephrase the text.

- the studies used in lines 332-334 are referring to Maud Rise Polynyas, while the text refers to Weddell Sea Polynyas.

Please adjust the text accordingly.

-line 377-382: "Weddell Polynya" - the polynya in 2017 was a Maud Rise Polynya and the studies used as reference are focusing on Maud Rise Polynyas. Please correct the text accordingly.

Section "Glacial occurrence of the Great Weddell Polynya"

-The statement in lines 285-287 is very vague and raises the question how the authors define a "pronounced maxima"? If

one compares the maxima in Temperature given in Figure 3c, The maxima around 47 ka is larger than in 44 ka, and of the same salinity, so why does one of these periods indicate a rise of WDW and the other not?

-the lines 292-298, are very hard to follow and raise some questions, e.g.,: is the mentioned d18O a general one or one of the previously introduced ones? Please make this more understandable

Section “Glacial Forcing and feedback mechanisms”

-In the paragraph about the AMOC (lines 434-450) the correlation between subST and Pa/Th is not obvious in the Figures 3b and c. Right now this paragraph feels vague and would benefit from more support.

-please make clear in the text and not only the caption that the Pa/Th proxy is from a different core

Conclusions:

-I would recommend the authors to be more careful about the statement that the Great Weddell Polynya may represent a remnant of the last Ice Age, as formation processes are very different (glacial polynya: hybrid formation of coastal and open-ocean processes vs. only open ocean processes)

-The last statement in lines 521-523 would benefit from a citation.

Figure 2

-The figure has been added, which is a nice addition, yet Figure 2b is not at all referenced in the text, which leaves the question about its importance.

-I would recommend scaling the y-axis scales in 2a on the left and right in the same way. Currently the lines seem to match very well, but with a closer look one notices that the right axis is using a slightly different scale.

-The caption of the figure is at various places wrong, like the blue area is 15 ka not 20, there are diamonds instead of triangles,...

Figure 3:

-I appreciate the efforts taken to revise Figure 3.

-I appreciate the effort of the clarification in the text and caption regarding Figure 3g “Winter sea ice based on the relative abundance of diatom winter sea ice “. It is still not obvious, how someone who is not familiar with paleoceanography, as the audience of the journal might be, interpret a 12% Winter Sea Ice? Does it mean that 12% of the Weddell Sea were ice covered? Please provide even more guidance.

-The importance of the 3% threshold in Figure 3g is not clear, as it is only mentioned in the caption

-Are the blue bars in Figure 3 referring to the periods mentioned in lines 281-283, according to the information given I would assume so. If this is the case they do not match. Please clarify.

Minor comments:

line 20: It feels like here is a verb missing in connection with “either the stratification of the upper water column”

line 24: “The glacial polynyas formed off...”- this sentence is too speculative and not proven and I don't think it belongs in the abstract

line 46: here is a unit for the temperature missing

line 100: “ the low number of well -preserved...” - if this is a common problem it would be good to add a citation here.

line 104: “se” - seems to be in the wrong place

line 115: it would be good to introduce the abbreviation “subST”- I assume it is “subSurface Temperature”?

line 124: I assume there are brackets missing around “Osw_ivc”

line 149: “Atka Bay” is not shown in the map, either add it or rephrase caption.

line 164: Is it possible to show the EDML location in Figure 1?

line 180: referencing is wrong

line 228- 230: is this supposed to be seen from Figure 3? Or where does this information originate from?

line 407: Here is a dot missing after Plateau.

lines 438: wrong referencing.

line 492+493: AIM is introduced twice

line 512: I recommend removing the word ‘closely’ as the correlation between subST and AMOC is not well developed as previously discussed by the authors.

Reviewer #2

(Remarks to the Author)

Having reviewed the resubmitted manuscript, I confirm that all my previous comments have been addressed with either corrections or appropriate explanations. Consequently, I have no further comments to offer. I believe it has been developed into a very good paper.

Reviewer #4

(Remarks to the Author)

The upper ocean water column stratification between the Antarctic Polar Front and the continental margins of Antarctica consists of a low salinity near freezing surface layer over a relatively warm salty deep water. The pycnocline separating these layers is weak and sometimes if the surface layer is salty enough, it disappears in winter, enabling deep reaching convection. The convection cools the regional deep water, while the upward convective flux of relatively warm water (+0.5 C) into the surface layer inhibits winter sea ice formation, leading to an open ocean polynya, e.g. the Great Weddell Polynya of the mid-1970s. The cooled deep water spreads across the Antarctic Circumpolar Current, ventilating into the global deep water masses. The resultant deep ocean ventilation is not exactly like Antarctic Bottom Water (AABW) that is denser and forms along the continental margin of Antarctica.

Open ocean polynyas differ from polynyas over the continental shelf (coastal polynyas) that are forced by katabatic winds blowing newly formed sea ice seaward (exporting freshwater), leaving behind a cold, salty (dense) shelf water column, which upon export to the deep ocean forms a key ingredient of the classic global spreading AABW.

A factor in the formation of open ocean polynya, such as the Weddell Polynya of the mid-1970s is reduced precipitation and increased cyclonic circulation of the Weddell Gyre, which occurs during negative Southern Annual Mode (-SAM: Gordon et al, 2007, which Pinho et al cite).

Pinho et al using sea floor sedimentary data investigate the upper ocean stratification in the areas of the Dronning Maud Land Polynya (~6.5°W longitude, over the continental slope) during the last glacial period, 75K to 20K years ago. Dronning Maud Land Polynya occurs within the westward flowing coastal slope current of the Weddell Sea cyclonic Gyre.

As reported in the literature (Gordon et al, 2007; de Lavergne et al, 2014; Gordon 2014, which Pinho et al cite), southern ocean polynyas, notably the Weddell Sea Polynya, may have been more common during the glacial periods, and will become less common in the present warming climate.

Investigating the southern ocean polynyas during glacial period can lead to better understanding of global ocean deep water ventilation. Thus I view the Pinho et al as an important contribution that will encourage further research into the temporal behavior of southern ocean polynya for the present era and geological past.

The Pinho et al manuscript is a revision from an earlier ms. I was not a reviewer of that initial submission. I reviewed the 'rebuttal' and revised ms. The authors responded to the extensive initial review (and as usually the peer review, led to a much improved presentation).

While I endorse publication of the Pinho et al study there is an issue that limits the links between the Great Weddell Polynya and the . As noted by Reviewer #1 (point #6), the Dronning Maud Land Polynya is closer to the continental margin of Antarctica than the Great Weddell Polynya of the mid-1970s, which occurs within center of the Weddell Sea cyclonic gyre. The Dronning Maud Land Polynya is more like the coastal polynyas than the open ocean polynya and may not be a perfect 'precursor' 1970s open ocean event.

Pinho et al rebuttal of Reviewer 1 point stating: "Our results therefore indicate a hybrid glacial polynya system where open-ocean deep convection processes were enhanced by katabatic wind forcing from the nearby expanded ice sheet during the last glacial period." And they added the following clarification in lines 396-402:

"During glacial periods, when grounded ice sheets advanced across the Antarctic continental shelves, coastal shelf polynyas similar to those observed today 17,18,81 were likely absent. Instead, the ice advance would shift the occurrence of glacial polynyas northward to regions above the deep ocean (Fig. 4a). Although the polynya itself was located in open ocean, strengthened katabatic winds blowing across the expanded ice sheet are considered an important mechanism for the polynya formation."

They may be right in relating the glacial age Dronning Maud Land Polynya to the Great Weddell Polynya, but more support for this is needed. I suspect that during the glacial age, the strong katabatic winds reached out to the continental slope (as glacial ice covered most of the continental shelf), which coincided with strong -SAM making the central Weddell Sea gyre more prone to open ocean polynyas (Gordon 2014).

I suggest that Pinho et al, make a stronger case for linking the Dronning Maud Land Polynya to the Great Weddell Polynya.

I suggest a title that mentions the glacial age Dronning Maud Land Polynya.

Version 2:

Reviewer comments:

Reviewer #1

(Remarks to the Author)

Review of the revised submission "Millennial-to-orbital-scale subsurface ocean warming and Polynya formation off Dronning Maud Land and during the last glacial" by Pinho et al.

I thank the authors for revising their manuscript, implementing all previously made comments. During the last read I found one typo: "interrupted" in line (63) and I guess "glacial Weddell polynya" in line 520 has been missed during the renaming to

“(glacial) DML polynya” from the previous version. Otherwise, I have no further comments on the paper.

Reviewer #4

(Remarks to the Author)

I support publication. The author have received constructive comments from the reviewers and have responded effectively. Hopefully their paper will generate lots of interest in the spatial and temporal variability of the multitude of polynyas types in the southern ocean.

I do have a general concern about the primary driver of the Dronning Maud Land Polynya:

The authors suggest that warmer temperature of the WDW drives the polynya: warmer WDW injected into the surface layer over a topographic feature, spurs a polynya, and increased precipitation over the coastal region. Why is WDW warming during the polynya (MIS4 and LGM) phase? The authors suggest that more extensive sea ice cover to the north enables warmer WDW spreading southward within the eastern limb of the Weddell Sea Gyre to reach the Dronning Maud Land region. They might be right.

Alternatively, warmer WDW might be due to increased southward flux of circumpolar deep water (CDW) heat across the ACC? This could be driven by stronger west to east winds over the ACC, producing a stronger ACC and increased eddy activity.

Or as suggested in the literature the saltier surface layer, due to decreased precipitation (-SAM) weakens the pycnocline and can initiate deep convection, which injects heat into the surface layer and a polynya.

So which is the primary driver of the Dronning Maud Land polynya: saltier surface water or warmer WDW?

“Subsurface” is often used. Is this the surface layer (ASW) above the pycnocline? The pycnocline? Or the WDW? I think they imply the subsurface is WDW. Why not call it WDW?

I like figure 4. I wonder if map panels might be added, showing WDW spreading for glacial and for non-glacial periods?

Dear Reviewers,

We thank Reviewers for their valuable comments/suggestions, which we have thoroughly analyzed and implemented. They are very constructive and appropriate. We believe that through their implementation the manuscript has greatly improved. We provide Word document files with tracked changes of the revised version of the manuscript. To simplify the evaluation of the performed modifications we (i) copied below in black/italic the comments/suggestions from Reviewers, and (ii) provided a detailed response to each comment/suggestion in blue/not-italic. All line numbers mentioned in our responses refer to the revised version of the manuscript with tracked changes.

Reviewer #1

Reviewer #1 (Remarks to the Author):

This study investigates the occurrence of open-ocean polynyas in the Weddell Sea during the glacial period from 75 000 - 20 000 years before present. By analysing proxy data from the Weddell Sea the changes in the subsurface ocean are derived. The results indicate occurrence of polynyas during this time and hypothetical formation processes are outlined.

The manuscript is interesting to read, but I would recommend some revisions. In a revision the authors' should provide more guidance of paleo terminology for the broader field of ocean and atmospheric science, clarify some messages and do structural changes, improve the visual representation of their graphics, and a careful review of the cited literature and referencing of figures. Further, some structural changes for better readability would improve the accessibility of the results. Yet, the manuscript provides novel insights in the past of polynyas.

Response #1. We sincerely thank Reviewer #1 for all the constructive comments and suggestions provided throughout this round of revision. We agree that clearer terminology would make our work more accessible to the broader ocean and atmospheric science community. In our revision, we have added extended descriptions of key paleoceanographic terms. We have made some restructuration to improve the logical flow of our arguments as suggested by Reviewer #1. We have significantly improved our figures based on your suggestions. We have conducted a thorough review of our citations and references.

We believe these revisions have substantially improved the manuscript's accessibility and scientific rigor while maintaining the novel insights into past polynya occurrence. We appreciate

the reviewer's recognition of the novelty of our work and hope our revisions adequately address all concerns raised.

Major Comments

Paleo terminology:

It would be useful for readers, who are not familiar with proxies, to explain more clearly which proxy represents which ocean variable and which combinations are used (lines 91-105).

Response #2. We thank Reviewer #1 for this important suggestion to improve accessibility for readers unfamiliar with paleoceanographic proxies. We have made some changes to provide clearer guidance for non-specialist readers. We also clarified, step-by-step, how the multi-proxy approach is applied in the following section (now lines 93–128).

*“For the last glacial period, the sediment archive on Bungenstock Plateau provided an exceptionally continuous occurrence of abundant and well-preserved calcitic foraminifers, which enabled the use of multiple carbonate proxies. This is rather rare in sediments around Antarctica. We measured stable oxygen and carbon isotopes ($\delta^{18}\text{O}$, $\delta^{13}\text{C}$), Mg/Ca ratios, and clumped isotopes (Δ_{47}) on the calcitic shells of the subsurface-dwelling planktonic foraminiferal species *Neogloboquadrina pachyderma sinistral* (*N. pachyderma sin.*). The use of these proxies was limited to the last glacial period and unfortunately not applicable to the deglacial and Holocene periods. The low number of well-preserved planktonic foraminifera in the deglacial and Holocene sediment samples only permitted a low-resolution $\delta^{18}\text{O}$ record to be generated but prevented the measurement of further Mg/Ca and clumped isotopes (Δ_{47}) proxies. For the glacial period, however, the abundant occurrence of planktonic se foraminifer allowed us to measure Δ_{47} , Mg/Ca, $\delta^{18}\text{O}$ and $\delta^{13}\text{C}$ and reconstruct past variations in ocean temperature, salinity and nutrients recorded at the preferred habitat-depth of *N. pachyderma sin.* at ~50 – 150 m^{19,20} (Supplementary Fig. 4). For the temperature reconstruction and its verification, we measured two independent parameters, Δ_{47} and Mg/Ca. With this approach, we aimed to test whether the often too warm Mg/Ca-based temperature reconstructions for cold polar marine environments (Refs. ^{21–27}) could be corrected with Δ_{47} data in order to solve a long-standing problem of polar paleoceanography. Ten Δ_{47} -based temperatures are paired with independent Mg/Ca data to support and verify the absolute Mg/Ca-derived subsurface temperature range, which is made possible by good agreement between the two independent temperature proxies (see Methods; Supplementary Fig. 8). Our approach resulted in modified Δ_{47} -based $\text{subST}_{\text{Mg/Ca}}$ variations, hereafter referred to as $\text{subST}_{\text{Mg/Ca vs. } \Delta_{47}}$, with lower temperatures and amplitudes compared to other temperature calibrations (see Supplementary Fig. 5). From 75 to 20 ka, $\text{subST}_{\text{Mg/Ca vs. } \Delta_{47}}$ vary between 1.9 and -2.75°C (Fig. 3c, Supplementary Fig. 5). Considering the freezing point of sea water, the minimum value appears about 0.7°C too low, which however lies within the error range of our reconstruction (Fig. 3). The*

Δ_{47} -based temperature data are in good agreement with the $\text{subST}_{\text{Mg/Ca vs. } \Delta_{47}}$ reconstruction (Fig. 3c) (see Methods; $r=0.75$; Supplementary Fig. 8). For the salinity reconstruction, the correction of the foraminiferal $\delta^{18}\text{O}$ record for changes in temperature ($\text{subST}_{\text{Mg/Ca vs. } \Delta_{47}}$) and global ice volume²⁸ is used as a qualitative proxy for seawater salinity $\delta^{18}\text{O}_{\text{sw-ivc}}$ ²⁹. The planktonic $\delta^{13}\text{C}$ ratio serves as a proxy for the dissolved inorganic carbon isotope composition ($\delta^{13}\text{C}_{\text{DIC}}$) of the surrounding water, which provides information on its nutrient inventory and source water mass. For example, the Pleistocene $\delta^{13}\text{C}_{\text{DIC}}$ values of Antarctic WDW range below 0.6 ‰, while those of the ASW are significantly higher^{30,31}.”

We believe these enhancements make our methodology much more accessible to readers from the broader ocean and atmospheric science community while maintaining the technical rigor required for specialist readers.

A more detailed explanation on $\text{subSTMg/Ca vs. } \Delta_{47}$ is needed, which seems the key result of the analysis. It is not clear how this data was derived or where these results are presented, as Figure 2d shows a line for subSTMg/Ca and diamonds for $\text{subST}_{\Delta_{47}}$, but not the always referred $\text{subSTMg/Ca vs. } \Delta_{47}$.

Response #3. We agree that more detailed explanation is needed for our combined temperature approach, which is indeed central to our analysis. We have made several important clarifications in the revised manuscript. We have removed the confusing terms " $\text{subST}_{\text{Mg/Ca}}$ " and " $\text{subST}_{\Delta_{47}}$ " throughout the manuscript and replaced them with the clearer designation " $\text{subST}_{\text{Mg/Ca vs. } \Delta_{47}}$ " to correctly indicate our combined temperature approach. We now explicitly state in the introduction (lines 107-122) that we combine two independent temperature proxies: Mg/Ca ratios and clumped isotope (Δ_{47}) in planktonic foraminifera.

“For the temperature reconstruction and its verification, we measured two independent parameters, Δ_{47} and Mg/Ca. With this approach, we aimed to test whether the often too warm Mg/Ca-based temperature reconstructions for cold polar marine environments (Refs. 21–27) could be corrected with Δ_{47} data in order to solve a long-standing problem of polar paleoceanography. Ten Δ_{47} -based temperatures are paired with independent Mg/Ca data to support and verify the absolute Mg/Ca-derived subsurface temperature range, which is made possible by good agreement between the two independent temperature proxies (see Methods; Supplementary Fig. 8). Our approach resulted in modified Δ_{47} -based $\text{subST}_{\text{Mg/Ca}}$ variations, hereafter referred to as $\text{subST}_{\text{Mg/Ca vs. } \Delta_{47}}$, with lower temperatures and amplitudes compared to other temperature calibrations (see Supplementary Fig. 5). From 75 to 20 ka, $\text{subST}_{\text{Mg/Ca vs. } \Delta_{47}}$ vary between 1.9 and -2.75°C (Fig. 3c, Supplementary Fig. 5). Considering the freezing point of sea water, the minimum value appears about 0.7°C too low, which however lies within the error range of our reconstruction (Fig. 3). The Δ_{47} -based temperature data are in good agreement with the $\text{subST}_{\text{Mg/Ca vs. } \Delta_{47}}$ reconstruction (Fig. 3c) (see Methods; $r=0.75$; Supplementary Fig. 8).”

However, we maintain a more specialized and detailed explanation on the combination approach and validation methodology in the supplementary material to avoid overwhelming the

main text with technical details. The figure legend now explicitly states that "clumped isotope (Δ_{47}) temperatures are shown as calibration references for the Mg/Ca-derived temperature record.

These changes provide the detailed explanation requested while clarifying how our key temperature dataset was derived and validated, making our methodology accessible to both specialist and non-specialist readers.

Further it would help the reader if some information about the temporal resolution of the results derived from the used ice core was provided.

Response #4. We agree that more information about temporal resolution would help readers better understand the scope and limitations of our analysis. We have added a new phrase (lines 169-172) that informs the reader about the temporal resolution capabilities of our sedimentary record:

"The anticorrelation is pronounced during the time interval 75 – 43 ka BP given the comparatively higher sedimentation rate, which is twice as high as during the interval from 43 – 30 ka BP (Supplementary Fig. 2d), allowing to examine millennial-scale changes with improved temporal resolution."

See also lines 434-440:

"On millennial timescales, the variability of $subST_{Mg/Ca \text{ vs. } \Delta_{47}}$ appears to be related to changes in the Atlantic Meridional Overturning Circulation (AMOC) (Fig. 3b), especially during the interval from ~75 – 38 ka BP (temporal resolution of ~125 years between adjacent samples), when $subST_{Mg/Ca \text{ vs. } \Delta_{47}}$ maxima coincide with AMOC maxima and vice versa (Fig. 2b, c). This correlation is less well developed in the subsequent time interval of ~38 – 20 ka BP, which is partly due to the lower temporal resolution (~250 years between adjacent samples) and the suppressed variability of the $subST_{Mg/Ca \text{ vs. } \Delta_{47}}$ record."

This addition clarifies that our detailed millennial-scale investigation is more focused on the 75 – 43 ka BP interval where our sedimentary record provides the highest temporal resolution, helping readers understand both the strengths and temporal constraints of our dataset. Note that we have provided a detailed age model figure in Supplementary Fig. 2 presenting the sedimentation rate throughout the entire study period (75 – 20 ka BP), which allows readers to assess the temporal resolution capabilities across the full record.

Moreover, abbreviations like MIS, AIM, CI, SMOW, VPDB should be explained in the manuscript and if processes like HS, AIM are important in relation to the polynya, they should briefly be summarized for readers unfamiliar with them.

Response #5. We agree that clearer explanation of abbreviations and key processes would improve accessibility for readers unfamiliar with paleoclimatic terminology. We have made the

following additions to enhance clarity. We have briefly clarified the abbreviations CI, MIS, AIM, and HS in the revised version of our manuscript.

The abbreviation “MIS” is defined in the legend of Figure 2:

“MIS = Marine Isotope Stages, which refers to past periods defined by $\delta^{18}\text{O}$ values, representing warm and cold climate intervals are used as stratigraphic markers³⁷ (see panel “a”).” (lines 199-201).

The term HS has been briefly described in lines 444-445.

“HS events are defined by cold glacial stadials in Greenland, associated with AMOC weakening events (Fig. 3b).”

We have added a brief explanation of AIM, which is well correlated with HS (see lines 489-494). For clarity, we have also defined the glacial stadials and interstadials in Greenland and Antarctica, highlighting their anticorrelated relationship.

“The majority of models suggest that the millennial-scale latitudinal shifts in the westerly winds resulted in a northward expansion of the belt of very cold and dry polar air, accompanied by strong katabatic winds and sea-ice propagation during glacial Antarctic cold stadials (Greenland interstadials) and, conversely, during warm interstadials known as Antarctic Isotope Maxima (AIM) (Greenland stadials, HS).”

Clarification of messages

The section “Glacial occurrence of the Great Weddell Polynya” interprets and compares the results to modern open-ocean polynyas. Yet, in the section “Glacial Forcing and feedback mechanisms” especially point (2) indicates that processes related to modern coastal polynya formation, like katabatic winds, are important. I would recommend the authors to clarify if point (2) is related to the previously shown results or if in the case of the here studied polynya a combination of processes is important and if the results here indicate a polynya comparable to a modern open-ocean polynya or coastal polynya.

Response #6. We thank Reviewer #1 for this important observation regarding the apparent inconsistency in our polynya process interpretations. We have clarified this confusion by adding explanatory text that better describes the glacial polynya scenario and how different formation mechanisms interact.

We have added the following clarification in lines 396-402:

“During glacial periods, when grounded ice sheets advanced across the Antarctic continental shelves, coastal shelf polynyas similar to those observed today^{17,18,81} were likely absent. Instead, the ice advance would shift the occurrence of glacial polynyas northward to regions above the deep ocean (Fig. 4a).

Although the polynya itself was located in open ocean, strengthened katabatic winds blowing across the expanded ice sheet are considered an important mechanism for the polynya formation.”

Our results therefore indicate a hybrid glacial polynya system where open-ocean deep convection processes were enhanced by katabatic wind forcing from the nearby expanded ice sheet during the last glacial period.

I would recommend to restructure the first 2 paragraphs in the Section ‘Glacial occurrence of the Great Weddell Polynya’ to tell the reader, where the species live and how it moves in the vertical and then interpret this with shoaling/ deepening of the water masses (lines 193-228)

Response #7. Yes, you are right. We have moved the first 2 paragraphs in the section "Glacial occurrence of the Great Weddell Polynya" to the section "Glacial upper-ocean stratification" to better guide the reader through the ecological context before interpreting the paleoceanographic implications (lines 242-271). It appropriately fits the last phrase from the previous paragraph, where we introduced there the habitat depth concept (line 238-241).

In the revised version of our manuscript, we have added a brief explanation of how independent lines of evidence indicate that our geochemical foraminiferal proxies are not influenced by sea ice (lines 258-263).

*“Although the Antarctic *N. pachyderma* genotype may be adapted to sea ice conditions⁵⁵, strong agreement between the independent temperature proxies Δ_{47} and Mg/Ca (Supplementary Fig. 8), corroborated by *N. pachyderma sin.* $\delta^{13}C$ (Fig. 3c, f), and supported by our apparent calcification depth estimate (Supplementary Fig. 4), provides evidence that our foraminiferal geochemical proxies are not affected by sea ice (Refs. ^{22,56}).”*

In lines 271-275, the authors claim that the Bungenstock Plateau works similarly as Maud Rise, and explain that isopycnals are doming over the plateau, yet Figure 1 c,d) which show transects across the Plateau, do not show an indication of isopycnal doming. I would recommend the authors to show more proof of the statement that Bungenstock Plateau works similarly as Maud Rise.

Response #8. We agree with Reviewer #1 on the need to provide supporting evidence for WDW updoming at the Bungenstock Plateau. We have added Supplementary Fig. 9, an east–west transect that clearly illustrates WDW updoming at the Bungenstock Plateau. This transect follows the preferred westward WDW circulation pathway within the Weddell Gyre (Fig. 1a). The WDW updomes from the Gunnerus Ridge at 30°W across our site location.

The link of the investigated polynyas to the AMOC needs to be more incorporated (lines 358 - 383), the current manuscript does not provide an easily accessible link of those two processes. Large open-ocean polynyas, like the Great Weddell Sea polynya, are known to form large amounts of AABW, which could play a role in the global overturning, it would be good if the authors could expand the discussion towards this.

Response #9. We agree with Reviewer #1 that the connection between AMOC and the glacial Weddell polynya requires clearer connection. We acknowledge that large-scale Great Weddell polynyas would result in enhanced AABW formation, which is crucial for global overturning circulation. However, our limited benthic foraminiferal results prevent us from making definitive assumptions regarding AABW production at our specific site.

We have added a new paragraph that addresses this issue and provides direction for future studies (lines 463-470):

“The large-scale Weddell polynya is recognized for producing substantial volumes of AABW¹⁶, and thus potentially influencing the global overturning circulation. At our study site, however, the small and relatively stable benthic-planktonic $\delta^{18}\text{O}$ offset is masked by a relatively large uncertainty range, which unfortunately does not allow a reliable interpretation on the mixing depth and the formation of AABW (Fig. 4a). Future research should incorporate additional proxy reconstructions to better assess the role of AABW formation in the context of the glacial Weddell polynya and the deep convective processes between Maud Rise and Bungenstock Plateau.”

Please note that we are presenting our benthic $\delta^{18}\text{O}$ record as a stratigraphic tool rather than a main interpreted proxy. We acknowledge that further analyses on benthic foraminifera are needed for thorough examination of deep water dynamics, such as AABW formation and transport, which is beyond the scope of the current study.

The discussion of polynya formation remains focused on millennial-scale responses to regional ice–ocean–atmosphere interactions in the context of the bipolar seesaw, which is strongly linked to changes in the strength of the AMOC. See the descriptive paragraph in lines 451-462:

“The positive correlation between millennial Southern Ocean subsurface warmings and AMOC maxima (Figs. 3b, c) suggests an interhemispheric teleconnection and a forcing consistent with the contrasting temperature changes in Greenland and Antarctica and the bipolar seesaw paradigm⁸⁷. Evidence from climate proxies and model results shows that glacial intensification in the AMOC during Northern Hemisphere interstadials is associated with a northward shift of the Intertropical Convergence Zone (ITCZ) and the Southern Hemisphere westerly winds, Antarctic cooling, and a northward extension of the sea-ice margin in the Southern Ocean^{88–90}. Such conditions are similar to the “-SAM” as mentioned above and would help to precondition the polynya formation by accumulating heat in the subsurface and reducing

salinity in the surface layer, so that the water column becomes buoyantly unstable (Fig. 4a). Strong katabatic winds could then tip the system into a convective state.

The important role of obliquity forcing is pointed out towards the end of the manuscript (lines 248,420-423) yet if it is an important finding regarding the formation of polynyas, I would recommend to highlight this earlier in the manuscript.

Response #10. We agree with Reviewer #1 that the important role of obliquity forcing should be highlighted earlier in the manuscript. We have made the following additions to introduce this key finding in the early sections:

In the first section "Glacial upper-ocean stratification", we have added the following text passage in lines 178-186:

"Superimposed on the millennial-scale $subST_{Mg/Ca \text{ vs. } \Delta 47}$ variability are patterns of orbital-scale variability, that appear to be associated with cyclic variations in Earth's obliquity (Fig. 3c, d). This is not entirely surprising, since changes in obliquity cause variations in seasonality and strongly affect the total summer energy budget received at high latitudes with a period of ~ 41 thousand years. The interval from $\sim 37 - 20$ ka BP, which includes the Last Glacial Maximum (LGM; $\sim 26 - 19$ ka BP, Ref. ⁴⁰) and the penultimate glacial Marine Isotope Stage (MIS) 4 ($\sim 71 - 59$ ka BP, Ref. ⁴¹) are both characterized by long-term $subST_{Mg/Ca \text{ vs. } \Delta 47}$ maxima coincident with minima in obliquity. The long-term $subST_{Mg/Ca \text{ vs. } \Delta 47}$ minimum in between is associated with an obliquity maximum."

In the section "Glacial occurrence of the Great Weddell Polynya," we have mentioned again the keep reader's attention to the obliquity forcing while adding contextual information (lines 274-277):

"During the LGM and partly also during MIS 4, the geographic extent of ice sheets and glacier cover was largest in the last 75,000 years. These time intervals correspond to low-obliquity phases with exceptionally cold Antarctic air temperatures during the last glacial period (Fig. 3a, c)."

These additions ensure that readers are introduced to the obliquity forcing concept early in the manuscript, before encountering the detailed discussion later in the text. Given that obliquity is an external forcing, we kept the principal discussion in the "Glacial Forcing and Feedback Mechanisms" section.

Citation of references

In several places in the manuscript the citations do not support the statement, e.g., the citation 56 (Gordon, 1991) is not an online resource and the cited results are not quantifiable. In the context of line 270, where this citation is used, a potential citation could be Cheon and Gordon

(2019); citation 43 (Gordon et al. 2007, line 271), the study does not investigate the role of a barotropic gyre spin-up as initial trigger for the Great Weddell Polynya; citation 73 (Cheon and Gordon (2019), line 345) in this work the role of katabatic winds is not investigated. In some places, the reference is located in the wrong place in the text, as for example, line 88.

Response #11. We sincerely appreciate Reviewer #1 for the careful attention to citation accuracy. We have thoroughly conducted a comprehensive review of all references in the manuscript to ensure accuracy, relevance, and proper placement throughout the text. Furthermore, we have made several formatting changes in the "References" section.

Visual representation of graphics:

In general the Figures 1 and 2 would need improvement for better readability and access for readers.

Figure 1: The use of discrete color levels will help the reader to interpret the data shown more easily. A diverging colormap, like the red/blue one, should have the white area at a value of 0 and often different colormaps are used for different ocean variables.

Response #12. We agree with Reviewer #1 that substantial visual improvements are needed in the figures. We have made several important changes to enhance readability and accessibility:

We have separated the first panel of the previous Figure 2 to create a standalone new Figure 2. This figure presents stratigraphic information and biogenic carbonate availability, including a comparison between our benthic and planktonic $\delta^{18}\text{O}$ records against the reference Deep South Atlantic $\delta^{18}\text{O}$ stack (Lisiecki & Stern, 2016) as well as the age-depth relationship based on our age model. One important message conveyed by this figure is that the Holocene period significantly lacks foraminifera for further measurements, such as Mg/Ca and Δ_{47} .

We have implemented discrete color levels for panels "a" and "c" as recommended, which significantly improves data interpretation. For panel "d," we chose a gradient colormap because it represents the same temperature profile and does not complicate visual assessment.

Without the benthic and planktonic $\delta^{18}\text{O}$ records now shown in the new Figure 2, we can focus Figure 3 on the timeframe discussed in the manuscript, which covers solely the last glacial period. This temporal focus eliminates confusion about data availability and concentrates reader attention on our key findings.

These changes substantially improve figure clarity and make our data more accessible to readers while maintaining scientific accuracy.

Further, please add a mark for Maud Rise in a) as its position is not commonly known and the text refers several times to it. Please add the transects shown in c) and d) to a) or b) so readers gain some orientation. Also please provide details for data shown in a,c,d (e.g., are those observations, models, observational climatologies).

Response #13. Done. We included the N-S transect accordingly and provided details on the used data, which is retrieved from World Ocean Atlas 2023 (lines 150-151).

Figure 2: There is a lot of information in this figure which makes the figure very hard to read. It is not obvious, which of the information are retrieved from the here presented ice core PS128_2 and which are from other studies. Even though the presentation is common for Paleo studies, it is not very common in other fields, e.g. having the youngest date on the right and the oldest on the left.

Response #14. We thank Reviewer #1 for the valuable comments regarding Figure 2's complexity and readability issues. We have made several significant changes to Figure 3 (formerly Figure 2) in the revised version to address these concerns:

Each panel now includes detailed information regarding the data location and clearly differentiates other studies from our results from marine sediment core PS128_2. We have added distinct visual markers and color coding to separate our data from reference records. Each proxy record now includes the core/site identifier in the legend.

The reset timeframe in Figure 3 removed blank spaces and more homogeneously organizes the visual information. This change improves the visual assessment of the focused glacial period instead of showing non-interpreted periods, eliminating confusion about data availability and concentrating reader attention on our key findings.

These changes substantially improve figure accessibility while maintaining the scientific integrity and comparative context essential for paleoceanographic studies.

Further, the coloring of some periods is confusing, especially as the blue red gradient in some parts is similar to the colorbars used for ocean and air temperature, is the coloring needed? Are the highlighted periods exactly the periods of polynya occurrence? The text refers to 3 periods of polynya occurrence (38-20ka, 59-56ka and MIS4) yet there are 5 periods highlighted.

Response #15. We agree with Reviewer #1 that the coloring was confusing and potentially misleading. The blue-red gradient coloring has been completely removed from the figure as it was not essential and created visual confusion with the temperature colorbars used elsewhere.

We have revised and homogenized the periods of polynya occurrence between the text and figure. In the revised version of the manuscript, we have more clearly detailed the periods of polynya occurrence to match exactly what is shown in the figure. We also identified a new interval of polynya occurrence, which is now properly discussed in the text.

The revised text now states (lines 281-284):

“During several intervals of the last glacial period, specifically ~38 – 20 ka BP, (including the LGM), 44 – 41 ka BP, 52 – 48 ka BP, ~59 – 56 ka BP, 70 – 65 ka BP and at 73 ka BP (Fig. 3), the cold and fresh ASW likely did not dominate the upper 200 m of the water column above the Bungenstock Plateau, as it does today.”

This revision ensures complete consistency between the figure highlighting and the text description, eliminating the previous discrepancy between the mentioned periods and highlighted periods. The figure now clearly shows all six identified polynya occurrence intervals without confusing color gradients.

It is not clear what is shown in figure 2h. Is it sea ice extent or a concentration or a sea ice extent of the area? The units do not fit the caption and text (line 252)

Response #16. We have clarified what is shown in Figure 3g (formerly Figure 2h) in both the figure legend and manuscript text. The data represents the relative abundance of diatoms, specifically used as a qualitative proxy for winter sea ice conditions. Values are expressed as relative abundance (%) of these diatom species within the total diatom assemblage.

The figure legend now clearly states (lines 217):

“Winter sea ice based on the relative abundance of diatom winter sea ice^{51,52}.”

Therefore, the relative abundance of diatoms, support the statement in lines 311-315.

Throughout the text of Section “Glacial upper-ocean stratification” the referencing of Figure 2 is not aligning with the Figure itself.

Response #17. We thank Reviewer #1 for identifying this referencing issue. We have thoroughly reviewed the "Glacial upper-ocean stratification" section and corrected all figure references to align with the current figure numbering and content. All references to Figure 2 (now Figure 3) have been updated throughout this section, and we have verified that each figure reference corresponds accurately to the appropriate panel and data being discussed. This issue has been completely resolved in the revised manuscript.

Minor Comments:

lines 42-43: does the description refer to the sections shown in Figure 1c,d?

Response #18. Yes, we changed the phrase accordingly (line 42).

lines 47-50: indicate that temperature and salinity are both contributing equally in density and stratification, yet salinity is the main contributor to density in polar regions.

Response #19. Done (line 47-50).

“Despite the generally equal contributions of temperature and salinity to density and stratification, salinity is the main contributor in polar regions¹. Even small perturbations in the forcing and feedback mechanisms that affect mainly the salinity budget can strengthen or completely destroy the stratification”.

line 63: the Great Weddell Polynya in the 1970s was not only triggered by density instabilities, it's one of many important processes, please rephrase this sentence.

Response #20. Yes. We have changed the phrase accordingly (lines 63-68).

“During the winters of the mid-1970s, the density instability in the upper-ocean led to the breakdown of the stratification and has been an important contributor to the formation of the Great Weddell Polynya (250,000 km², Fig. 1a), which ventilated the deep ocean by convective overturning to depths of ~2,700 m and extended from the Maud Rise southwestward along the Bungenstock Plateau (e.g. Refs. ^{16,17}).”

line 69: it is not known for sure, that there is no ice formation in the Great Weddell Polynya, as it has shown for preceding smaller scale polynyas that they can form large amounts of sea ice (see Zhou et al., 2022), please rephrase this sentence

Response #21. Done. We have rephrased the sentence to acknowledge the uncertainty regarding sea ice formation in the Great Weddell Polynya and incorporate recent findings.

“It has been argued that the overturning has maintained the open-ocean polynya through the massive release of heat from the deep ocean, bringing WDW into the surface layer which may have limited the formation of winter sea-ice compared to periods without polynya formation ^{16,17}.”

This revision recognizes that while polynya formation may reduce sea ice production relative to non-polynya conditions, it does not completely prevent sea ice formation, reflecting the more nuanced understanding from recent research. Please note that the sea ice context remains focused on the comparison between past polynya versus no-polynya scenarios as intended for this study.

lines 85: define ka

Response #22. Done (lines 85-87).

“Our proxy record is from ~70 km north of the shelf break and covers the last glacial period from ~75 – 20 kilo annum (ka) before present (BP).”

lines 90: it would be worth expanding this a bit more, as in the manuscript feedbacks are also explored.

Response #23. Yes. Done (lines 89-92).

“The presence of an open-ocean glacial polynya and the associated ice-ocean-atmosphere feedback mechanisms are challenging to reconstruct and have never been demonstrated with palaeoceanographic data.”

line 130: EDML is undefined.

Response #24. Done (line 164).

“The $\delta^{18}O_{NPS}$ record is not clearly related to changes in Antarctic air temperature, as reflected by the EPICA Dronning Maud Land (EDML) $\delta^{18}O$ ice core record ³⁸ (Fig. 2a and 3a), nor to changes in $subST_{Mg/Ca}$ vs. $\Delta 47$ (Fig. 3c).”

line 133: age model, what does it mean, this word is used for the first time here

Response #25. We acknowledge the confusion regarding the first use of the term "age model" without proper explanation, though it is quite common in paleoceanographic studies. We have revised the phrase to provide clearer context for readers unfamiliar with paleoceanographic terminology (lines 167-169).

“This anticorrelation is robust despite chronological uncertainties in the age model of core PS128_2.”

line 147-150: purpose of this statement it feels right now unconnected.

Response #26. Done. We have rephrased this statement to better clarify the similarities and differences between MIS4 and LGM periods and their connection to our main findings (lines 182-185).

“The interval from ~37 – 20 ka BP, which includes the Last Glacial Maximum (LGM; ~26 – 19 ka BP, Ref. ⁴⁰) and the penultimate glacial Marine Isotope Stage (MIS) 4 (~71 – 59 ka BP, Ref. ⁴¹) are both characterized by long-term $subST_{Mg/Ca}$ vs. $\Delta 47$ maxima coincident with minima in obliquity.”

The red and blue lines in Figure 2 represent similar patterns, therefore the statement and the link to the $d18O$ proxies is not clear.

Response #27. We have deleted the red and blue lines in order to eliminate visual confusion and improve figure clarity. The similar patterns between these lines were indeed creating ambiguity regarding the relationship to $\delta^{18}O$ proxies. See now our figure 3, former Figure 2.

line 186: “glacial interval” - which period is meant?

Response #28. We acknowledge the ambiguity in our use of "glacial interval". Here we meant the time intervals during the entire last glacial period where temperature, salinity and planktonic $\delta^{13}\text{C}$ records are well correlated.

line 206: 'preventing further sea-ice from forming' - rephrase

Response #29. Done (line 289).

line 212: Are you referring to your results or Greco et al. ?

Response #30. We acknowledge the confusion regarding the attribution of results. We refer to our results, which are supported by Greco et al. 2019.

line 254: is kyr =ka?

Response #31. The abbreviation "kyr" is usually a time span, while "ka" refers to an age or time before present, which implies a point in time in the past rather than a duration.

line 255: "the described sea-ice-driven process" are the authors referring here to Haumann et al.?

Response #32. Yes, we have included the citation to clearly refer to Haumann et al. in line 315.

line 266: "main trigger" -the topographic effect is a potential trigger, not the main, as the polynya formation in the 1970s is a combination of processes - please rephrase

Response #33. Done. We have clarified it in the phrase accordingly (lines 324-327).

"While wind-induced and sea-ice-related processes likely prime the system for the development of an open-ocean polynya and deep convection in our study region, the topographic effect of seamounts has been proposed as one of the main potential triggers^{53,66}."

line 282: what does "glacial increase" in this context mean?

Response #34. We have restructured the phrase to correctly refers to higher alkalinity levels in deep waters during glacial periods compared to interglacial (lines 344-348):

"Such glacial archives are the exception on the Antarctic continental margin and have been interpreted to reflect enhanced biological productivity in a polynya setting^{34,44}, whereby the glacial increase in deep water water alkalinity in Weddell Sea favored the preservation of calcitic shells⁷¹."

line 282: doubling of the word "water".

Response #35. Done (line 344-348). See previous response.

lines 276-290, does this statement refer to only the LGM or all polynya periods?

Response #36. This statement refers not only to the LGM but probably also to preceding glacial periods as for comparison and temporal contextualization. This serves to show that in our study region the Weddell polynya is favored by a glacial background. We have added a new phrase that clarifies it (lines 348-349):

“This implies that open-ocean polynyas were present during the last glacial period as well as in preceding glacials in our study region.”

lines 312-335: some of these statements are not fully aligned with results and statements in previous sections, e.g., the formation of sea ice - previous statements throughout the text said there is no ice formation in polynyas.

Response #37. As already clarified, our study is focused on the comparison between polynya versus no-polynya scenarios and as such, sea ice formation is relative to this comparison and not only limited to polynya formations. The schematic figure 4 shows that sea ice formation was not completely prevented during polynya formation. We have included a text passage which clearly states that during polynya formation sea ice extent is relatively smaller than in periods without polynya (lines 68-71). Our interpretations are based on relative comparisons rather than absolute statements about sea ice absence.

See the revised text in lines 68-71:

“It has been argued that the overturning has maintained the open-ocean polynya through the massive release of heat from the deep ocean, bringing WDW into the surface layer which may have limited the formation of winter sea-ice compared to periods without polynya formation ^{16,17}.”

line 359: point to the figure showing AMOC

Response #38. Done (line 435).

line 362: what is the temporal resolution

Response #39. We have included the temporal resolution for both time intervals to help readers understand the analytical constraints and data interpretation (lines 434-440).

“On millennial timescales, the variability of $subST_{Mg/Ca vs. \Delta 47}$ appears to be related to changes in the Atlantic Meridional Overturning Circulation (AMOC) (Fig. 3b), especially during the interval from $\sim 75 - 38$ ka BP (temporal resolution of ~ 125 years between adjacent samples), when $subST_{Mg/Ca vs. \Delta 47}$ maxima coincide with AMOC maxima and vice versa (Fig. 2b, c). This correlation is less well developed in the subsequent time interval of $\sim 38 - 20$ ka BP, which is partly due to the lower temporal resolution (~ 250 years between adjacent samples) and the suppressed variability of the $subST_{Mg/Ca vs. \Delta 47}$ record.”

line 364: please remind the reader of the periods which are compared in the following.

Response #40. Done (line 441).

line 404: what is happening during an Antarctic warming event, which role does it play in context of the study

Response #41. In the revised manuscript, we clarify that Antarctic warming events are associated with the restriction of polynya formation driven by coupled ice–ocean–atmosphere changes, a scenario that is oppositely correlated with Antarctic glacial stadials (lines 480-488). The revised text maintains the primary focus on processes during Antarctic cold stadials, which reinforce our main interpretations (i.e., focused on Antarctic cold stadials, Fig. 3).

“Due to the apparent gap in our knowledge about the nature of changes in the westerly winds, we can only speculate that the opposing effects on wind strength may have balanced each other out, thereby resulting in only relatively minor changes in westerly wind intensity. The majority of models suggest that the millennial-scale latitudinal shifts in the westerly winds resulted in a northward expansion of the belt of very cold and dry polar air, accompanied by strong katabatic winds and sea-ice propagation during glacial Antarctic cold stadials (Greenland interstadials) and, conversely, during warm interstadials known as Antarctic Isotope Maxima (AIM) (Greenland stadials, HS).”

Reviewer #2 (Remarks to the Author):

This paper presents a multiproxy analysis of seafloor sediment cores collected off Dronning Maud Land (DML) in the Weddell Sea to elucidate changes in water column structure and stratification in the Antarctic Ocean between 75,000 and 20,000 years ago. The results suggest that variations in the thermal structure between Antarctic Surface Water (ASW) and Warm Deep Water (WDW) may have driven upper-ocean stratification or the formation of polynyas. This is a significant finding that may offer valuable insights into the origin of the present-day Great Weddell Polynya and enhance our understanding of modern climate phenomena involving ice sheet–ocean–atmosphere interactions.

While the study presents novel and compelling results, several aspects require revision or further clarification. Most notably, the absence of Holocene data for some proxies is concerning. Although the paper focuses on the glacial period, the historical Great Weddell Polynya is a Holocene phenomenon. Therefore, it is essential to compare the glacial record with Holocene-scale oceanic polynya events. Additionally, the omission of available Holocene data may raise questions about the selective presentation of results and whether inconvenient data have been intentionally withheld.

Response #42. We appreciate the constructive comments from Reviewer #2 regarding the absence of Holocene data in our study. We would like to provide a comprehensive clarification of this important methodological consideration.

The reason is quite simple: The planktonic $\delta^{18}\text{O}$ data presented in our manuscript were used exclusively for stratigraphic correlation and age model development. Unfortunately, the Holocene sediment intervals in our core contain insufficient well-preserved foraminifera to conduct clumped isotope (Δ_{47}) or Mg/Ca analyses, which constitute the primary paleothermometry proxies interpreted in our study. In many sequences of the last 12 ka there were no foraminifera (planktic and benthic) at all. This limitation is likely due to low productivity of foraminiferal shells and their preservation, and in no way reflects selective data presentation or intentional withholding of inconvenient results. To illustrate that, previous paleoceanographic studies in the eastern Weddell Sea region have documented distinct preservation patterns of calcitic foraminiferal shells, with enhanced preservation during glacial periods and significantly reduced preservation during interglacial intervals, including the Holocene (Mackensen et al., 1989, 1996; Smith et al., 2010). Interestingly, this represents a very exceptional condition on the Antarctic continental margin and has been interpreted to reflect enhanced biological productivity in a polynya setting (Mackensen et al., 1996; Smith et al., 2010). Thus, the availability of biogenic carbonate strongly depends on glacial-interglacial conditions in our region.

Moreover, our research specifically targets glacial-period ocean dynamics (75,000-20,000 years BP) as a distinct climatic regime well-known to be worthy of independent investigation. The inclusion of Holocene data is not necessary for our scientific objectives for several reasons: (i) Glacial and Holocene periods operated under fundamentally different climatic boundary conditions, including ice sheet extent, atmospheric CO_2 concentrations, and orbital configurations. (ii) Our study addresses millennial-scale subsurface warming and polynya formation during glacial conditions, which represents a specific paleoceanographic phenomenon that does not require Holocene comparison for validation. (iii) Most importantly, the assumption that open polynya formation was a typical Holocene feature lacks empirical support. The well-documented Weddell Polynya of the 1970s represents a brief, exceptional event rather than a characteristic Holocene phenomenon. To our knowledge, no studies have demonstrated that such large-scale polynya formation was a persistent feature of Holocene Southern Ocean dynamics. (iv) Our glacial-period focus allows for detailed examination of millennial-scale variability and its relationship to broader climate oscillations (such as

Dansgaard-Oeschger events) without the complexity that would arise from incorporating a fundamentally different climatic regime.

We emphasize complete transparency regarding our data limitations. The new Figure 2 explicitly shows the absence of foraminiferal data during the Holocene interval, demonstrating that this represents a preservation constraint rather than selective data presentation. One important message conveyed by this figure is that the Holocene period significantly lacks foraminifera for further measurements, such as Mg/Ca and Δ_{47} .

We have included text passages in the introduction that clearly informs the reader that we show planktonic $\delta^{18}\text{O}$ data only for stratigraphic reasons.

See new added text in lines 100-103:

“The low number of well-preserved planktonic foraminifera in the deglacial and Holocene sediment samples only permitted a low-resolution $\delta^{18}\text{O}$ record to be generated but prevented the measurement of further Mg/Ca and clumped isotopes (Δ_{47}) proxies.”

See new added text in lines 138-140:

“All foraminifers from the interval 0-20 ka were used up to create an age model for core PS128_2, which unfortunately excluded the measurement of further calcitic proxies like Δ_{47} and Mg/Ca.”

See also the new figure 2, panel “a”. We have also included detailed discussion of these limitations in our Supplementary Information (see section “Stratigraphic correlation of sediment records and related measuring procedure”).

Therefore, the absence of Holocene data reflects both practical constraints (i.e., most important aspect) given the low availability of foraminifers in our specific regional settings as well as a scientifically justified research scope rather than any methodological shortcoming.

L 82-84: “we reconstruct past orbital- and millennial-scale variations in upper-ocean stratification from sediment core PS128_2, recovered from the Bungenstock Plateau off DML in the eastern Weddell Sea (Fig. 1).”

Comment: It should be clarified that this study utilizes composite records from three cores, not from PS128_2 alone. Furthermore, the Supplementary Information does not clearly indicate which core each proxy dataset was derived from. More fundamentally, the rationale for including core PS111_13-4—located approximately 90 km away—requires explanation.

Response #43. We agree with Reviewer #2 and acknowledge that we need to more clearly explain our composite record approach. We have provided comprehensive clarification on that

in the Supplementary Information as well as the rationale for our methodological approach. All foraminiferal proxy data ($\delta^{18}\text{O}$, $\delta^{13}\text{C}$, Mg/Ca, and Δ_{47}) come exclusively from core PS128_2-3. Core PS111_13-4, was included specifically to improve the stratigraphic framework at our site location through XRF titanium record correlation. This core provides essential Holocene ^{14}C age control that is unavailable at site PS128_2 (i.e., depth interval of 27-45 cm) due to insufficient foraminiferal preservation during the Holocene interval (see also Response #43).

See the added text in the revised version of our manuscript in lines 132-138.

“In order to better document the reliability and quality of our age model towards the end of the last glacial period, we also show the planktonic ^{14}C datings and $\delta^{18}\text{O}$ data (core PS128_2) for the last 20 ka, which document the deglacial climate transition into the Holocene. In addition, we used a core-to-core correlation to supplement the Holocene stratigraphy of core PS128_2 (upper 0.5 m) with ^{14}C datings from core PS111_13, which lies 90 km further north on the continental slope (Supplementary Information, Supplementary Fig. 1, Supplementary Tab. 1).”

See also the new text in the Supporting Information.

“Marine sediment cores PS128_2-2 (69.4115°S, 5.5897°W) and PS128_2-3 (69.4107°S, 5.5890°W) (less than 100 m apart from each other) ¹, were used to generate a composite record that doubles the amount of sediment material available for multiproxy analyses and merges proxy information from both cores. We correlated the magnetic susceptibility records and converted the depth scale of core PS128_2-2 to the sediment depth of core PS128_2-3 (Supplementary Fig. 1). Planktonic and benthic stable isotopes ($\delta^{18}\text{O}$, $\delta^{13}\text{C}$, Δ_{47}), planktonic Mg/Ca ratios, as well as planktonic ^{14}C were completely measured on PS128_2-3 for the time interval >18 ka. For the interval from 18 to 16 ka, all planktonic foraminifers were used up by ^{14}C datings, $\delta^{18}\text{O}$ and $\delta^{13}\text{C}$ measurements. For the past 16 ka, the number of well-preserved foraminifers was too low to perform ^{14}C , Mg/Ca and Δ_{47} measurements, despite a possible merging of identical sample sections from the two cores. The low numbers of foraminifers only allowed for $\delta^{18}\text{O}$ and $\delta^{13}\text{C}$ measurements.

The XRF scanning at AWI could only be performed on core PS128_2-2, as core PS128_2-3 was completely sampled on board. The core-to-core correlation was used to transfer the XRF scanning data from PS128_2-2 to the depth scale of PS128_2-3. In the following, we considered the merged data sets from PS128_2-2 and PS128_2-3 as information from site PS128_2.

In order to improve the stratigraphy at our site location, we correlated the XRF titanium record (ln Ti) of site PS128_2 to that from PS111_13-4 (Supplementary Fig.1). This allows the transfer of Holocene ^{14}C ages from PS111_13-4 to site PS128_2. The interval 158-266 cm from core PS111_13-4 corresponds to the depth interval of 27-45 at site PS128_2. Accordingly, the sedimentation rates is six times higher at PS111_13-4, as expected in core sites closer to the Antarctic margin.”

L 127: “Fig. 1” >> “Fig 2a”

Response #44. Done (line 153).

L 139-140: *“our reconstructed glacial subSTMg/Ca vs. $\Delta 47$ variability between ~70 and 20 ka BP (Fig. 2c)”*

Comment: An explanation is required for the absence of Mg/Ca ratio data during the Holocene.

Response #45. Yes, we agree with Reviewer #2 and we have deeply addressed this point as confirmed in previous Response #43 (and partially in Response #44).

L 141, L 141, L 144, L 150: *“(Fig. 2c)” >> “(Fig. 2d)”*

Response #46. Done. We have prepared a new figure (see Response #15). We prepared a new figure (see Response #15). All figure references throughout the manuscript have been systematically checked and revised to ensure consistency with the updated figure numbering scheme. The panel references have also been updated to correspond correctly with the revised figure layout.

L 167: *Caption of Figure 2h “diatom faunal”*

Comment: “faunal” is used for animals. Diatoms are plants, so “floral” is correct.

Response #47. We changed the figure legend according to Reviewer #1. In the revised version of the manuscript, we refer to more specifically the relative abundance of diatom group, which is what we show in the panel figure in %. See lines 217-218 and Response #17.

L 178: *“(Fig. 2c, e)” >> “(Fig. 2d, e, f)”*

Response #48. Done. In the revised version of the manuscript, we changed it to “Fig. 3c, d”, as we essentially deal with temperature and salinity proxies in the corresponding statement.

L 182-183: *“The $\delta^{13}\text{C}$ record of *N. pachyderma sin.* provides additional information on the signature of subsurface waters.”*

Comment: An explanation is needed for the absence of planktonic $\delta^{13}\text{C}$ data during the Holocene. Since Fig. 2a presents planktonic $\delta^{18}\text{O}$ data, it is presumed that $\delta^{13}\text{C}$ measurements were obtained simultaneously and should therefore be available.

Response #49. We would like to clarify the apparent discrepancy regarding planktonic $\delta^{13}\text{C}$ data availability during the Holocene. The Holocene $\delta^{18}\text{O}$ data serve as essential stratigraphic markers rather than interpretive proxies (see detailed Response #43). We highlight that our main interpreted proxies are: Mg/Ca and $\Delta 47$, which are not possible to be further measured in the Holocene. Thus, discussion/interpretation cannot be made in our study for the Holocene period.

Comment: The authors state that the $\delta^{18}\text{O}$ offset between planktonic and benthic foraminifera is stable; however, a detailed examination reveals several intervals where the two are in very close

agreement. If so, these periods should be explained as not representing deep convection that would imply the presence of Antarctic Bottom Water (AABW). Additionally, the absence of Holocene records for benthic foraminifera requires clarification. Moreover, carbon isotope ratios of benthic foraminifera, which are typically measured alongside oxygen isotopes, should be incorporated into the discussion of bottom water properties; if they are not included, the reasons for their omission should be provided.

Response #50. We thank Reviewer #2 for this detailed observation regarding the $\delta^{18}\text{O}$ offset between planktonic and benthic foraminifera. On intervals of close planktonic–benthic $\delta^{18}\text{O}$ agreement and implications for deep convection/AABW. Our data show a small, consistently positive benthic–planktonic $\delta^{18}\text{O}$ offset that averages $0.3 \pm 0.1\text{‰}$ across the glacial interval. While there are intervals where the two curves come into close agreement, the absolute offset magnitude is comparable to the combined analytical and species-correction uncertainties, which limits firm interpretation of mixing depth. In short, intervals of close agreement do not, in our view, securely indicate absence of deep convection; rather, the offset is too small relative to uncertainties to serve as a robust diagnostic of convective penetration or AABW formation at this site.

We have incorporated discussion of these implications in the past regional context and any further assessment might be too speculative (lines 292-298 and 463-470). See also Response #10.

See text in lines 292-298:

“The averaged benthic-planktonic $\delta^{18}\text{O}$ offset of $0.3 \pm 0.1\text{‰}$ suggests that convective mixing and cooling of the deep ocean probably did not penetrate to 2,000 m depth, at least at our site. However, the offset value is low and still within a certain error range resulting from the analytical precision for benthic and planktonic $\delta^{18}\text{O}$ measurements (which add up to 0.12‰) and an uncertainty in species-specific benthic $\delta^{18}\text{O}$ correction ($\sim 0.1\text{‰}$). This indicates that the benthic-planktonic $\delta^{18}\text{O}$ offset cannot be reliably interpreted.” (lines 292-298).

See text in lines 463-470:

“The large-scale Weddell polynya is recognized for producing substantial volumes of AABW ¹⁶, and thus potentially influencing the global overturning circulation. At our study site, however, the small and relatively stable benthic-planktonic $\delta^{18}\text{O}$ offset is masked by a relatively large uncertainty range, which unfortunately does not allow a reliable interpretation on the mixing depth and the formation of AABW (Fig. 4a). Future research should incorporate additional proxy reconstructions to better assess the role of AABW formation in the context of the glacial Weddell polynya and the deep convective processes between Maud Rise and Bungenstock Plateau.”

Since the focus of our study centers on upper ocean conditions and polynya formation processes, we do not present benthic $\delta^{13}\text{C}$ data for the following reasons:

Our primary objectives focus on subsurface warming and upper ocean stratification, which are better constrained by our temperature proxies (Mg/Ca, Δ_{47}) and planktonic isotopic records. Our benthic $\delta^{18}\text{O}$ data serve primarily as stratigraphic markers rather than interpretive proxies.

Importantly, one critical insight from our study comes from separating the classical temperature and salinity effects on foraminiferal $\delta^{18}\text{O}$ signals using temperature proxies, without which we would not be able to recognize polynya formation. Here, our current study establishes the foundation for understanding upper ocean dynamics in the last glacial period, providing essential context for future investigations of deep water formation processes.

L 239-275 and L 292-311: About mechanism of open-ocean polynya

Comment: Recently, Narayanan et al. (2024, Science Advances, 10, eadj0777) discussed that the formation of open-ocean polynyas in 2016 and 2017 was linked to salt export driven by Ekman transport under strong easterly winds. Their study also explores the influence of the Southern Annular Mode (SAM). It is recommended that the authors address the consistency or divergence between the present interpretations and those of Narayanan et al. (2024), and incorporate this comparison into the discussion.

Response #51. We thank Reviewer #2 for this valuable recommendation to incorporate the recent findings of Narayanan et al. (2024). We have included this important work in the revised version of our manuscript and addressed the consistency between their interpretations and ours. We have incorporated their key findings regarding salt export driven by Ekman transport under strong easterly winds into our discussion sections.

See the added phrase in lines 333-335:

“In addition to that, an eastward, wind-induced Ekman transport of saline waters onto the northern flank of Maud Rise may have contributed to an increased salt supply, which more than compensated for the freshening sea ice melt ⁶⁸.”

See the added phrases in lines 379-384.

“Exceptionally, the recent Weddell Polynya formation in 2017 occurred in the context of “+SAM” . However, the polynya was smaller and restricted to the Maud Rise region but was also characterized by weakly-stratified upper ocean conditions ⁶⁸. It has been discussed that the polynya formation in 2017 was driven by an additional salt input related to intensified eastward surface wind stress in combination with a spin up in Weddell Gyre’s barotropic circulation ⁶⁷.”

The Narayanan et al. (2024) interpretation is not contradictory to our findings but rather provides complementary mechanistic insights. Both studies emphasize the critical role of reduced upper ocean stratification in enabling polynya formation. This convergence strengthens understanding that stratification control represents the fundamental driver of large-scale polynya formation in the Weddell Sea across different temporal scales and climatic conditions.

Reviewer #3 (Remarks to the Author):

I read the manuscript with interest and appreciate the effort of the authors. Unfortunately, I cannot recommend publication in Nature Comms. In general, I found that interpretations were not always supported by the data, and/or data were interpreted a certain way with little or no explanation as to why or exploration of alternative interpretations. As such, I suggest a longer-format paper, where some of the nuances of the data (some of which is found in the supplement) can be discussed and addressed in the text. Finally, I recommend a bit of caution when discussing SAM and other modern teleconnections during glacial periods; a safer bet is simply referring to westerly wind variations.

Response #52. We thank Reviewer #3 for the evaluation of our manuscript. While we appreciate her/his concerns, we respectfully disagree with several aspects of her/his assessment and would like to provide a comprehensive response addressing the main points.

We strongly disagree with the assertion that our interpretations are not supported by the data. Our study presents multiple lines of independent evidence that converge on consistent paleoceanographic interpretations. We demonstrate strong correlations between independent temperature proxies (Mg/Ca and Δ_{47}), and stable isotope records ($\delta^{18}\text{O}$, $\delta^{13}\text{C}$), providing robust internal consistency checks. Our proxy records show mechanistically consistent responses to known paleoceanographic processes. We provide quantitative assessments of proxy relationships and temporal correlations that support our interpretations with appropriate statistical rigor.

Our study focuses on paleoceanographic processes rather than technical proxy development, making it well-suited for Nature Communications' interdisciplinary scope. We believe that moving extensive technical details from the Supplementary Information to the main text would: (i) Obscure the primary paleoceanographic findings for the broader readership; (ii) create an unnecessarily technical manuscript that loses focus on the key scientific contributions; and (iii)

reduce accessibility to the interdisciplinary audience that Nature Communications serves as also emphasized by Reviewer #1.

The most important technical points for a paleoceanographic study are mentioned in the revised version of our manuscript. The Supplementary Information provides comprehensive technical details for specialized readers while maintaining the main text's focus on paleoceanographic insights and broader climate implications.

Our work represents several important firsts in Antarctic paleoceanography. (i) First high-resolution foraminiferal records from the Antarctic margin spanning the last glacial period with sufficient temporal resolution to resolve millennial-scale variability. (ii) First application of clumped isotope thermometry (Δ_{47}) to Antarctic paleoceanography during the last glacial period, providing independent temperature constraints. (iii) First integration of ^{14}C dating, Mg/Ca, Δ_{47} , and stable isotopes in a single high-resolution record. Unique foraminiferal preservation during glacial periods enables proxy applications typically impossible in Antarctic margin sediments. These contributions provide fundamental insights into ice-ocean-atmosphere interactions during glacial periods and offer crucial constraints for understanding modern polynya formation mechanisms.

Regarding the caution about discussing SAM and modern teleconnections during glacial periods, we acknowledge this concern but maintain that our approach is scientifically justified. Our references to "SAM-like conditions" focus on fundamental atmospheric circulation patterns rather than assuming identical teleconnection structures. The underlying physics of wind-driven ocean circulation, precipitation-evaporation balance, and ice-ocean interactions remain greatly consistent across different climate states. The paleoclimate community routinely uses modern circulation analogies (NAO-like, ENSO-like, SAM-like, or Summer-like) to describe past climate patterns while acknowledging differences in boundary conditions. We explicitly acknowledge that glacial boundary conditions differ from modern conditions and use these analogies as conceptual frameworks rather than direct equivalents.

We are confident that our work fits Nature Communications' scope for several reasons. Our findings have implications for oceanography, glaciology, climate dynamics, and paleoclimatology. The results provide insights into modern climate phenomena (Great Weddell Polynya) and ice sheet-ocean interactions relevant to future climate projections. Novel application of clumped isotope thermometry to Antarctic paleoceanography represents a methodological advance with broad applications. The manuscript length is suitable for comprehensive presentation of multiproxy results without excessive technical detail. We believe

that multiproxy paleoceanographic studies should be encouraged to include technical information in the Supplementary Information, rather than redirecting the study toward a new specialized focus.

We believe our manuscript makes significant contributions to understanding Antarctic ocean dynamics and ice-climate interactions that warrant publication in Nature Communications rather than a more specialized venue as suggested by Reviewer #3.

As a general comment, the number of proofreading errors throughout the manuscript was surprising. I observed many typos, inconsistent referencing (missing citations; some references were numerical, some by name where not appropriate, etc.), and referencing errors (check all publication dates and spellings of authors names) in both the MS and the supplement. I do not have time to highlight or correct them, but care should be taken to ensure they are corrected in any future version.

Response #53. We thank Reviewer #3 for bringing attention to the proofreading and referencing issues throughout the manuscript. We acknowledge that these errors detract from the scientific content and apologize for these oversights. We conducted a systematic review and correction of the entire manuscript and Supplementary Information. This issue has been resolved.

A few things about the data:

1. The age model seems ok, but something should be said about the age reversals, the actual error vs analytical error, and some discussion of the sedimentation rate change at 19 ka is... required.

Response #54. We respectfully disagree with the characterization of our age model as merely "ok." Our chronological framework represents one of the most robust radiocarbon-based age models available for the Antarctic continental margin spanning the glacial period. The exceptional preservation of foraminifera in our study region has enabled us to obtain an unprecedented density of ¹⁴C dates, which is particularly remarkable given the well-documented scarcity of suitable carbonate material in Antarctic marine sediment cores from this time period (i.e., last glacial period). We were very surprised that this was not commented upon by Reviewer #3.

We clarify that there are no age reversals present in our glacial-period record. Concerned inconsistencies may reflect misinterpretation of our data presentation. However, our Holocene ages do show some age reversals. Some of these age reversals are within the 1-sigma error range of the ¹⁴C data and thus do statistically not represent a reversal (especially when considering the 2-sigma range). These Holocene ages originate from core PS111_13 (upper continental slope) and were transferred to site PS128_2 via correlation (XRF data) (Fig. 2; Supplementary Information). The sedimentation rate there is 6x higher and the influence of bioturbation is also more pronounced than at the position of core PS128_2 on the Bungenstock Plateau. Presumably, bioturbation contributed to the rather weakly pronounced age reversals. At core PS111_13 the planktic foraminifera were consumed for age dating. This prioritization meant that no more foraminifera were available for further Holocene analysis.

The comprehensive discussion of actual versus analytical errors has been thoroughly addressed in Mollenhauer et al. (2021), which provides the methodological foundation for our chronological approach but it is out of the scope of our work. We acknowledge that an error in our initial sedimentation rate calculation at 19 ka was identified and corrected. Importantly, this correction did not affect our final age model or any of the interpretations presented in the manuscript.

2. I was surprised not to see any reference to N. pachy habitat at the Antarctic Margin, where this core is located. It is well known (see Hendry et al., 2009) that N. pachy lives in brine channels in the sea ice, and this can impact all manners of geochemical results. That the benthic and planktic isotopes appear similar is maybe support for them living in the water column, maybe. Or, it may reflect another problem.

Response #55. We appreciate the reviewer's concern regarding *N. pachyderma* habitat preferences and we now cite further studies on this from the Southern Ocean; however, we believe this comment reflects an incomplete understanding of the species' ecological distribution in our study region setting and unfortunately overlooks key evidence presented in our manuscript (Supplementary Information; Supplementary Fig. 4).

Our manuscript includes detailed calcification depth estimates demonstrating that *N. pachyderma* in our study region calcifies at 115-120 m water depth (Supplementary Information; Supplementary Fig. 4), indicating substantial populations residing in the underlying water column rather than being restricted to sea ice environments. This depth range places the species well below the immediate influence of sea ice processes.

Recent comprehensive study (Greco et al., 2019) demonstrates that *N. pachyderma* exhibits peak concentrations just below the chlorophyll maximum, with vertical and spatial distributions

primarily controlled by surface properties such as sea-ice concentration and chlorophyll availability. Importantly, these studies show that the species is not obligatorily associated with sea ice or brine channels, contrary to the reviewer's assertion. We use Greco et al. (2019) because it best reflects the ecological behavior of *N. pachyderma* under conditions most comparable to polynya settings.

We have added a new text that concisely explains why our foraminiferal geochemical proxies are not influenced by sea ice (see lines 258-263):

*“Although the Antarctic *N. pachyderma* genotype may be adapted to sea ice conditions⁵⁵, strong agreement between the independent temperature proxies Δ_{47} and Mg/Ca (Supplementary Fig. 8), corroborated by *N. pachyderma* sin. $\delta^{13}\text{C}$ (Fig. 3c, f), and supported by our apparent calcification depth estimate (Supplementary Fig. 4), provides evidence that our foraminiferal geochemical proxies are not affected by sea ice (Refs.^{22,56}).”*

The similarity between benthic and planktonic $\delta^{18}\text{O}$ values in our record is not problematic, as suggested by Reviewer #3, but rather consistent with established regional patterns. Previous studies from nearby cores in the eastern Weddell Sea (PS1388, Mackensen et al., 1989; PS1506, Mackensen et al., 1994) have documented similar benthic-planktonic isotope relationships, providing strong regional context for our observations. This isotopic similarity supports, rather than questions, the water column habitat of our planktonic species.

While we acknowledge that sea ice-associated calcification can influence geochemical signatures, our multiple lines of evidence, including calcification depth estimates, regional isotopic patterns, cross-correlation between independent proxies and ecological distribution data collectively demonstrate that *N. pachyderma* in our study area primarily represents water column conditions rather than sea ice-influenced environments.

*3. Following up on the above, why are the benthic and planktic stable isotope values so similar (and so positive at the LGM?). Also, how do the authors explain the reversal in benthic and planktic isotope gradient at 275 cm? I also do not see the significant variability on millennial timescales, as stated by the authors on line 126. How do the authors define significant? How is $\delta^{13}\text{C}$ of planktics (and *N. pachy* in the polar regions specifically) usually interpreted? Is it usually interpreted? As an aside, I was surprised to see NBS-19 still being used as a standard, as it is no longer available and has not been for years.*

Response #56. As supported by regional studies (Mackensen et al., 1989, 1994), similar benthic and planktonic $\delta^{18}\text{O}$ values are characteristic of the eastern Weddell Sea region. The reduced offset between these records can be easily explained by deep convection processes reaching

depths of ~2,000 m during polynya formation events, homogenizing the water column isotopic signature. It is important to note that our benthic $\delta^{18}\text{O}$ record serves primarily as a stratigraphic tool rather than a paleoceanographic proxy (see previous Responses #10 and #51).

The reviewer correctly observes limited variability in the planktonic $\delta^{18}\text{O}$ record itself. However, this reflects the well-established principle that temperature and salinity effects exert opposing influences on calcite $\delta^{18}\text{O}$ signals. Our study's primary contribution lies in deconvolving these competing effects through our Δ_{47} and Mg/Ca analyses, which reveal clear and indeed significant millennial-scale variability in both temperature and salinity (the main interpreted proxies in our manuscript).

We did not claim that *N. pachyderma* $\delta^{13}\text{C}$ values in polar regions have a standard interpretation. Rather, we provided context for how these values might be interpreted in our specific setting (please see lines 234-241). The $\delta^{13}\text{C}$ data serve as supporting evidence when integrated with our primary temperature and salinity proxies, rather than as standalone paleoceanographic indicators, see for example Hillenbrand et al. (2017) (this reference is mentioned by Reviewer #3 in the following comment below). This type of approach is common practice in paleoceanography. This may give the impression that Reviewer #3 may have overlooked certain details in our manuscript.

Regarding the NBS-19 standard, we maintain reference materials from the original NBS-19 stock for calibration purposes, supplemented by IAEA-603. While NBS-19 is no longer commercially available, laboratories with existing stocks continue to use it for consistency with historical datasets. This is broadly understood in the field. Our analytical protocols ensure measurement reproducibility and accuracy through rigorous quality control procedures. The suggestion that this approach compromises data quality is unfounded and reflects a misunderstanding of standard laboratory practices in stable isotope geochemistry. This shows that Reviewer 3 has made comments that cast doubt on our measurements, which we find to be highly inappropriate.

4. I notice that the authors state that theirs is the most southern Mg/Ca dataset ever published, but Hillenbrand et al (2017), Rathburn (1999), Mawbey et al. (2020), and others have published data from the Antarctic margin farther south than this site. Were any corrections applied to the Mg/Ca data (e.g., Evans, Gray)? This seems particularly important given the salinity interpretations. The discussion of the choice of temperature calibration should also be clearer. What is the error of the calibration chosen? Why might the coldest temperatures occur

between 40 and 45 Ka? How might the different morphotypes of N. Pachy (s) seen in the SEM photos impact Mg/Ca data? All things to think about and address.

Response #57. Our statement regarding the southernmost Mg/Ca dataset specifically refers to glacial-period records around Antarctica. The studies cited by the reviewer focus primarily on modern calibration studies and Holocene conditions. Our dataset represents the first high-resolution Mg/Ca and Δ_{47} records from this latitude spanning the last glacial period, providing an exceptional temporal framework with unprecedented foraminiferal preservation and dating density.

We have dedicated substantial discussion in our supplementary material to our calibration approach, which employs Δ_{47} as an independent temperature proxy to validate our Mg/Ca interpretations (we showed descriptive Supplementary Tab. 1 and Supplementary Figs. 5 and 8). Our calibration temperature equation reveals a very similar calibration slope in comparison to Kozdon et al. (2009) (Supplementary Fig. 8). The consistency in temperature amplitudes validates the thermal sensitivity of our Mg/Ca record. We provided comprehensive comparisons with available species-specific calibration equations for the Southern Ocean. Our analysis demonstrates that existing regional calibrations produce unrealistically warm temperatures (see for reference Morley et al. 2024; Nat. Comm), while our Δ_{47} -validated approach yields more realistic results consistent with regional oceanographic conditions. This again gives the impression that Reviewer #3 may have overlooked aspects of our work.

Reviewer #3 questions the coldest temperatures between 45-40 ka. The temperature minimum between 45-40 ka, questioned by the Reviewer #3, is clearly explained in our manuscript and illustrated in Figure 3c (former Fig. 2d). This cooling reflects the combined influence of orbital forcing (particularly obliquity) and millennial-scale variability. The long-term temperature trend shows clear orbital control, with additional millennial-scale oscillations superimposed. Similarly, strong millennial-scale cooling, is evident between 65-60 ka, demonstrating the consistent relationship between our temperature record and known climate forcing mechanisms.

Upon this point, Reviewer #3 offered minimal critique of the paleoceanographic mechanisms and interpretations, which casts doubt on her/his claim that our data do not support the interpretation.

Regarding *N. pachyderma* morphotype variations, recent studies (Mikis et al., 2019; Prabhakar et al., 2024) demonstrate that geochemical signatures are not significantly altered by morphotype variation, supporting the robust applicability of this species as a paleoenvironmental proxy in Antarctic waters. Our SEM imaging (in combination with the weights of *N. pachyderma*) primarily serves to document the exceptional preservation state of

our foraminiferal specimens, which is crucial for reliable Δ_{47} and Mg/Ca interpretations, as dissolution is widely known to introduce significant artifacts in temperature reconstructions. We emphasize that our study's primary contribution lies in providing the first high-resolution, multi-proxy paleoceanographic record spanning the glacial period from this critical Antarctic margin location, enabling unprecedented insights into millennial-scale ocean-climate dynamics in the Southern Ocean.

5. Temperatures for clumped isotopes and Mg/Ca below the freezing point of seawater suggest an issue (see habitat issue above and correction comment). This needs to be addressed and discussed in the text.

Response #58. This observation reflects the inherent analytical challenges of applying paleothermometry in extreme polar environments and has been appropriately addressed in our methodology:

Sub-freezing temperature estimates are a well-documented phenomenon when applying clumped isotope thermometry near the physical limits of seawater temperature. Given that our study site experiences calcification temperatures at or near the minimum possible seawater temperature ($\sim -1.8^\circ\text{C}$), analytical uncertainties in Δ_{47} measurements can occasionally yield nominally impossible values in individual measurements.

Our analytical approach follows established protocols for handling boundary conditions in paleothermometry. The Gaussian distribution of replicate measurements means that approximately 16% of individual analyses may fall outside the 68% confidence interval, including below the freezing point. This statistical behavior is expected and does not invalidate the overall temperature reconstruction.

As detailed in our supplementary material, we have implemented a "hard boundary" correction for only the two instances where mean Δ_{47} -derived temperatures fell below -1.8°C . These values were adjusted to -2°C (the practical freezing point of seawater) to maintain physical realism while preserving the relative temperature variations that are central to our paleoclimatic interpretations. Our Mg/Ca temperatures are calibrated against the Δ_{47} -derived temperatures, ensuring internal consistency within our dataset. The boundary correction applied to Δ_{47} values maintains the integrity of this calibration relationship while addressing the physical constraints of seawater temperature.

These near-freezing temperatures are consistent with the extreme glacial conditions expected in the eastern Weddell Sea during the last glacial period and support our interpretation of a highly stratified, cold surface ocean with episodic polynya-driven warming events.

References

- Greco, M., Jonkers, L., Kretschmer, K., Bijma, J., & Kucera, M. (2019). Depth habitat of the planktonic foraminifera *Neogloboquadrina pachyderma* in the northern high latitudes explained by sea-ice and chlorophyll concentrations. *Biogeosciences*, *16*(17), 3425–3437. <https://doi.org/10.5194/bg-16-3425-2019>
- Haumann, F. A., Gruber, N., & Münnich, M. (2020). Sea-Ice Induced Southern Ocean Subsurface Warming and Surface Cooling in a Warming Climate. *AGU Advances*, *1*(2). <https://doi.org/10.1029/2019AV000132>
- Hillenbrand, C.-D., Smith, J. A., Hodell, D. A., Greaves, M., Poole, C. R., Kender, S., et al. (2017). West Antarctic Ice Sheet retreat driven by Holocene warm water incursions. *Nature*, *547*(7661), 43–48. <https://doi.org/10.1038/nature22995>
- Kozdon, R., Eisenhauer, A., Weinelt, M., Meland, M. Y., & Nürnberg, D. (2009). Reassessing Mg/Ca temperature calibrations of *Neogloboquadrina pachyderma* (sinistral) using paired $\delta^{44/40}\text{Ca}$ and Mg/Ca measurements. *Geochemistry, Geophysics, Geosystems*, *10*(3). <https://doi.org/10.1029/2008GC002169>
- Mackensen, A., Grobe, H., Hubberten, H.-W., Spiess, V., & Fütterer, D. (1989). Stable isotope stratigraphy from the Antarctic continental margin during the last one million years. *Marine Geology*, *87*(2–4), 315–321. [https://doi.org/10.1016/0025-3227\(89\)90068-6](https://doi.org/10.1016/0025-3227(89)90068-6)
- Mackensen, A., Hubberten, H. -W., Scheele, N., & Schlitzer, R. (1996). Decoupling of $\delta^{13}\text{C}_{\text{CO}_2}$ and phosphate in recent Weddell Sea deep and bottom water: Implications for glacial Southern Ocean paleoceanography. *Paleoceanography*, *11*(2), 203–215. <https://doi.org/10.1029/95PA03840>
- Mikis, A., Hendry, K. R., Pike, J., Schmidt, D. N., Edgar, K. M., Peck, V., et al. (2019). Temporal variability in foraminiferal morphology and geochemistry at the West Antarctic Peninsula: a sediment trap study. *Biogeosciences*, *16*(16), 3267–3282. <https://doi.org/10.5194/bg-16-3267-2019>
- Mollenhauer, G., Grotheer, H., Gentz, T., Bonk, E., & Hefter, J. (2021). Standard operation procedures and performance of the MICADAS radiocarbon laboratory at Alfred Wegener Institute (AWI), Germany. *Nuclear Instruments and Methods in Physics Research Section B: Beam Interactions with Materials and Atoms*, *496*, 45–51. <https://doi.org/10.1016/j.nimb.2021.03.016>
- Morley, A., de la Vega, E., Raitzsch, M., Bijma, J., Ninnemann, U., Foster, G. L., et al. (2024). A solution for constraining past marine Polar Amplification. *Nature Communications*, *15*(1), 9002. <https://doi.org/10.1038/s41467-024-53424-w>
- Narayanan, A., Roquet, F., Gille, S. T., Gülk, B., Mazloff, M. R., Silvano, A., & Naveira Garabato, A. C. (2024). Ekman-driven salt transport as a key mechanism for open-ocean polynya formation at Maud Rise. *Science Advances*, *10*(18). <https://doi.org/10.1126/sciadv.adj0777>

- Prabhakar, M., Thirumalai, K., Cronin, T. M., Gemery, L., Thomas, E. K., & Rafter, P. A. (2024). Morphotypical and Geochemical Variations of Planktic Foraminiferal Species in Siberian and Central Arctic Ocean Core Tops. *Journal of Foraminiferal Research*, 54(1), 1–19. <https://doi.org/10.61551/gsjfr.54.1.1>
- Smith, J. A., Hillenbrand, C.-D., Pudsey, C. J., Allen, C. S., & Graham, A. G. C. (2010). The presence of polynyas in the Weddell Sea during the Last Glacial Period with implications for the reconstruction of sea-ice limits and ice sheet history. *Earth and Planetary Science Letters*, 296(3–4), 287–298. <https://doi.org/10.1016/j.epsl.2010.05.008>

Dear Reviewers,

We thank Reviewers for their valuable comments/suggestions, which we have thoroughly analyzed and implemented. They are very constructive and appropriate. We believe that through their implementation the manuscript has greatly improved. We provide Word document files with tracked changes of the revised version of the manuscript. To simplify the evaluation of the performed modifications we (i) copied below in black/italic the comments/suggestions from Reviewers, and (ii) provided a detailed response to each comment/suggestion in blue/not-italic. All line numbers referenced in our responses correspond to the clean revised version. All edits can be found in the tracked-changes version of the manuscript.

REVIEWER COMMENTS

Reviewer #1 (Remarks to the Author):

Review of the revised submission "Millennial-scale subsurface ocean warming in the eastern Weddell Sea during the last glacial period" by Pinho et al.

I thank the authors for revising their manuscript, especially describing paleo terminology better, improving the visual representation of the results and efforts taken to improve the manuscript.

Yet, I am not completely convinced by some statements made in the manuscript and would still appreciate some more guidance through some parts of the manuscript. Therefore I would like some more revisions as pointed out by the comments below.

Response #1. We thank the Reviewer #1 for the constructive assessment of our revision. We appreciate the additional guidance requested and further refined the manuscript accordingly.

Major comments:

Modern day polynya classifications

-In modern day physical oceanography, there is a distinction between the Weddell Sea Polynya in the 1970s and Maud Rise Polynyas in 2016 and 2017, as they are resulting from different processes and also have impacts on different scales.

-line 63: Several studies have shown that the formation of the Weddell Sea Polynya / Great

Weddell Polynya was triggered by several processes and not only a density instability. Please rephrase the text.

Response #2. We thank Reviewer #1 for highlighting the modern distinction between the Great Weddell Polynya of the 1970s and the Maud Rise polynyas in 2016–2017. In the revised manuscript, we consistently differentiate the Great Weddell Polynya and Maud Rise polynyas. We also clarify that the Great Weddell Polynya was initiated by multiple processes rather than a single density instability, and we have updated lines 64–68 accordingly (see revised text below).

“During the winters of the mid-1970s, a combination of processes, including density-driven instabilities via interactions with topographic features such as Maud Rise ^{16,17}, atmospheric forcing ^{18,19} and atmosphere-ice-ocean dynamics ²⁰ led to the breakdown of the stratification and the formation of the Great Weddell Polynya (maximum extent of 350,000 km², Fig. 1a) ²¹” (lines 64-68).

- the studies used in lines 332-334 are referring to Maud Rise Polynyas, while the text refers to Weddell Sea Polynyas. Please adjust the text accordingly.

Response #3. Yes. We thank Reviewer #1 for noting that the studies cited relate to Maud Rise polynyas. We have revised the text to explicitly attribute those processes to Maud Rise polynyas and to clarify their relationship as a potential precursor for the historic Great Weddell Polynya (e.g., Cheon and Gordon, 2019; Dufour et al., 2017; Gordon, 1978; Kurtakoti et al., 2018; Martinson et al., 1981).

“These processes are related to the Maud Rise Polynya, which is considered a precursor for the development of the historic Great Weddell Polynya (e.g., Refs. ^{16,17}.” (lines 347-349).

-line 377-382: “Weddell Polynya” - the polynya in 2017 was a Maud Rise Polynya and the studies used as reference are focusing on Maud Rise Polynyas. Please correct the text accordingly.

Response #4. We thank Reviewer #1 for observing this mismatch between text and citation. We have revised the text passage to explicitly refer to the Maud Rise Polynya. We also made slight changes in the following phrase to clearly distinguish the Great Weddell Polynya and Maud Rise Polynya.

“In contrast, the most recent Maud Rise Polynya formation in 2016 and 2017 occurred in the context of “+SAM” ⁸². The polynya was smaller in comparison to the Great Weddell Polynya in the 1970s and restricted to the Maud Rise region but was also characterized by weakly-stratified upper ocean conditions ^{74,82}.” (lines 409-413).

Section “Glacial occurrence of the Great Weddell Polynya”

-The statement in lines 285-287 is very vague and raises the question how the authors define a “pronounced maxima”? If one compares the maxima in Temperature given in Figure 3c, The maxima around 47 ka is larger than in 44 ka, and of the same salinity, so why does one of these periods indicate a rise of WDW and the other not?

Response #5. We appreciate the reviewer’s point. Our intent was to describe intervals of increased subsurface temperature and salinity rather than “pronounced maxima.” We have revised the wording to avoid ambiguity and explicitly state the diagnostic used to infer the presence of relatively warm deep water (WDW) as shown below (lines 290-293). The revised text in line 241 also illustrate that. The terms “maxima” and “minima” are now reserved for long-term variations paced by obliquity forcing.

“Instead, pronounced increases in subsurface temperature and salinity combined with decreases in $\delta^{13}\text{C}_{\text{DIC}}$ (Fig. 3c, d, f) indicate times when the influence of WDW was closest to the ocean surface.” (lines 293-296).

-the lines 292-298, are very hard to follow and raise some questions, e.g.,: is the mentioned $\delta^{18}\text{O}$ a general one or one of the previously introduced ones? Please make this more understandable

Response #6. We agree that the passage was unclear and that the $\delta^{18}\text{O}$ reference needed to be explicit. The $\delta^{18}\text{O}$ values discussed refer to our new measurements in Figure 2a from core PS128_2. We have revised the text for clarity and to better convey the uncertainty (lines 298-308).

“To evaluate whether deep convection occurred, we compared our $\delta^{18}\text{O}$ values of benthic and planktonic foraminifera from core PS128_2 (Fig. 2a). The average benthic-planktonic $\delta^{18}\text{O}$ difference is $0.3 \pm 0.1\text{‰}$. This small offset suggests that convective mixing and cooling of the deep ocean possibly did not penetrate to 2,000 m depth, at least at our site. However, this interpretation is uncertain because the observed offset is close to the combined analytical and correction uncertainties. The measurement precisions for benthic and planktonic $\delta^{18}\text{O}$ together amount to $\pm 0.12\text{‰}$, and the uncertainty in the species-specific correction for benthic $\delta^{18}\text{O}$ adds $\sim 0.1\text{‰}$. Given that these errors are similar in magnitude to the observed offset, the benthic–planktonic $\delta^{18}\text{O}$ gradient cannot be interpreted with confidence. Accordingly, we refrain the use of benthic and planktonic $\delta^{18}\text{O}$ to stratigraphic purposes only (Fig. 2a).” (lines 302-312).

Section “Glacial Forcing and feedback mechanisms”

-In the paragraph about the AMOC (lines 434-450) the correlation between subST and Pa/Th is

not obvious in the Figures 3b and c. Right now this paragraph feels vague and would benefit from more support.

Response #7. We thank Reviewer #1 for this helpful comment. We agree that Pa/Th and subST are not perfectly correlated, in part because both exhibit short-term variability and our sampling resolution varies through time. To better support the AMOC linkage, we revised the paragraph to (i) focus explicitly on the time intervals shaded in blue in Fig. 3, (ii) make the sign and timing of the relationships explicit, and (iii) acknowledge resolution limits and proxy uncertainties. The revised text reads (lines 484-501):

“On millennial timescales, the variability of $subST_{Mg/Ca \text{ vs. } \Delta 47}$ appears to be related to changes in the Atlantic Meridional Overturning Circulation (AMOC) ^{53–56} (Fig. 3b). In our record, millennial-scale warming in $subST_{Mg/Ca \text{ vs. } \Delta 47}$ off DML parallels decreases in $^{231}Pa/^{230}Th$ from Bermuda Rise in the North Atlantic, indicating AMOC strengthening (Fig. 3b, c). Cooling intervals coincide with increases in $^{231}Pa/^{230}Th$ during Heinrich Stadials (HS), which are cold glacial stadial events in Greenland related to phases of AMOC weakening ^{53–56}. This relationship is particularly clear from ~75 – 38 ka BP, when sampling resolution is higher (~125 years between adjacent samples) (Fig. 2b). From ~38 – 20 ka BP, the correlation weakens, likely due to the two-fold lower temporal resolution (~250 years) (Fig. 2b), which is reflected in the suppressed millennial-scale variability of the $subST_{Mg/Ca \text{ vs. } \Delta 47}$ record. When comparing long-term variability, this period is characterized by a relative strong and more stable AMOC compared to the previous period (~75–38 ka BP) due to the absence of pronounced short-term minima, with the exception of the extreme HS2 minimum ⁵⁵ (Fig. 3b). The amplitudes of the minima HS3 (at ~31 ka BP) and HS2 (~24 ka BP) are controversially discussed and have been suggested to be much lower ⁹² than inferred from the $^{231}Pa/^{230}Th$ proxy (Fig. 3b). If so, this would further support a predominantly strong AMOC with low-variability during ~38–20 ka BP, coinciding with a long-term maximum in $subST_{Mg/Ca \text{ vs. } \Delta 47}$.” (lines 484-501).

-please make clear in the text and not only the caption that the Pa/Th proxy is from a different core

Response #8. Done. See the previous Response #7 and the corresponding text in lines 484-501.

Conclusions:

-I would recommend the authors to be more careful about the statement that the Great Weddell Polynya may represent a remnant of the last Ice Age, as formation processes are very different (glacial polynya: hybrid formation of coastal and open-ocean processes vs. only open ocean processes)

Response #9. Yes, we agree with Reviewer #1 that our conclusions should not imply that the Great Weddell Polynya is a direct remnant of glacial polynya states. We have revised the conclusions to clearly distinguish process regimes and to frame our inference in terms of analogs

for boundary conditions and feedbacks rather than direct equivalence (see text below, lines 556-567).

“Our proxy data from site PS128-2 on the Bungenstock Plateau indicate that the recurring presence of the Glacial DML Polynya was a typical feature of the last ice age. Though the Glacial DML Polynya may share some similarities with the openings of the Great Weddell Polynya in the 1970s and the Maud Rise Polynya in 2016 and 2017 they are the result of different processes and impacts on different timescales. While the formation of the Glacial DML Polynya is consistent with a hybrid polynya mode that combines characteristics of a coastal polynya and an open ocean polynya, the offshore events of the last century are primarily open-ocean driven. We therefore propose that the modern occurrences are not a direct remnant of the last ice age, but rather an expression of a regionally persistent sensitivity to atmosphere-ice-ocean interactions, which was strongly expressed during the last glacial when polynya-deep-convection operated on millennial timescales.” (lines 556-567).

In addition, the revised manuscript clarifies that the inferred glacial polynya represents a hybrid coastal–open-ocean mode, involving both latent- and sensible-heat processes, and therefore likely exhibits a dominant process distinct from that of the Great Weddell Polynya (see lines 24-26 in the abstract, lines 435-437, 468-469).

-The last statement in lines 521-523 would benefit from a citation.

Response #10. Done (lines 578-581).

Figure 2

-The figure has been added, which is a nice addition, yet Figure 2b is not at all referenced in the text, which leaves the question about its importance.

Response #11. We thank Reviewer #1 for pointing this out. Figure 2b is important because it (i) documents the age–depth relationship for our age model, including Holocene ages transferred from PS111_13 to PS128_2 via XRF correlation, and (ii) shows how sampling resolution varies through time. We have revised the manuscript to explicitly reference Fig. 2b at key points where these aspects matter (lines 141, 177, 490–494).

-I would recommend scaling the y-axis scales in 2a on the left and right in the same way.

Currently the lines seem to match very well, but with a closer look one notices that the right axis is using a slightly different scale.

Response #12. Done. The rescaling has even improved the correlation.

-The caption of the figure is at various places wrong, like the blue area is 15 ka not 20, there are diamonds instead of triangles,...

Response #13. Thank you for catching the caption inconsistencies. We have corrected the figure caption to match the plotted elements and shading intervals (blue shading to 15 ka rather than 20 ka, and markers as diamonds rather than triangles). We also checked symbol colors, line styles, and panel references for consistency.

Figure 3:

-I appreciate the efforts taken to revise Figure 3.

-I appreciate the effort of the clarification in the text and caption regarding Figure 3g “Winter sea ice based on the relative abundance of diatom winter sea ice “. It is still not obvious, how someone who is not familiar with paleoceanography, as the audience of the journal might be, interpret a 12% Winter Sea Ice? Does it mean that 12% of the Weddell Sea were ice covered? Please provide even more guidance.

Response #14. We agree that the current phrasing could mislead non-specialists. The “winter sea-ice diatom species (%)” metric is a proxy: it reflects the relative abundance of diatom taxa indicative of winter sea-ice presence (*Fragilariopsis curta* and *Fragilariopsis cylindrus*, combined as the *F. curta* group) in a single sediment core from the Scotia Sea. It does not mean that 12% of the Weddell Sea was ice-covered. Rather, higher percentages indicate stronger local winter sea-ice influence (more frequent/longer seasonal sea-ice cover, proximity to the winter sea-ice edge, and associated water-mass conditions) at the core site. We have clarified both the axis label and the caption to guide non-specialist readers:

- Y-axis title in panel g of Figure 3: “Winter sea-ice diatom species (%)”
- Caption addition (lines 222–225, see below):

*“g, High percentages of winter sea-ice diatom species (*Fragilariopsis curta* and *Fragilariopsis cylindrus*, combined as the *F. curta* group) are indicative of stronger local winter sea-ice influence in the Scotia Sea at core PS67/197-1 (more frequent/longer seasonal sea-ice cover, proximity to the winter sea-ice edge, and associated water-mass conditions)^{59,60}. (lines 223-227).*

- Clarification in the main text (lines 326-327): We interpret panel g as a local proxy of winter sea-ice presence at the core site based on the relative abundance of winter sea-ice diatom indicator species, rather than as a direct percentage of Weddell Sea area covered by ice.

-The importance of the 3% threshold in Figure 3g is not clear, as it is only mentioned in the caption

Response #15. We agree that the 3% threshold was insufficiently justified and is not essential to our conclusions. We have removed the 3% line from Fig. 3g and revised the caption accordingly so that the panel is interpreted as a continuous.

-Are the blue bars in Figure 3 referring to the periods mentioned in lines 281-283, according to the information given I would assume so. If this is the case they do not match. Please clarify.

Response #16. Thank you for pointing this out. We re-checked the blue-shaded intervals in Figure 3 against the periods cited and found mismatches. We have now aligned the shading to exactly match the text-defined intervals and updated both the figure and caption to make this explicit (see lines 290-293) and Figure 3.

Minor comments:

line 20: It feels like here is a verb missing in connection with "either the stratification of the upper water column"

Response #17. Done (lines 21-22).

line 24: "The glacial polynyas formed off..." - this sentence is too speculative and not proven and I don't think it belongs in the abstract

Response #18. Yes, done. We removed and replaced it with a concise, evidence-grounded formulation that reflects our main conclusions without overreach (see below).

"This glacial polynya formed off Dronning Maud Land (DML) reflects a hybrid coastal-open-ocean polynya mode."
(lines 24-26).

line 46: here is a unit for the temperature missing

Response #19. Done (line 47).

line 100: "the low number of well-preserved..." - if this is a common problem it would be good to add a citation here.

Response #20. Done (line 105). We added citations to support this commonly noted issue; note that the introductory statements in lines 96–98 had already highlighted this as a general problem.

line 104: "se" - seems to be in the wrong place

Response #21. Done (line 107).

line 115: it would be good to introduce the abbreviation “subST”- I assume it is “subSurface Temperature”?

Response #22. Done (lines 118-119).

line 124: I assume there are brackets missing around “Osw_ivc”

Response #23. Yes, done (line 128).

line 149: “Atka Bay” is not shown in the map, either add it or rephrase caption.

Response #24. Thank you for noting this. Atka Bay was previously labeled in Fig. 1b but was hard to see. We have increased the font size and changed the label color to black to ensure clear visibility. The caption has been kept, as the map now clearly shows Atka Bay.

line 164: Is it possible to show the EDML location in Figure 1?

Response #25. Yes, done (Figure 1a).

line 180: referencing is wrong

Response #26. Done (line 186).

line 228- 230: is this supposed to be seen from Figure 3? Or where does this information originate from?

Response #27. Yes, this information comes from figures 2a, 3c, d and e. We have included the references in the text in line 239.

line 407: Here is a dot missing after Plateau.

Response #28. Done (line 449).

lines 438: wrong referencing.

Response #29. Done (line 488).

line 492+493: AIM is introduced twice

Response #30. Done. We removed the second instance (line 543-544) to eliminate the duplication.

line 512: I recommend removing the word 'closely' as the correlation between subST and AMOC is not well developed as previously discussed by the authors.

Response #31. Done (line 570-571).

Reviewer #2 (Remarks to the Author):

Having reviewed the resubmitted manuscript, I confirm that all my previous comments have been addressed with either corrections or appropriate explanations. Consequently, I have no further comments to offer. I believe it has been developed into a very good paper.

Response #32. We sincerely thank Reviewer #2 for the positive assessment and constructive feedback. We are grateful that you consider the revised manuscript to be a very good paper, and we appreciate the time and effort you invested in helping us improve it.

Reviewer #4 (Remarks to the Author):

The upper ocean water column stratification between the Antarctic Polar Front and the continental margins of Antarctica consists of a low salinity near freezing surface layer over a relatively warm salty deep water. The pycnocline separating these layers is weak and sometimes if the surface layer is salty enough, it disappears in winter, enabling deep reaching convection. The convection cools the regional deep water, while the upward convective flux of relatively warm water (+0.5 C) into the surface layer inhibits winter sea ice formation, leading to an open ocean polynya, e.g. the Great Weddell Polynya of the mid-1970s. The cooled deep water spreads across the Antarctic Circumpolar Current, ventilating into the global deep water masses. The resultant deep ocean ventilation is not exactly like Antarctic Bottom Water (AABW) that is denser and forms along the continental margin of Antarctica.

Open ocean polynyas differ from polynyas over the continental shelf (coastal polynyas) that are forced by katabatic winds blowing newly formed sea ice seaward (exporting freshwater), leaving behind a cold, salty (dense) shelf water column, which upon export to the deep ocean forms a key ingredient of the classic global spreading AABW.

A factor in the formation of open ocean polynya, such as the Weddell Polynya of the mid-1970s is reduced precipitation and increased cyclonic circulation of the Weddell Gyre, which occurs during negative Southern Annual Mode (-SAM: Gordon et al, 2007, which Pinho et al cite).

Pinho et al using sea floor sedimentary data investigate the upper ocean stratification in the areas of the Dronning Maud Land Polynya (~6.5°W longitude, over the continental slope) during the last glacial period, 75K to 20K years ago. Dronning Maud Land Polynya occurs within the westward flowing coastal slope current of the Weddell Sea cyclonic Gyre.

As reported in the literature (Gordon et al, 2007; de Lavergne et al, 2014; Gordon 2014, which Pinho et al cite), southern ocean polynyas, notably the Weddell Sea Polynya, may have been more common during the glacial periods, and will become less common in the present warming climate.

Investigating the southern ocean polynyas during glacial period can lead to better understanding of global ocean deep water ventilation. Thus I view the Pinho et al as an important contribution that will encourage further research into the temporal behavior of southern ocean polynya for the present era and geological past.

The Pinho et al manuscript is a revision from an earlier ms. I was not a reviewer of that initial submission. I reviewed the 'rebuttal' and revised ms. The authors responded to the extensive initial review (and as usually the peer review, led to a much improved presentation).

While I endorse publication of the Pinho et al study there is an issue that limits the links between the Great Weddell Polynya and the . As noted by Reviewer #1 (point #6), the Dronning Maud Land Polynya is closer to the continental margin of Antarctic than the Great Weddell Polynya of the mid-1970s, which occurs within center of the Weddell Sea cyclonic gyre. The Dronning Maud Land Polynya is more like the coastal polynyas than the open ocean polynya and may not be a perfect 'precursor' 1970s open ocean event.

Pinho et al rebuttal of Reviewer 1 point stating: 'Our results therefore indicate a hybrid glacial polynya system where open-ocean deep convection processes were enhanced by katabatic wind forcing from the nearby expanded ice sheet during the last glacial period.' And they added the following clarification in lines 396-402:

"During glacial periods, when grounded ice sheets advanced across the Antarctic continental shelves, coastal shelf polynyas similar to those observed today 17,18,81 were likely absent. Instead, the ice advance would shift the occurrence of glacial polynyas northward to regions above the deep ocean (Fig. 4a). Although the polynya itself was located in open ocean, strengthened katabatic winds blowing across the expanded ice sheet are considered an

important mechanism for the polynya formation.”

They may be right in relating the glacial age Dronning Maud Land Polynya to the Great Weddell Polynya, but more support for this is needed. I suspect that during the glacial age, the strong katabatic winds reached out to the continental slope (as glacial ice covered most of the continental shelf), which coincided with strong -SAM making the central Weddell Sea gyre more prone to open ocean polynyas (Gordon 2014).

I suggest that Pinho et al, make a stronger case for linking the Dronning Maud Land Polynya to the Great Weddell Polynya.

Response #33. We thank Reviewer #4 for the positive assessment and constructive feedback. We also appreciate the suggestion to make a stronger, more explicit linkage between the DML Polynya and the Great Weddell Polynya. In the revised manuscript, we clarify this connection by bringing together proxy coherence, geographical overlap, and mechanism. First, consistent proxy signals across PS128_2 (our study) and the Bungenstock Plateau cores (PS1506, PS1388) indicate repeated polynya-favorable conditions within the historical Great Weddell Polynya corridor (Mackensen et al., 1989; 1994). Second, we add an explicit spatial paragraph with a quantitative distance estimate in “Glacial occurrence of the Great Weddell Polynya,” supported by a small overlay in Fig. 1a showing the 1974–1976 Great Weddell Polynya outlines (Carsey, 1980) core locations (Fig. 1b), and the 68–69°S band. Third, we add a mechanistic paragraph in “Glacial Forcing and feedback mechanisms” that links –SAM/gyre dynamics to a hybrid coastal–open-ocean geometry under glacial boundary conditions. Specifically for the hybrid coastal–open-ocean polynya discussion, please see Response #9.

Section “Occurrence of the Glacial DML Polynya”:

“The increased productivity during glacial periods extended at least to the northern edge on Bungenstock Plateau ²⁸, as indicated by sediment records from sites PS1506 (68.7°S; 2,426 m) and PS1388 (69.0°S; 2,526 m; Fig. 1b), both north of PS128_2 (Fig. 1b). This implies that the Glacial DML Polynya was present during the last glacial period as well as in preceding glacials extending northward to at least 68.7°S.” (lines 362-367).

“Taken together, these evidences point to the existence of the Glacial DML Polynya, which extended from the ice edge in the south northwards to at least 68.7° S at ~5–6° W. This northern position overlaps with the southern boundary of the historic Great Weddell Polynya. Observations from the mid-1970s place the southern edge of the Great Weddell Polynya near ~69°S at ~5°W (Fig. 1a), and waters around ~68°S remained polynya-prone even in 1977 (a year without an active polynya), indicating a sensitive preconditioning zone where ocean heat can thin or remove sea ice ⁴⁴.

Relative to this modern baseline, a “glacial Great Weddell Polynya” likely stretched to at 69.4°S (PS128_2), representing only a modest expansion of ~45 km southward beyond the modern southern edge near 69°S. This would

place the Glacial DML Polynya within a plausible “glacial Great Weddell Polynya” environment. Such excursions fall well within observed behavior, as the modern Great Weddell Polynya can shift by ~100 km over just three days⁴⁴.” (lines 375-387).

Section: “Glacial forcing and feedback mechanisms”:

“Since background climate conditions control the Great Weddell Polynya, we argue that glacial boundary conditions amplified regional preconditioning, favoring a strong and extensive Glacial DML Polynya probably as large as the modern Great Weddell Polynya. Concurrently, ice advance towards the shelf break and katabatic winds would have shifted glacial polynyas northward into a hybrid coastal–open-ocean mode. Under glacial conditions, a southward extension of the glacial Great Weddell Polynya into our study region would be plausible, since this setting would be physically well founded. (lines 465-471).

We refrain from a strict one-to-one claim; nevertheless, the overlapping footprint, the established preconditioning pathway, and the consistent proxy signals across PS128_2, PS1506, and PS1388 together provide a solid, quantitatively grounded rationale for linking the hybrid Glacial DML polynya to Great Weddell Polynya open-ocean activity. We now refer to this hybrid polynya as the Glacial DML Polynya (see lines 301–302), and we have revised the text accordingly. We believe the revisions now communicate this connection effectively.

Additionally, the revised manuscript adopts a more cautious tone, framing the Glacial DML Polynya as a hybrid polynya mode rather than a perfect precursor to the 1970s open-ocean events as observed by both Reviewers #1 and 4.

See for example text in lines 555-567 in the conclusion section:

“Our proxy data from site PS128-2 on the Bungenstock Plateau indicate that the recurring presence of the Glacial DML Polynya was a typical feature of the last ice age. Though the Glacial DML Polynya may share some similarities with the openings of the Great Weddell Polynya in the 1970s and the Maud Rise Polynya in 2016 and 2017 they are the result of different processes and impacts on different timescales. While the formation of the Glacial DML Polynya is consistent with a hybrid polynya mode that combines characteristics of a coastal polynya and an open ocean polynya, the offshore events of the last century are primarily open-ocean driven. We therefore propose that the modern occurrences are not a direct remnant of the Last Ice Age, but rather an expression of a regionally persistent sensitivity to atmosphere-ice-ocean interactions, which was strongly expressed during the last glacial when polynya-deep-convection operated on millennial timescales.” (lines 556-567).

I suggest a title that mentions the glacial age Dronning Maud Land Polynya.

Response #34. We thank Reviewer #4 for this helpful suggestion. To directly address it while maintaining our mechanistic emphasis, we have adopted the reviewer’s guidance and revised the title to:

“Millennial-to-orbital-scale subsurface ocean warming and Polynya formation off Dronning Maud Land during the last glacial”

This title explicitly highlights the glacial-age DML Polynya, clarifies the dual emphasis on polynya formation and stratification, and signals the temporal scale of our analysis (millennial to orbital scales). We are confident this revision better reflects the scope and contributions of the manuscript and fully satisfies the reviewer's request.

References

- Carsey, F.D., 1980. Microwave Observation of the Weddell Polynya. *Mon. Weather Rev.* 108, 2032–2044. [https://doi.org/10.1175/1520-0493\(1980\)108<2032:MOOTWP>2.0.CO;2](https://doi.org/10.1175/1520-0493(1980)108<2032:MOOTWP>2.0.CO;2)
- Cheon, W.G., Gordon, A.L., 2019. Open-ocean polynyas and deep convection in the Southern Ocean. *Sci. Rep.* 9, 6935. <https://doi.org/10.1038/s41598-019-43466-2>
- Dufour, C.O., Morrison, A.K., Griffies, S.M., Frenger, I., Zanowski, H., Winton, M., 2017. Preconditioning of the Weddell Sea Polynya by the Ocean Mesoscale and Dense Water Overflows. *J. Clim.* 30, 7719–7737. <https://doi.org/10.1175/JCLI-D-16-0586.1>
- Gordon, A.L., 1978. Deep Antarctic Convection West of Maud Rise. *J. Phys. Oceanogr.* 8, 600–612. [https://doi.org/10.1175/1520-0485\(1978\)008<0600:DACWOM>2.0.CO;2](https://doi.org/10.1175/1520-0485(1978)008<0600:DACWOM>2.0.CO;2)
- Kurtakoti, P., Veneziani, M., Stössel, A., Weijer, W., 2018. Preconditioning and Formation of Maud Rise Polynyas in a High-Resolution Earth System Model. *J. Clim.* 31, 9659–9678. <https://doi.org/10.1175/JCLI-D-18-0392.1>
- Mackensen, A., Grobe, H., Hubberten, H. W. and Kuhn, G. Benthic foraminiferal assemblages and the $\delta^{13}\text{C}$ -signal in the Atlantic sector of the Southern Ocean: Glacial-to-interglacial contrasts, in *Carbon Cycling in the Glacial Ocean: Constraints on the Ocean's Role in Global Change*, edited by R. Zahn et al., pp. 105–114, Springer, Berlin (1994).
- Mackensen, A., Grobe, H., Hubberten, H.-W., Spiess, V., Fütterer, D., 1989. Stable isotope stratigraphy from the Antarctic continental margin during the last one million years. *Mar. Geol.* 87, 315–321. [https://doi.org/10.1016/0025-3227\(89\)90068-6](https://doi.org/10.1016/0025-3227(89)90068-6)
- Martinson, D.G., Killworth, P.D., Gordon, A.L., 1981. A Convective Model for the Weddell Polynya. *J. Phys. Oceanogr.* 11, 466–488. [https://doi.org/10.1175/1520-0485\(1981\)011<0466:ACMFTW>2.0.CO;2](https://doi.org/10.1175/1520-0485(1981)011<0466:ACMFTW>2.0.CO;2)

Dear Reviewers,

We thank Reviewers for their valuable comments/suggestions, which we have thoroughly analyzed and implemented. They are very constructive and appropriate. We believe that through their implementation the manuscript has greatly improved. We provide Word document files with tracked changes of the revised version of the manuscript. To simplify the evaluation of the performed modifications we (i) copied below in black/italic the comments/suggestions from Reviewers, and (ii) provided a detailed response to each comment/suggestion in blue/not-italic. All line numbers referenced in our responses correspond to the clean revised version. All edits can be found in the tracked-changes version of the manuscript.

REVIEWER COMMENTS

Reviewer #1 (Remarks to the Author):

Review of the revised submission "Millennial-to-orbital-scale subsurface ocean warming and Polynya formation off Dronning Maud Land and during the last glacial" by Pinho et al.

I thank the authors for revising their manuscript, implementing all previously made comments. During the last read I found one typo: "interrupted" in line (63) and I guess "glacial Weddell polynya" in line 520 has been missed during the renaming to "(glacial) DML polynya" from the previous version. Otherwise, I have no further comments on the paper.

Response #1. We thank reviewer #1 for the final remarks. Done (lines 63 and 538).

Reviewer #4 (Remarks to the Author):

I support publication. The author have received constructive comments from the reviewers and have responded effectively. Hopefully their paper will generate lots of interest in the spatial and temporal variability of the multitude of polynyas types in the southern ocean.

I do have a general concern about the primary driver of the Dronning Maud Land Polynya:

The authors suggest that warmer temperature of the WDW drives the polynya: warmer WDW injected into the surface layer over a topographic feature, spurs a polynya, and increased

precipitation over the coastal region. Why is WDW warming during the polynya (MIS4 and LGM) phase? The authors suggest that more extensive sea ice cover to the north enables warmer WDW spreading southward within the eastern limb of the Weddell Sea Gyre to reach the Dronning Maud Land region. They might be right.

Alternatively, warmer WDW might be due to increased southward flux of circumpolar deep water (CDW) heat across the ACC? This could be driven by stronger west to east winds over the ACC, producing a stronger ACC and increased eddy activity.

Or as suggested in the literature the saltier surface layer, due to decreased precipitation (-SAM) weakens the pycnocline and can initiate deep convection, which injects heat into the surface layer and a polynya.

So which is the primary driver of the Dronning Maud Land polynya: saltier surface water or warmer WDW?

Response #2. We thank Reviewer #4 for the positive assessment and final remarks. We interpret the Warm Deep Water (WDW) warming as a consequence of northward-expanded sea-ice cover during glacial conditions, which would suppress vertical mixing and reduce air–sea heat exchange, favoring the retention of ocean heat at subsurface depths.

We also agree with the reviewer’s alternative explanation that a strengthened Antarctic Circumpolar Current (ACC) could have enhanced poleward heat transport into the Weddell Gyre, thereby contributing to warmer WDW in the Dronning Maud Land sector. A very recent study published in Nature Geoscience (Wu et al., 2026) supports this mechanism, and we have incorporated this additional mechanism in the revised manuscript (lines 340–355 and 464–465).

“Alternatively, warmer WDW temperatures may reflect a stronger southward transfer of CDW heat across the Antarctic Circumpolar Current (ACC) in the Atlantic–Indian sector of the Southern Ocean during MIS 4 and 2. Wu et al.⁷³ suggested that under low obliquity conditions, a more northerly ACC position around ~45°S placed the current directly under an equatorward-displaced Southern Hemisphere westerly belt and coincided with enhanced meridional density contrasts. These changes likely increased isopycnal tilt, strengthening the glacial ACC in the Indian sector⁷³. In addition, a coeval reduction in the leakage of the Agulhas Current into the South Atlantic would have promoted the buildup of heat and salinity in the South Indian Ocean, further amplifying the meridional density gradient⁷³. Taken together, stronger density gradients and increased surface heat-flux anomalies associated with sea-ice expansion and intrusions of warm water would have intensified buoyancy forcing and, in turn, reinforced ACC transport in the South Indian Ocean during MIS 4 and 2⁷³. Since CDW is supplied to the Weddell Gyre from the Indian-sector Southern Ocean, a strengthened ACC would likely enhance poleward heat transport, increasing the delivery of warm CDW into the gyre and thereby promoting warmer WDW.” (lines 340-355)

“Additionally, an intensified ACC would have boosted poleward heat delivery to the Weddell Gyre, further warming WDW⁷³.” (lines 464-465)

At present, we cannot unambiguously identify a single primary driver of glacial Dronning Maud Land polynya formation, because both proposed mechanisms: a saltier surface layer (atmospheric forcing) and warmer WDW (oceanic forcing) are likely linked to the same glacial background climate condition. We therefore consider it most likely that both processes acted together to weaken upper-ocean stratification and promote conditions conducive to polynya development during the last glacial period.

“Subsurface” is often used. Is this the surface layer (ASW) above the pycnocline? The pycnocline? Or the WDW? I think they imply the subsurface is WDW. Why not call it WDW?

Response #3. Thank you for noting the potential ambiguity of “subsurface.” In our study, the planktonic foraminifera proxy reflects changes in the vertical structure of the upper ocean at ~150 m water depth, specifically the relative vertical position of Antarctic Surface Water (ASW) and WDW through time. During intervals associated with polynya conditions, the proxy indicates an upward shift of WDW influence toward shallower depths, whereas during more stratified intervals the ASW signature is more prominent at ~150 m because WDW resides deeper. For this reason, we used the more general term “subsurface” to denote the water mass signal recorded at the proxy depth, which can reflect either a shoaling WDW influence or a deepening of the ASW layer depending on the state (see lines 245-253)

- I like figure 4. I wonder if map panels might be added, showing WDW spreading for glacial and for non-glacial periods?

Response #4. We appreciate your positive feedback on Figure 4 and your suggestion to add map panels illustrating WDW spreading under contrasting climate states. We agree this could be a valuable visualization in principle. However, because our study focuses on the last glacial interval and does not include interglacial periods (non-glacial periods) or independent constraints sufficient to robustly reconstruct spatial WDW pathways under multiple climate states, we prefer not to add additional map panels that could imply a level of spatial reconstruction beyond what our data directly support. We have therefore kept Figure 4 as is.

References

Wu, S., Mazaud, A., Michel, E., Erb, M.P., Stocker, T.F., Amsler, H.E., Le Tallec-Carado, P., Lamy, F., Jaccard, S.L., 2026. Zonally asymmetric changes in the Antarctic Circumpolar Current strength over the past million years. *Nat. Geosci.* 19, 201–208.
<https://doi.org/10.1038/s41561-025-01901-2>